# Discovering Novel LLM Experts via Task-Capability Coevolution

**Andrew Dai**[*]  **Boris Meinardus**[*]  **Ciaran Regan**  **Yingtao Tian**  **Yujin Tang**
Sakana AI

## Abstract

Frontier model developers aim to train models continually to possess emergent, diverse capabilities. To extend capabilities, the current pre-training and post-training paradigm requires manually starting training runs with static datasets or reward functions every time. Addressing this limitation, our work pursues the insight that open-endedness (via the coevolution of models and tasks) can discover models with increasingly novel skills in a single run. We introduce a new model development framework that extends coevolution to large language model (LLM) discovery, open-ended *Assessment Coevolving with Diverse Capabilities* (AC/DC). AC/DC evolves both LLMs via model merging and natural language tasks via synthetic data generation. AC/DC discovers growing archives of LLMs that surpass the capabilities of larger LLMs while taking up less GPU memory. In particular, our LLM populations achieve a broader Coverage of expertise than other curated models or baselines on downstream benchmarks, without *any* explicit benchmark optimization. Furthermore, AC/DC improves Coverage over time, continually innovates on tasks and models, and improves performance in multi-agent best-of-N selection. Our findings highlight the potential of coevolution as a means of discovering broader sets of capabilities from base LLMs. Overall, AC/DC brings us one step closer to a profoundly new paradigm of LLM development, where continual improvements to the diversity of model capabilities can be accelerated by leveraging existing models as stepping stones to increasingly powerful models.

## 1 Introduction

LLMs and foundation models (Brown et al., 2020; Bommasani et al., 2021; Kaddour et al., 2023) underpin key advances in AI for open-ended discovery and innovation (Nguyen et al., 2016; Lehman et al., 2023; Zhang et al., 2023; Lu et al., 2024). Such innovation capacity in future AI systems, innate to human civilization, would not only have profound implications for automated scientific discovery, but would also accelerate AI research itself. How do we get closer to LLMs as engines of knowledge accumulation and serendipitous discovery, with the ability to stumble upon greatness (Stanley & Lehman, 2015) and drive paradigm shifts (e.g., the Transformer (Vaswani et al., 2017))? Additionally, how do we imbue LLMs with innovation capacity and broader capabilities, given the prohibitively expensive costs and inaccessibility of running bigger and bigger models (Pan & Wang, 2025) or obtaining more data (Muennighoff et al., 2023), especially for typical ML researchers?

While excitement grows around LLMs for scientific discovery (Romera-Paredes et al., 2023; Novikov et al., 2025), the current paradigm of LLM development struggles to keep up with the accumulation of knowledge on learnable or discovered data. Developers must continually adapt to incremental improvements in static datasets (Albalak et al., 2024; Kandpal et al., 2025), environments (Intellect, 2025; Lambert et al., 2024; Novikov et al., 2025), learning algorithms (Shao et al., 2024; Liu et al., 2025b), and architectures (Yang et al., 2025; Muennighoff et al., 2024), to push the boundaries of frontier models. Continually training on synthetic data (Wang et al., 2023d; Xu et al., 2023; Maini et al., 2024; Havrilla et al., 2024) and broad-domain reward signals (Zhao et al., 2025) brings us closer to self-improving LLMs. Still, only one model is produced at a time. Trusting a single big static model to solve all real-world problems would therefore be challenging, due to concerns about fractured entangled representations (Kumar et al., 2025) and costs (Li et al., 2025).

---

[*]Equal contribution. Correspondence: `adai[at]tcd.ie`, `boris.meinardus00[at]gmail.com`

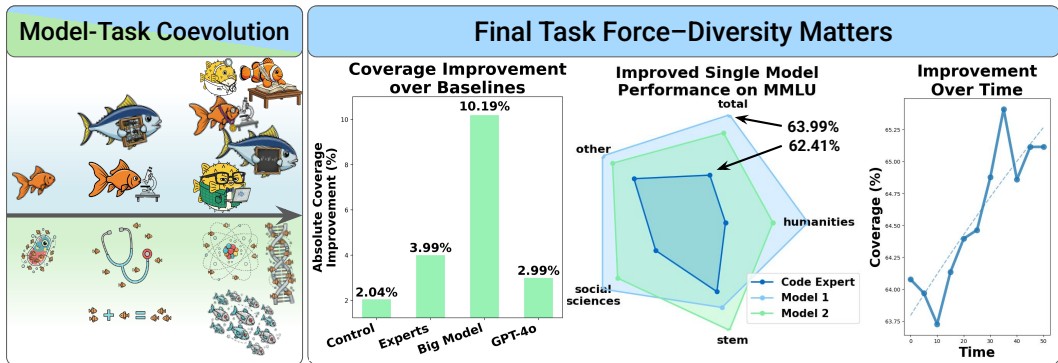

Figure 1: **Method Overview.** AC/DC coevolves an increasing set of diverse LLMs alongside an increasingly diverse and complex set of tasks, measuring the discovered models' capabilities. Our discovered collective of models (across different model families tested) covers more skills than baselines across a wide range of benchmarks. Moreover, AC/DC discovers improved single model performance (as seen by MMLU (Hendrycks et al., 2021a) performance) and demonstrates improvement over time (shown as an average of MMLU and MMLU Pro (Wang et al., 2024) accuracy).

In contrast to individual models, collective intelligence (CI) (exemplified by human civilization) is capable of endlessly achieving feats far greater than any single human could (Mitchell, 2009). CI has even inspired new paradigms in AI (Ha & Tang, 2022) and multi-agent LLM systems (Liang et al., 2023; Inoue et al., 2025), making them more robust during test-time scaling. By discovering whole collectives of small and accessible LLMs with diverse capabilities, we can overcome the limitations and weaknesses of any single trained model or the need to train different models separately.

To overcome the challenge of CI discovery, *open-endedness* (OE) is an emerging paradigm aiming for never-ending discovery via open-ended algorithms (Stanley et al., 2017). Pursuing such AI-generating algorithms (Clune, 2020), open-ended coevolution takes inspiration from the creativity explosion of natural evolution and human innovation, and pursues ever-changing learning environments for populations of increasingly intelligent agents to gain diverse capabilities (Wang et al., 2019; Dennis et al., 2020). Leveraging recent advances in OE through LLMs (Faldor et al., 2024; Aki et al., 2024; Nisioti et al., 2024), we introduce a new framework to discover a whole population of expert LLMs through open-ended *Assessment Coevolving with (/w) Diverse Capabilities* (AC/DC). AC/DC combines evolutionary model merging (Akiba et al., 2025) and synthetic data generation (Lu et al., 2025) to enable LLM populations to continually adapt to novel challenges that are generated, while satisfying minimal criteria for model and task quality (Brant & Stanley, 2017), all without explicit benchmark optimization (Lehman & Stanley, 2011a).

Following coevolution via AC/DC, we discovered a broad array of LLMs with diverse expertise and response styles that solve synthetic tasks spanning engineering, the sciences, and creative writing. When we selected a fixed-size subset of LLMs that make up the broadest skill Coverage on synthetic tasks, and then evaluated them on various LLM benchmarks that are out-of-distribution (OOD) to synthetic tasks at test time, we found that our population of smaller evolved LLMs (with a combined lower LLM parameter count than compared baseline models) was able to solve and cover more benchmark tasks than bigger LLMs of the same model family, as well as the initial seed LLM population. Our results also suggest that our LLM collectives surpass or reach GPT-4o (Hurst et al., 2024) levels of knowledge covered with a significantly lower proportion of collective model parameters. Furthermore, a single evolved model achieved better MMLU (Hendrycks et al., 2021b) performance than the best starting seed model (cf. Fig. 1), more iterations of coevolution led to continually improved model population performance at test time, and cooperative final answer (Best-of-N) selection was more often successful with our LLM collectives than with baselines.

In short, our main contributions are (1) the AC/DC method applying coevolution to a novel joint LLM-and-synthetic-data discovery framework, (2) a demonstration of autonomous discovery of diverse LLM experts solving OOD tasks more broadly than baseline methods (some directly optimizing for benchmarks) and off-the-shelf models, and (3) evidence of a path to open-ended improvement of LLMs without explicit benchmark optimization, through an analysis of AC/DC.

## 2 BACKGROUND AND PRELIMINARIES

This section introduces key concepts central to AC/DC: model merging operations that enable evolutionary discovery of LLM populations, Coverage metrics that quantify collective capabilities, and quality-diversity (QD) principles that guide our coevolutionary process.

**Evolutionary Model Merging.** Model merging combines multiple existing LLMs to produce new models with lower resource requirements than training from scratch (Wortsman et al., 2022; Ilharco et al., 2023). Akiba et al. (2025) introduced evolutionary model merge (EvoMerge), which automates the merging process through evolutionary optimization using CMA-ES (Hansen & Ostermeier, 2001). Building on this foundation, we employ two key evolutionary operations:

*Crossover:* We sample two parent LLMs randomly and merge them using weighted linear interpolation of their task vectors, as done in CycleQD (CQD) (Kuroki et al., 2025). The task vector $\tau_{pi} = \theta_{parent_i} - \theta_{base}$ represents the difference between parent LLM $i$ and a base LLM (see Appendix App. M for more details).

*Mutation:* We generalize existing mutation operations by applying noise to the singular values of weight matrices in merged LLMs. For each weight matrix $W$, we compute its singular value decomposition $W = U\Sigma V^T$ and apply perturbations to the first $k$ singular values in $\Sigma$ before reconstruction, loosely inspired by Sun et al. (2025). This approach modifies the representational structure while preserving the overall weight matrix geometry (see Appendix App. M for more details).

**Coverage Metric.** Coverage measures the collective problem-solving capacity of LLM populations. Given $Q$ total number of questions and $N$ number of LLMs, Coverage is:

$$\text{Coverage} = \frac{1}{Q} \sum_{q=1}^{Q} \left( \bigvee_{i=1}^{N} (x_{q,i} = y_q) \right) \tag{1}$$

where $x_{q,i}$ is the output of LLM $i$ for question $q$, $y_q$ is the ground truth answer for question $q$, and $\bigvee_{i=1}^{N}$ denotes the logical OR operation over all $N$ LLMs. Coverage quantifies whether at least one LLM in the population solves each problem, capturing the collective intelligence potential of diverse LLM ensembles. Unlike individual LLM accuracy, Coverage emphasizes complementary capabilities that emerge from LLM diversity.

**Skill Vectors.** We represent LLM capabilities through binary skill vectors, where each indexed element indicates task completion status. They serve as behavioral signatures that enable direct comparison of LLMs without predefining niches (as in MAP-Elites (Mouret & Clune, 2015b)). The distance between skill vectors informs the diversity of complementary LLM capabilities.

**Quality-Diversity (QD).** QD generates collections of diverse, high-quality solutions (Pugh et al., 2016; Lehman & Stanley, 2011b), unlike traditional optimization, which seeks a single optimal solution. For model selection, we apply Dominated Novelty Search (DNS) (Bahlous-Boldi et al., 2025), a recent QD algorithm particularly suited to skill vector representations (similar to Meyerson & Miikkulainen (2017)). DNS computes local competition fitness $\tilde{f}_i$ by measuring each solution's distance from better-performing solutions in the descriptor space. For solution $i$, $\tilde{f}_i$ is computed as:

$$\tilde{f}_i = \begin{cases} \frac{1}{k} \sum_{j \in \mathcal{K}_i} d_{i,j} & \text{if } |\mathcal{D}_i| > 0 \\ +\infty & \text{otherwise} \end{cases} \tag{2}$$

where $\mathcal{D}_i$ contains solutions fitter than solution $i$, $\mathcal{K}_i$ contains indices of $k$ solutions in $\mathcal{D}_i$ with smallest distances $d_{i,j}$ between solutions $i$ and $j$. Local competition encourages diversity by rewarding solutions that are distant from higher-performing neighbors in the behavioral space.

**Open-Ended Coevolution.** Brant & Stanley (2017; 2020) demonstrate that defining minimal criteria (MC) for both agents and environments enables more open-ended outcomes in coevolution, filtering out undesired outcomes while enabling exploration to flourish. AC/DC coevolves populations of LLMs and synthetic tasks, where models must satisfy quality thresholds while maximizing quality and behavioral diversity through their skill vector representations. This creates a dynamic environment where increasingly sophisticated capabilities can emerge through the interplay between model evolution and task complexity. Related work discussion in App. G highlights AC/DC as a novel system combining concepts from various fields.

# 3    AC/DC: ASSESSMENT COEVOLVING WITH DIVERSE CAPABILITIES

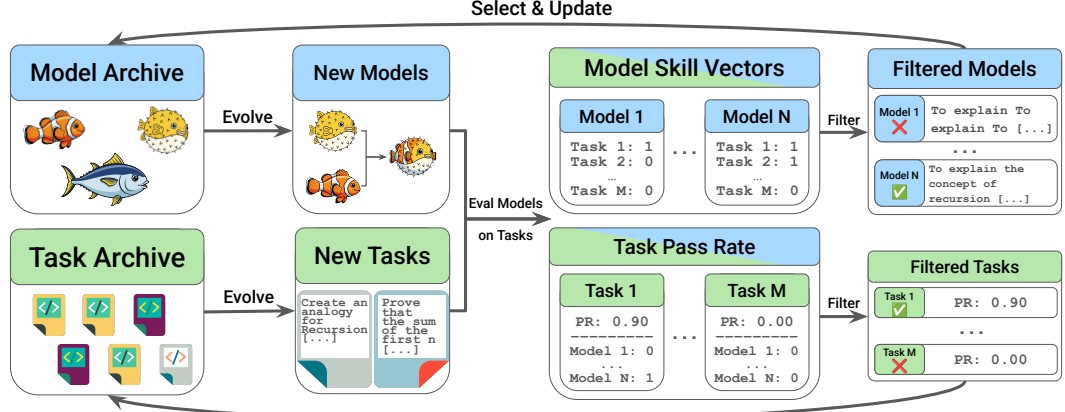

Figure 2: **Algorithm Overview.** AC/DC continuously coevolves a model (LLM) archive and a synthetic task archive. LLMs are evolved using model merging crossover, and weight noising as a mutation operation. Tasks are evolved using a large scientist LLM that transforms existing task descriptions to generate increasingly novel and complex tasks. Models are evaluated on this data. We then compute a skill vector (i.e., signature of quality and diversity) for each model and a pass rate for each task. Based on those, we first apply minimal criterion (MC) filters (gibberish LLM filter, impossible task filter) and then select the models and tasks to update the archives, respectively.

We describe an open-ended algorithm for automatically discovering diverse LLMs that can collectively cover a wide range of skills. AC/DC coevolves two archives: an LLM archive $\mathcal{A}_M$ selected via DNS on skill vectors, and a synthetic active task archive $\mathcal{A}_Q$ containing an increasingly complex and novel set of challenges that the LLM archive must solve. We illustrate the algorithm in Fig. 2 and provide the pseudocode below (Algorithm 1). Inspired by Brant & Stanley (2017), we highlight minimal criteria (MC) for both LLMs and tasks. For further details, see Appendix App. N.

---

**Algorithm 1** AC/DC: Assessment Coevolving with Diverse Capabilities
---

1: **Initialize:** Model archive $\mathcal{A}_M \leftarrow$ seed and init models, Task archive $\mathcal{A}_Q \leftarrow$ seed and init tasks
2: **for** $g = 1$ **to** $G$ **do**                                                     ▷ Model Evolution Phase
3:      $P \leftarrow$ SELECTPARENTS($\mathcal{A}_M$)                                      ▷ Select $P$ parents
4:      $O \leftarrow$ CROSSOVERMUTATE($P, N$)                                     ▷ Generate $N$ offspring
5:      $E \leftarrow$ EVALUATE($O, \mathcal{A}_Q$)        ▷ Evaluate skill vectors, get responses to tasks; App. F.2
6:      $T \leftarrow$ GIBBERISHFILTER($E$)               ▷ Trim away/discard degenerate models; App. F.5
7:      $\mathcal{A}_M \leftarrow$ DNSUPDATE($\mathcal{A}_M, T$)                            ▷ Select top $M$ models; App. A.3.1
8:      **if** $g \bmod G_{task} = 0$ **then**                                          ▷ Task Evolution Phase
9:          $Q \leftarrow$ GENERATETASKS(scientist LLM)                       ▷ Generate $N_{gen}$ tasks; App. F.1
10:         $Q_{nov} \leftarrow$ NOVELTYFILTER($Q$)                              ▷ Remove similar tasks; App. F.4
11:         $Q_{valid} \leftarrow$ VALIDATETASKS($Q_{nov}$)                       ▷ Reflection & validation; App. F.1
12:         $\mathcal{A}_Q, \mathcal{A}_{Q_g} \leftarrow$ UPDATETASKARCHIVE($\mathcal{A}_Q, \mathcal{A}_{Q_g}, Q_{valid}$)   ▷ $\mathcal{A}_{Q_g}$ is global task archive
13:         REEVALUATEARCHIVE($\mathcal{A}_M, Q_{valid}$)               ▷ Update skill vectors with new task pool
14:     **end if**
15:     SAVEARCHIVES($\mathcal{A}_M, \mathcal{A}_Q, g$)
16: **end for**
17: **return** SELECTTASKFORCE($\mathcal{A}_M, \mathcal{A}_{Q_g}$)                          ▷ Select $N_{tf}$ diverse models

---

**Model Archive Evolution.** Evolution begins with at least three seed LLMs, representing fine-tuned versions of the same base architecture. We maintain $M$ active LLMs, i.e., LLMs considered as parents for the next generation (as in DNS). We also save a historical archive of LLMs every $G_{task}$ generations (before task adaptation), as candidates for a future task force. Given the existing parent population in $M$, we apply crossover and mutation operators as described in Sec. 2, producing

$N$ offspring and yielding $M + N$ candidates for evaluation (Lines 3-4). Each candidate LLM is evaluated on our synthetic task archive and assigned skill vectors (Line 5). We apply a novel MC filter called the "gibberish filter" to eliminate degenerate models by sampling outputs from the LLM for three random tasks each and employing a judge LLM to assess text coherence (Line 6). For the remaining models, we employ DNS to compute adjusted local competition scores $\tilde{f}$ (cf. App. A.3.1). We retain the highest-fitness model and select the remaining $P - 1$ models with top $\tilde{f}$ scores such that we have at most $M$ active models (Line 7).

**Task Archive Evolution.** Alongside LLM agents, we coevolve an increasingly challenging and diverse set of tasks. We employ a large scientist LLM to synthesize tasks in line with METR Task Standard Team (2024) (but simplified), where each task comprises a question-answer pair with an accompanying scoring function defined in Python (Line 9; App. F.1). We extend Lu et al. (2025) with a code extraction tool that enables robust evaluation of tasks requiring code generation, allowing the scientist LLM to programmatically parse and execute subject model responses. We maintain at most $Q_{max}$ active tasks per generation that are used to evaluate the current generation of models, alongside a global task archive updated every $G_{task}$ generations. We employ two vector databases for efficient similarity search: one for active tasks (newest) and another for the global archive.

Task evolution begins with $N_{seed}$ manually curated seed tasks (cf. App. E.3) and generates $N_{init}$ initial tasks through our evolution pipeline (Lines 9-13): (1) *Task Proposal Phase* samples a parent task and three random tasks from the active task database. Based on the parent task's difficulty profile—determined by average pass rates across the current model population—we classify adaptation requirements as: increase difficulty, decrease difficulty, or generate a novel variant. The scientist LLM receives the parent task, three random reference tasks, and an adaptation type to generate a candidate task. (2) *Novelty Filtering* retrieves the three most similar tasks from the global archive using cosine similarity in embedding space. A judge LLM then determines whether the proposed task introduces sufficient novelty relative to existing tasks. (3) *Reflection and Validation* applies iterative refinement through self-evaluation cycles where the scientist LLM first attempts to solve its own generated task, and we execute the scoring function to identify implementation issues. Compilation errors trigger automatic correction with error feedback, while logic-based errors prompt task refinement. (4) *Quality Assurance and MC* implements additional filtering to remove impossible tasks that no LLM was able to solve, replacing them with their parent tasks. Accepted tasks are added to both the global archive and the active tasks. See App. E.4 for generated example tasks.

**Model Selection for Downstream Evaluation.** After coevolution over multiple generations, we select $N_{tf}$ models (from the historical archive) for our *task force* that maximize the number of correctly solved tasks across our synthetic task distribution $\mathcal{A}_{Q_g}$ (Line 17) (see App. D.2 for experiments with different selection strategies). This selection process operates independently of downstream benchmarks, avoiding optimization pressure and maintaining model generalization for OOD domains.

## 4 QUANTITATIVE RESULTS

We compare our task force Coverage (Eq. (1)) against several baseline approaches. We evaluate on a diverse set of benchmarks covering general knowledge, math, STEM, and code (see App. A.4.1 for details). See App. A.2 for model specifications and App. A.1 for hyperparameters.

**Baselines Setup.** We compare against four baselines: (1) *Experts (N=3):* Hand-selected instruct models (code, math, general) prompted once each with temperature 0 (for a discussion on N=8 experts, see App. D.4). (2) *Control (N=3/N=8):* The general instruct model prompted 3 or 8 times with temperature 0.7. (3) *Big Model:* A single large instruct model prompted once with temperature 0. (4) *GPT-4o:* Prompted once as the Big Model.

**Best-of-N Selection Setup.** Next to Coverage, we also evaluate Best-of-N (BoN) single-answer selection from multiple candidates using standard benchmark versions, testing whether Coverage improvements translate to practical deployment scenarios. We implement three techniques for the three benchmark types (MCQ, math, code). For further details, see App. A.4.

**Coverage.** Tab. 1 presents Coverage results across four base model families (see App. B.1 for details), revealing important patterns in AC/DC's performance across different architectures and scales. AC/DC works on mulitple model families, achieving positive improvements on average across all model families and configurations (+2.04% to +10.19% across comparisons). Qwen 2.5

Table 1: **AC/DC (ours) Coverage improvement** across different models. Results show average performance improvement across all benchmarks for N=3 and N=8 configurations over the respective baseline. Gains are significant in most individual and aggregated (cf. App. K).

| Base Model | vs Experts N=3 (%) | vs Control (%) | | vs Big Model (%) | | vs GPT-4o (%) | |
|---|---|---|---|---|---|---|---|
| | | N=3 | N=8 | N=3 | N=8 | N=3 | N=8 |
| Qwen2 7B | +2.06 | -0.45 | -1.04 | +0.69 | +8.83 | -6.08 | +2.05 |
| Qwen2.5 7B | +4.40 | +0.40 | +0.61 | +3.85 | +9.78 | +1.02 | +6.95 |
| Qwen3 14B | -0.21 | +0.49 | +1.54 | +4.22 | +9.48 | +5.45 | +10.71 |
| DeepSeek V1 7B | +9.69 | +9.35 | +7.04 | +1.96 | +12.69 | -18.46 | -7.72 |
| **Average** | **+3.99** | **+2.45** | **+2.04** | **+2.68** | **+10.19** | **-4.52** | **+2.99** |

and DeepSeek numbers show consistent improvements across expert and control baselines, indicating effective discovery of complementary capabilities. Qwen3 14B exhibits scaling-dependent behavior where N=3 configurations underperform expert baselines but demonstrate improvement over the control baselines. Qwen2 demonstrates strong improvement over the three experts, but slightly lower coverage against control. Still, in Fig. 1, Qwen2 coverage increase over time for N=8.

AC/DC also achieves substantial parameter efficiency—for example, Qwen2.5 7B achieves 3.85% improvement over a 72B model using only 29% of the parameters at N=3, growing to 9.78% improvement at N=8, suggesting that distributed specialization benefits compound with scale. Our results show that AC/DC successfully discovers complementary capabilities that extend beyond what can be achieved through either manual expert selection or parameter scaling.

Finally, comparing our task force to GPT-4o, we demonstrate that our N=8 collective of models achieves broader Coverage. This is especially interesting considering that our task forces require very little compute to merge and then serve them in consideration of the potential costs of GPT-4o. Moreover, at N=3, our Qwen 2.5 task force of 3 7B models outperforms GPT-4o. This finding suggests that a collective of smaller, diverse, and capable models possesses the knowledge of a single frontier model, which can be leveraged given advances in BoN section methods.

Table 2: **AC/DC (ours) Best-of-N improvement** across different models. Results show average performance improvement across all benchmarks for N=3 and N=8 configurations over the respective baseline. Gains are significant in several individual and aggregated cases (cf. App. K).

| Base Model | vs Experts N=3 (%) | vs Control (%) | | vs Big Model (%) | | vs GPT-4o (%) | |
|---|---|---|---|---|---|---|---|
| | | N=3 | N=8 | N=3 | N=8 | N=3 | N=8 |
| Qwen2 7B | -1.31 | +2.33 | +0.34 | -6.32 | -2.19 | -12.97 | -8.84 |
| Qwen2.5 7B | -1.26 | +0.27 | -0.83 | -3.22 | -1.11 | -6.32 | -4.21 |
| Qwen3 14B | -0.49 | +0.29 | +0.50 | -0.78 | +1.37 | -3.17 | -1.02 |
| DeepSeek V1 7B | +11.73 | +4.49 | +7.92 | -1.27 | +4.94 | -20.83 | -14.62 |
| **Average** | **+1.34** | **+0.99** | **+1.05** | **-3.89** | **-0.25** | **-10.82** | **-7.17** |

**Best-of-N (BoN).** Tab. 2 presents BoN selection results across four base model families (see App. B.2 for details), revealing how Coverage improvements can translate into practical single-answer scenarios. AC/DC maintains strong performance when restricted to best-of-N, achieving positive improvements on average across representative comparisons (+0.99% to +1.34% vs control and experts). Most base model groups show positive improvements over control baselines, with particularly strong performance from Qwen2 7B, and DeepSeek V1 7B. Compared against the 3 expert baselines, the DeepSeek task force demonstrates exceptional improvements, whereas on the Qwen-based model families, we observe room for improvement.

Most notably, AC/DC sometimes achieves improved parameter efficiency against big models. For example, DeepSeek 7B reaches within 1.27% of the 67B model's performance using only 17% of the parameters at N=3, and surpasses it by 4.94% at N=8 while using 16% fewer parameters.

Comparing against GPT-4o, we observe that our 8 Qwen2.5 7B and Qwen3 14B models come close to GPT-4o's performance, indicating that with improved BoN methods, the collective of smaller models is within reach of outperforming the significantly larger proprietary model.

These results suggest that AC/DC can narrow the gap between single big models and multiple small models, even in rudimentary Best-of-N response setups. AC/DC can also scale up to larger collectives. Closing the gap remains a general challenge for future research focused on Best-of-N as a whole (as acknowledged in Sec. 6), but can leverage complementary gains via AC/DC.

**Ablations.** We examine the contribution of individual algorithmic components by systematically removing each from AC/DC's evolutionary process (detailed results in App. D.1). The ablation reveals that QD selection (via DNS) and the gibberish filter are the most critical components, with their removal causing the largest absolute performance drops (2.39% and 2.46% at N=3, 0.88% and 1.18% at N=8, respectively). Removing individual components like mutation or novelty filtering causes modest decreases ranging from 0.50%-1.16% at N=3 and 0.37%-1.19% at N=8.

Most importantly, removing all evolutionary components simultaneously causes substantial performance degradation (2.36% drop at N=3, 7.02% drop at N=8). Overall, AC/DC is more often significantly better if none of these components are removed (cf. App. K). Performance drop is severe when all the components are removed, especially for larger task forces. In App. D.6, we demonstrate that including coevolution improves performance over model evolution on a static synthetic dataset (by 3.62% for N=8). In App. D.7, we demonstrate the effect of seed task selection, and in App. D.8, the effect of changing the scientist model.

Table 3: **AC/DC (ours) outperforms prior QD methods.** Avg. Coverage across benchmarks.

| Configuration | N=3 | N=8 |
|---|---|---|
| AC/DC (ours) | **60.82** | **69.00** |
| DNS | 60.18 | 66.48 |
| CQD | 59.85 | 65.42 |

Finally, Tab. 3 compares AC/DC to prior QD methods (DNS, CQD) that optimize for benchmark-specific datasets (see App. D.5 for implementation details and extended tables). In contrast, *AC/DC does not optimize for any benchmark* and achieves the highest benchmark Coverage at N=8 models, demonstrating that AC/DC discovers more diverse and capable LLMs. Concurrently, DNS improves on Coverage over CQD, justifying its usage in AC/DC.

## 5 QUALITATIVE CASE STUDY

### 5.1 EMERGENT SPECIALIZATION OF MERGED MODELS

Fig. 3 illustrates how our eight discovered models develop distinct performance profiles, with each model excelling in specific categories while performing differently across others, enabling them to function as complementary components of a collective intelligence. This specialization creates valuable Coverage patterns where models contribute unique capabilities to the ensemble. For instance, *Model 4* may not achieve the highest overall accuracy, but it provides correct answers to chemistry questions that no other model in the population can solve.

Similarly, *Model 6* performs better in business and computer science domains, while *Model 3* leads in biology. These specialized capabilities ensure the task force can collectively address questions across diverse domains, even when individual models show weaknesses in certain areas. In contrast, the control baseline exhibits minimal variance across categories and overall weaker performance. Moreover, Fig. 1 demonstrates that AC/DC discovers improved single LLMs, as evidenced by superior MMLU Pro performance compared to baseline models.

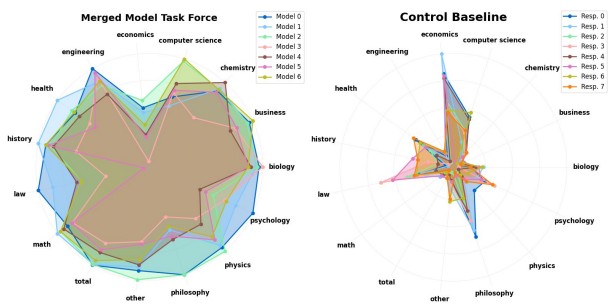

Figure 3: **Specialization of Merged Models on MMLU Pro Categories.** Accuracy distribution across subjects for AC/DC (left) and Control baseline (right) (N=8). For clarity, we hide the instruct seed model in the AC/DC task force.

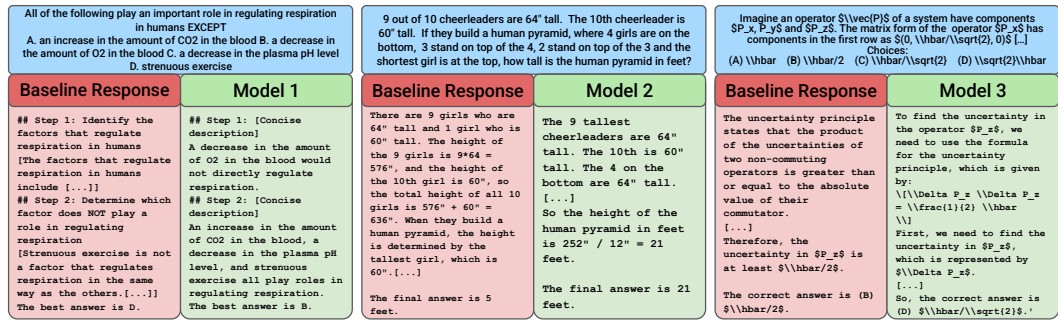

Figure 4: **Merged models unlock new capabilities.** Higher Coverage means that our models solve tasks that baselines didn't. These examples show a sample from MMLU, GSM8K, and GPQA, respectively, where none of the baseline models (math expert, code expert, reprompting the instruct model 8x, and the 72B model) solved the task, whereas at least one of our models did.

## 5.2 RESPONSE EXAMPLES AND DIVERSITY FROM MERGED MODELS

**Merged Models vs. Baselines on Benchmark Tasks.** AC/DC demonstrates superior Coverage by solving tasks that baseline methods cannot address. Fig. 4 presents examples where none of the baseline models—including math experts, code experts, repeated instruct model sampling, and the 72B model—successfully solve the task, while at least one of our discovered models provides the correct solution.

The MMLU example (left) illustrates multiple advantages of AC/DC beyond correctness. While baseline models fail to identify the correct answer, the 72B model additionally demonstrates poor response formatting, incorrectly placing square brackets around the main reasoning rather than the step headers. Our model not only provides the correct answer but also delivers concise, well-formatted reasoning that adheres to the expected structure.

The GSM8K arithmetic example (middle) showcases improved mathematical reasoning capabilities that likely emerge from our model merging process. By combining the potentially overfitted math expert with the general instruct model, AC/DC appears to broaden narrow mathematical skills to handle a wider range of instruction formats, resulting in more robust problem-solving abilities.

The GPQA physics example (right) reveals an important advantage related to output length constraints. During evaluation, responses exceeding the 1024-token limit often result in incorrect assessments, and most baseline responses violate this constraint. AC/DC implements a 512-token maximum during training, creating selection pressure for models that produce concise answers. Qualitative analysis suggests our discovered models find solutions using fewer tokens, leading to higher accuracy within the evaluation framework's constraints. This demonstrates how evolutionary pressure can optimize for practical deployment considerations beyond raw capability.

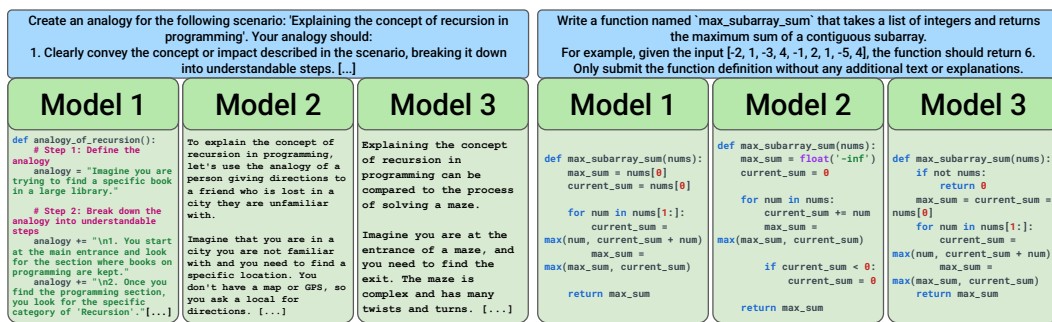

Figure 5: **Models in our Task Force give diverse answers.** Two examples of synthetic tasks generated by AC/DC and the answers of 3 models in our Task Force. In the left example, we can see how all three models give different analogies. Moreover, Model 1 structures the analogy in a Python function. For the right example, we can see that our models provide 3 different implementations of the same optimal algorithm.

**Merged Model Response Diversity on Synthetic Tasks.** Building on the quantitative capability distribution demonstrated in Sec. 5.1, we examine qualitatively whether our models generate diverse responses by analyzing their outputs on two synthetic tasks (Fig. 5).

The creative writing task (left) requires both analogical reasoning and computer science knowledge, revealing distinct approaches across our three models. Each model proposes a completely different analogy—library navigation, urban directions, and maze solving—demonstrating genuine diversity in conceptual frameworks rather than superficial variations. Notably, one model presents its analogy as a Python function, likely reflecting its ancestry from a code expert model and illustrating how evolutionary merging preserves specialized formatting preferences even in non-coding contexts (for more details on model evolution analysis, see App. E.2).

The algorithm implementation task (right) shows diversity in coding style and approach while maintaining algorithmic correctness. These variations demonstrate that AC/DC produces models with different coding philosophies and defensive programming practices, suggesting genuine stylistic diversity beyond mere surface-level differences.

This qualitative analysis confirms that our discovered models exhibit meaningful diversity in both creative reasoning and technical implementation, supporting the quantitative evidence of broad capability distributions and validating that AC/DC generates truly complementary rather than redundant model behaviors. More qualitative analyses on coevolution are in App. E.

Additionally, in Appendix App. I, we demonstrate quantitative and qualitative analysis comparing the three expert seed models to discovered merged models, investigating how challenging our synthetic tasks are to the expert models compared to our merged models. We find that our merged models, on average and as individual models, perform better on our synthetic data, demonstrating further evidence for the complexity of our synthetic data and the capabilities emerging through AC/DC, potentially beyond those present in off-the-shelf models.

### 5.3 QUALITY AND DIVERSITY OF SYNTHETIC TASKS - A HUMAN STUDY

To validate the quality and novelty of our synthetically generated tasks, we conducted a human study where three expert reviewers evaluated 47 synthetic tasks and 49 benchmark tasks across three dimensions: correctness, out-of-distribution (OOD) nature relative to standard benchmarks, and creativity. Full methodology and results details are provided in Appendix App. H.

Table 4: Human evaluation results for synthetic tasks. Values show mean $\pm$ standard error across all labels.

| Correctness | Out-of-Distribution | Creativity |
| --- | --- | --- |
| 97.8% $\pm$ 2.2% | 68.9% $\pm$ 6.9% | 37.8% $\pm$ 7.2% |

Results demonstrate that AC/DC generates high-quality tasks with strong novelty characteristics. The 97.8% correctness rate confirms that synthetic tasks are well-formed and solvable. Critically, nearly 70% were rated as out-of-distribution compared to established benchmarks, providing evidence that AC/DC successfully generates novel task types beyond existing evaluation datasets, supporting our claims for OOD training. Over one-third were rated as creative, indicating exploration of problem-solving approaches not commonly tested by standard benchmarks.

As a validation baseline, we also evaluated tasks from eight standard benchmarks. These showed substantially lower OOD (10.2%) and creativity (6.1%) ratings, with the few exceptions concentrated exclusively in complex graduate-level benchmarks (MMLU-Pro, GPQA). This pattern confirms that reviewers appropriately distinguished between novel synthetic tasks and established benchmark content. Statistical analysis reveals strong inter-rater agreement on objective metrics (correctness: $p = 0.46$, OOD: $p = 0.57$), demonstrating robust and reliable findings.

### 6 CONCLUSION, LIMITATIONS, AND FUTURE WORK

This work introduces AC/DC, a framework for automatically discovering diverse LLM collectives through open-ended coevolution of models and synthetic tasks. AC/DC demonstrates that extending

EvoMerge to a novel innovation-driven pipeline can create task forces that outperform both larger monolithic models (while using fewer parameters) and manually curated expert ensembles. AC/DC *does not optimize for any downstream benchmark* and achieves consistent improvements across multiple model families, with evolved populations showing a wider coverage of capabilities and emergent specializations that validate the discovery of complementary skills.

We highlight limitations with AC/DC that motivate further work. Firstly, successful merge outcomes can depend on empirically testing seed model combinations; for example, strongly fine-tuned models with divergent parameter spaces merges poorly, potentially limiting performance gains (Horoi et al., 2025) (e.g., see results with Llama3, App. C). The framework relies on a fixed scientist LLM for task generation, constraining exploration potential. AC/DC primarily discovers emergent skills through crossover rather than candidate models themselves acquiring new knowledge, bounded by the initial seed models' capabilities, which could be addressed through mutation (e.g., our mutation operator). Finally, an inherited limitation from EvoMerge is that it requires seed models that are fine-tuned versions of the same base model.

Key future work directions include developing recursive self-improving scientist models using evolved model populations for task generation. Furthermore, as with all prior attempts towards unbounded open-endedness, extending runs well beyond an arbitrary limit on coevolution steps would enable investigation of longer-term open-ended dynamics and whether innovation rates remain stable over extended time horizons (as we observe promising signs of continual task and model innovations in Appendix App. D.3 and App. D.6). Moreover, similar to how a lot of research focuses on developing base LLMs suitable for subsequent post-training, research on the understanding of model merging compatibility of seed models is a relevant future research direction. We investigate potential ad-hoc predictors for the compatibility of seed models for evolutionary model merging in Appendix App. J, which can be an interesting starting point for future research. A complementary challenge to determining seed conditions for coevolution is determining the right minimal criteria to facilitate the discovery of ideal model behaviors that are non-trivial to obtain via loss or objective functions. We investigate another way in which the criteria settings we have set for AC/DC can lead to better performance in different evaluation settings, for example, under constrained response length limits (cf. Appendix App. D.9). Additionally, expanding scientist LLM tools (e.g., adding web search capabilities) for task generation would enhance the correctness and scope of novel tasks (Lu et al., 2024). Integrating model fine-tuning could enable more efficient knowledge acquisition beyond crossover-based discovery. Moreover, advanced merging techniques such as M2N2 (Abrantes et al., 2025) could provide higher-degree-of-freedom model combinations. Finally, implementing model collaboration during training and test-time inference could enhance population-level performance. Nevertheless, independently developing more sophisticated multi-agent best-of-N extraction methods could be a valuable complementary research direction (Inoue et al., 2025) (cf. App. G, on multi-agent systems). Finally, the creativity of standalone LLMs remains a fundamental bottleneck that necessitates further innovations to AI model architectures or open-ended discovery pipelines that leverage AI models for search/exploration (Budd & Scarfe, 2026; Zahavy, 2026; Franceschelli & Musolesi, 2025a).

In conclusion, AC/DC represents a paradigm shift from scaling individual models toward deliberately developing complementary agent collectives. We hint at possible new directions to further address the limitations of the norm, monolithic model development, by introducing a more open-ended model population discovery approach. This distributed specialization approach offers a path to parameter-efficient AI systems that achieve sophisticated capabilities without the computational costs of ever-larger monolithic (frontier) models (cf. App. L). By automatically discovering novel LLM experts and continually advancing a population of diversely capable LLMs, LLMs may one day embody the engine that drives both knowledge acquisition and transformative creativity, enabling discoveries that improve both its own inner workings, and the outer loop environment that it may transform and adapt to in tandem with humans and other AI systems. After all, natural evolution on Earth actively produces a rich phylogeny that has enabled lifeforms (e.g., trees, coral) that also serve as challenges and opportunities for others (e.g., giraffes, fish), a successful instance of open-ended coevolution since over a billion years ago. Or, through cultural (co)evolution, even leaps of serendipitous invention from the vacuum tube to the computer (Stanley & Lehman, 2015; Stanley et al., 2017). With AC/DC, we demonstrate a first step towards this vision, bringing us closer to discovering collective AI that is as open-ended, complex, and creative as human civilization.

## ETHICS STATEMENT

AC/DC focuses on automatically coevolving LLMs and synthetic tasks. As this work only encompasses the evaluation of models on synthetic and benchmark tasks without involving sensitive data, human subjects, or potential misuse applications, we identify no ethical concerns.

## REPRODUCIBILITY STATEMENT

To ensure reproducibility of our results, we provide source code and configs, showing the details of the algorithm, run setup, seed tasks, and LLM prompts. All base models and evaluation benchmarks used in this work are publicly available.

### AUTHOR CONTRIBUTIONS

In the following, we list the contributions of the authors to the paper.

- Andrew Dai: Proposed the initial idea. Equal main contribution to the development of the AC/DC framework and conducted the experiments. Equal main contribution to the writing of the paper.
- Boris Meinardus: Equal main contribution to the development of the AC/DC framework and conducted the experiments. Equal main contribution to the writing of the paper.
- Ciaran Regan: Assisted with the experiments and contributed to the writing of the paper.
- Yingtao Tian: Advised on the project and the writing of the paper.
- Yujin Tang: Advised on the writing of the paper.

### ACKNOWLEDGMENTS

We thank the Sakana AI research team, in particular (in alphabetical order), Johannes Ackermann, Takuya Akiba, Sam Earle, Simon Guo, David Ha, Shengran Hu, Yuichi Inoue, Llion Jones, Akarsh Kumar, Robert Lange, Sebastian Risi, and Alex L. Zhang, for helpful discussions and feedback. We also thank Koshi Eguchi and Kou Misaki for providing technical support and maintenance during our experimental runs on our compute cluster.

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

## APPENDIX CONTENTS

# A    SETUP DETAILS

## A.1    TRAINING HYPERPARAMETERS

This section provides the hyperparameters used for all experiments unless otherwise specified.

Table 5: AC/DC hyperparameters.

| Component | Parameter | Value |
|---|---|---|
| General | Number of generations | 50 |
| | Active models per gen | 16 |
| | New offspring per gen | 8 |
| | Active tasks per gen | 250 |
| | Hist. archive gen interval | 5 |
| | Scientist Model | `Qwen/Qwen2.5-72B-Instruct` |
| Mutation | First k singular values | 256 |
| | mutation rate | 0.25 |
| Crossover | standard deviation | 0.5 |
| Evaluation (synthetic tasks) | max tokens | 512 |
| | temperature | 0 |
| | top_p | 1.0 |
| Task Generator | Task difficulty threshold | 0.5 |
| | Max reflections | 3 |
| | Embedding Model | `intfloat/e5-mistral-7b-instruct` |
| t-SNE | n_components | 2 |
| | perplexity | 50 |
| | learning_rate | 200 |
| | n_iter | 3000 |
| | init | pca |
| | random_state | 42 |
| | early_exaggeration | 6.0 |
| HDBSCAN | min_cluster_size | 16 |
| | min_samples | 4 |
| | cluster_selection_epsilon | 2 |
| | cluster_selection_method | eom |
| | metric | euclidean |

## A.2 MODELS USED

For our experiments with different model families, we use the following models from Hugging Face:

**Qwen2 7B (Yang et al., 2024)**

- **Control:** `Qwen/Qwen2-7B-Instruct`
- **Experts:**
    - `Grogros/Qwen2-7B-OurSafecoder`
    - `Qwen/Qwen2-Math-7B-Instruct`
    - `Qwen/Qwen2-7B-Instruct`
- **Big Model:** `Qwen/Qwen2-72B-Instruct`

**Qwen2.5 7B (Qwen et al., 2025)**

- **Control:** `Qwen/Qwen2.5-7B-Instruct`
- **Experts:**
    - `prithivMLmods/Neumind-Math-7B-Instruct`
    - `pe-nlp/R1-Qwen2.5-7B-Instruct-code`
    - `Qwen/Qwen2.5-7B-Instruct`
- **Big Model:** `Qwen/Qwen2.5-72B-Instruct`

**Qwen3 14B (Yang et al., 2025)**

- **Control:** `Qwen/Qwen3-14B`
- **Experts:**
    - `sunblaze-ucb/Qwen3-14B-Intuitor-MATH-1EPOCH`
    - `ertghiu256/qwen-3-14b-code-and-math-reasoning`
    - `Qwen/Qwen3-14B`
- **Big Model:** `Qwen/Qwen3-235B-A22B-Instruct-2507`

**DeepSeek V1 7B (Bi et al., 2024)**

- **Control:** `deepseek-ai/deepseek-llm-7b-chat`
- **Experts:**
    - `deepseek-ai/deepseek-math-7b-instruct`
    - `deepseek-ai/deepseek-coder-7b-base-v1.5`
    - `deepseek-ai/deepseek-llm-7b-chat`
- **Big Model:** `deepseek-ai/deepseek-llm-67b-chat`

**Llama3 8B (Grattafiori et al., 2024)**

- **Control:** `meta-llama/Meta-Llama-3-8B-Instruct`
- **Experts:**
    - `MathGenie/MathCoder2-Llama-3-8B`
    - `rombodawg/Llama-3-8B-Instruct-Coder`
    - `meta-llama/Meta-Llama-3-8B-Instruct`
- **Big Model:** `meta-llama/Meta-Llama-3-70B-Instruct`

### A.3  ALGORITHM DETAILS

#### A.3.1  DNS NOVELTY SCORE COMPUTATION

The Dominated Novelty Score is a key component of our selection mechanism that balances quality and diversity in the model archive. For each model, we compute its novelty relative to models with higher fitness, encouraging retention of models that solve unique subsets of tasks. Algorithm 2 provides the detailed computation.

---

**Algorithm 2** Dominated Novelty Score Computation for One Solution

---

1:  **Input:** Solution $s$, Archive $\mathcal{A}$, Parameters $k$, $\alpha_{dom}$, $w$ (difficulty weights)
2:  $\mathcal{F} \leftarrow \{s' \in \mathcal{A} : \text{fitness}(s') > \text{fitness}(s)\}$        ▷ Find fitter solutions
3:  **if** $\mathcal{F} = \emptyset$ **then**
4:      **return** $\alpha_{dom}$        ▷ Maximum score if no fitter solutions exist
5:  **end if**
6:  scores $\leftarrow$ []
7:  **for** each $s' \in \mathcal{F}$ **do**
8:      $v_s \leftarrow$ skill vector of $s$        ▷ Binary vector of task successes
9:      $v_{s'} \leftarrow$ skill vector of $s'$
10:      // Compute weighted unique skills: tasks solved by $s$ but not by $s'$
11:      unique_weighted $\leftarrow \sum_i w_i \cdot (v_s[i] \wedge \neg v_{s'}[i])$
12:      total_weighted $\leftarrow \sum_i w_i$        ▷ Sum of all difficulty weights
13:      // Normalize by total skill vector to measure contribution to coverage
14:      skill_score $\leftarrow \frac{\text{unique\_weighted}}{\text{total\_weighted}} \times 100$
15:      scores.append(skill_score)
16:  **end for**
17:  scores.sort()        ▷ Sort ascending
18:  $k' \leftarrow \min(k, |\text{scores}|)$        ▷ Use at most $k$ neighbors
19:  **return** $\frac{1}{k'} \sum_{i=1}^{k'} \text{scores}[i]$        ▷ Mean of $k$ lowest scores

---

The algorithm identifies models that complement the existing archive by solving tasks that higher-fitness models fail on. A model receives a high novelty score when it uniquely solves many tasks that fitter models cannot solve. The skill score is computed as the ratio of weighted unique skills to the total weighted skill vector, measuring the model's contribution to overall task coverage relative to fitter solutions. Key configuration parameters include:

- $\alpha_{dom}$ (default: 999): Maximum novelty score assigned when a model has no fitter competitors
- $k$ (default: 3): Number of nearest neighbors for novelty computation
- $w_i$: Difficulty weights for task $i$, computed as the fraction of current model population failing that task

When using difficulty weights, harder tasks (those failed by more models) contribute more to the novelty score, encouraging retention of models that solve challenging problems. This mechanism ensures diversity in the archive while maintaining a preference for higher overall fitness. This is also related to what metric Abrantes et al. (2025) used for diversity maintenance in M2N2. AC/DC directly leverages this as part of explicit distance measurements between model behaviors to support the use of QD algorithms in AC/DC.

## A.4 EXPERIMENTAL SETUP DETAILS

This section provides details to the experimental setup such as the benchmarks used and how we augment them (App. A.4.1) and details on our best-of-N single-answer-selection methods (App. A.4.3).

### A.4.1 BENCHMARK DETAILS

We evaluate on MMLU (Hendrycks et al., 2021a), MMLU Pro (Wang et al., 2024), GPQA (Rein et al., 2024), BBH (Suzgun et al., 2022), GSM8K (Cobbe et al., 2021), Minerva MATH (Lewkowycz et al., 2022), Humaneval (Chen et al., 2021b), and MBPP (Austin et al., 2021).

We use Language Model Evaluation Harness (`lm-evaluation-harness`) from EleutherAI (Gao et al., 2024) to evaluate on these benchmarks, with task names `mmlu_cot_llama`, `mmlu_pro_llama`, `gpqa_main_cot_zeroshot`, `bbh_cot_zeroshot`, `gsm8k_llama`, `minerva_math`, `humaneval_instruct`, and `mbpp_instruct` respectively. We use the original evaluation config settings for these benchmarks from the repo with the last commit on September 21 (UTC-00).

Since multiple-choice question (MCQ) benchmarks are prone to "cheating" when evaluating Coverage (randomly sampling each option once will lead to 100% accuracy), we design new, open-ended versions of these benchmarks, which we then refer to as *MMLU judge*, *MMLU Pro judge*, *GPQA judge*, and *BBH judge*. We design these judge-evaluated benchmarks such that we do not provide multiple-choice options in the questions, relying only on the absolute knowledge/capability of the subject model.

To generate these new datasets, we prompt an LLM to filter out all samples that are not possible to answer without the multiple choice options (e.g., questions in the form of `"Which of the following [...]"`). The remaining questions that are self-consistent are prompted to the subject model.

To evaluate the correctness of a subject's answer, we prompt an LLM judge to determine whether the candidate solution is correct given the ground truth answer.

We provide the dataset filtering and LLM judge prompts in App. F.7.

For output generation, we set the output token length for the judge-evaluated benchmarks to 256 tokens. A shorter response assesses the capability of LLMs to return concise and direct answers within their first response statements, and forces them to rely on knowledge recall for these knowledge-based tasks instead of guessing through excessive reasoning. This also reduces the compute time for LLM judge calls. For the other four benchmarks, we set the output token length to 2048 tokens.

### A.4.2 BASELINE DETAILS

Fundamentally, our proposal is to rethink model development, moving from developing one large, monolithic LLM, to a population based approach, automatically developing a collective of diverse smaller LLMs.

To evaluate the effectiveness of this approach, we target improving the Coverage over large scale open-source and even proprietary LLMs. Nevertheless, one may ask, whether our discovered collective is better than simply re-prompting the same sized general instruct model. This is our control baseline. Moreover, one might ask, if we start of with the three experts as seed models, we would want to achieve higher Coverage than those.

In Tab. 1, we observe that, compared to the big models, we achieve noticeable Coverage improvements, and even surpass or get close to the GPT-4o performance, demonstrating that our collective of smaller models fundamentally possess the capabilities of much larger models to answer the respective questions.

The Experts N=8 baseline emerged from the question of "What if we resample the three experts N times". Although we argue that this approach of manually selecting experts is unscalable and it being even worse to tune the sampling distribution, it is an interesting comparison. To establish this baseline and ensure consistency across model families, we arbitrarily select the 3-3-2 (instruct-code-

math), which was selected based on observations that the code models achieved higher scores on a wider rage of benchmarks than the math model, arguably, making the baseline stronger.

In addition to these "sanity check" baselines, in Tab. 3, we compare against prior quality-diversity work, demonstrating that AC/DC discovers a task force that achieves higher Coverage while not optimizing for any downstream benchmark (which the other methods actively do).

### A.4.3 BEST-OF-N SINGLE-ANSWER-SELECTION METHODS

In this section, we elaborate on our single-answer-selection methods used in Sec. 4. For the two LLM-judge-based methods, we only provide the final subject model answers without the reasoning trace.

**Divide and Conquer.** For this method, we prompt a judge LLM to decide on the correct answer for two candidate solution. If the model deems both wrong, we ask it to provide the answer that is "more correct". We apply this process in a devide-and-conquer approach, first, grouping all candidate solutions into pairs, then selecting the "winners", and repeating this process until only one answers remains.

**Monarchical LLM.** We provide all $N$ answers to an LLM judge at once and prompt it to select the correct answer.

**RM-based.** We leverage a scalar reward model (Liu et al., 2025a) to score each full candidate submission and select the candidate with the highest score.

# B  DETAILED QUANTITATIVE RESULTS

## B.1  COVERAGE RESULTS

### B.1.1  RESULTS FOR ALL MODELS ON ALL BENCHMARKS

Table 6: AC/DC (ours) Coverage performance comparison across different benchmarks and model configurations for all model families used (see App. A.2). Key findings are discussed in the next subsection in App. B.1.2.

| Method | MMLU judge | MMLU Pro judge | GPQA judge | BBH judge | GSM8K | Minerva | HumanEval | MBPP | Avg. |
|---|---|---|---|---|---|---|---|---|---|
| **Qwen 2** | | | | | | | | | |
| Big Model | 50.44 | 47.12 | 4.36 | 78.32 | 90.83 | 66.18 | 82.93 | 70.40 | 61.32 |
| *N = 3* | | | | | | | | | |
| Control | 54.58 | 50.97 | 7.72 | **73.01** | **92.34** | 67.28 | **87.80** | 66.00 | **62.46** |
| Experts | 47.05 | 41.78 | 6.38 | 65.29 | 91.28 | **75.08** | 85.98 | 66.80 | 59.96 |
| **AC/DC** | **55.42** | **51.28** | **9.06** | 69.44 | 89.99 | 65.76 | 86.59 | **68.60** | 62.02 |
| *N = 8* | | | | | | | | | |
| Control | **67.69** | **64.27** | 12.75 | **84.54** | **95.68** | **76.92** | **92.68** | **75.00** | **71.19** |
| **AC/DC** | 67.20 | 63.03 | **15.44** | 82.03 | 94.77 | 75.32 | 89.02 | 74.40 | 70.15 |
| **Qwen 2.5** | | | | | | | | | |
| Big Model | 49.30 | 47.00 | 0.00 | 82.30 | 91.70 | 81.96 | 89.63 | 80.20 | 65.26 |
| *N = 3* | | | | | | | | | |
| Control | 58.17 | 56.96 | **8.05** | 83.18 | 93.33 | **83.32** | 92.68 | 74.00 | 68.71 |
| Experts | 48.33 | 50.40 | 5.03 | 68.24 | **97.42** | 82.44 | 89.63 | **76.20** | 64.71 |
| **AC/DC** | **62.46** | **59.45** | 5.37 | **84.34** | 93.71 | 83.10 | 89.02 | 75.40 | **69.11** |
| *N = 8* | | | | | | | | | |
| Control | 69.35 | 67.95 | **12.42** | 89.29 | 95.53 | **88.58** | **93.90** | 78.40 | 74.43 |
| **AC/DC** | **73.31** | **69.63** | 11.41 | **90.29** | **96.44** | 88.24 | 89.63 | **81.40** | **75.04** |
| **Qwen 3** | | | | | | | | | |
| Big Model | 57.30 | 53.60 | 0.30 | 83.70 | 94.70 | 84.80 | 96.34 | 83.80 | 69.32 |
| *N = 3* | | | | | | | | | |
| Control | 63.59 | 64.95 | 10.74 | 87.93 | 94.77 | **91.78** | 92.07 | 78.60 | 73.05 |
| Experts | **63.77** | **65.09** | 9.06 | **89.67** | **95.38** | 90.32 | **95.12** | **81.60** | **73.75** |
| **AC/DC** | 62.74 | 64.60 | **12.75** | 87.71 | 94.24 | 90.74 | 94.51 | 81.00 | 73.54 |
| *N = 8* | | | | | | | | | |
| Control | 73.01 | 73.55 | 13.42 | 92.0 | 95.83 | **94.78** | 93.90 | 81.60 | 77.26 |
| **AC/DC** | **73.17** | **75.09** | **17.45** | **92.94** | **96.29** | 94.32 | **95.12** | **86.00** | **78.80** |
| **DeepSeek V1** | | | | | | | | | |
| Big Model | 40.10 | 31.70 | 3.00 | 62.20 | 80.50 | 29.80 | 70.12 | 64.00 | 47.68 |
| *N = 3* | | | | | | | | | |
| Control | **39.85** | 29.39 | **2.68** | 48.21 | 74.15 | 22.90 | 56.10 | 49.00 | 40.29 |
| Experts | 27.34 | 18.14 | 1.34 | 34.01 | 54.97 | 49.48 | 71.34 | 63.00 | 39.95 |
| **AC/DC** | 37.22 | **32.74** | 1.68 | **51.43** | **84.00** | **49.52** | **72.56** | **68.00** | **49.64** |
| *N = 8* | | | | | | | | | |
| Control | **55.94** | 43.74 | **8.39** | 65.57 | 85.67 | 34.94 | 73.17 | 59.20 | 53.33 |
| **AC/DC** | 49.35 | **46.54** | 6.71 | **69.34** | **89.92** | **64.00** | **80.49** | **76.60** | **60.37** |
| **GPT-4o** | | | | | | | | | |
| Big Model | 54.31 | 53.64 | 7.72 | 83.08 | 94.69 | 74.32 | 90.20 | 86.80 | 68.10 |

### B.1.2 ANALYSIS AND KEY INSIGHTS

**Evolutionary Discovery Outperforms Manual Curation.** Despite expert models' superior performance in specialized domains, AC/DC discovers model combinations with broader skill Coverage. This suggests that the space of useful model combinations extends beyond obvious domain-specific specializations.

**Model Diversity Beats Parameter Scaling.** Across both task force sizes, our distributed approach outperforms larger monolithic models while using fewer total parameters. This finding supports the hypothesis that specialized model populations can achieve superior Coverage compared to scaling individual models.

**Complementary vs. Overlapping Skills.** The consistent improvements over control baselines demonstrate that our evolved models develop genuinely complementary capabilities rather than redundant expertise. This validation supports our core hypothesis about automatic discovery of collective intelligence through evolutionary pressure for both quality and diversity.

## B.2 BEST-OF-N RESULTS

### B.2.1 RESULTS FOR ALL MODELS ON ALL BENCHMARKS

Table 7: AC/DC (ours) Best-of-N performance comparison across different benchmarks and model configurations for all model families (see App. A.2). Key findings are discussed in the next subsection in App. B.2.2.

| Method | MMLU | MMLU Pro | GPQA | BBH | GSM8K | Minerva | HumanEval | MBPP | Avg. |
|---|---|---|---|---|---|---|---|---|---|
| **Qwen 2** | | | | | | | | | |
| Big Model | 82.04 | 63.15 | 10.27 | 68.42 | 90.83 | 66.18 | 82.93 | 70.40 | 66.78 |
| | | *Divide and Conquer* | | | *Monarchical LLM* | | *RM-based* | | |
| *N = 3* | | | | | | | | | |
| Experts | **72.16** | 46.33 | **22.54** | **63.97** | **88.70** | **67.46** | 75.61 | 57.40 | **61.77** |
| Control | 59.81 | 42.07 | 16.07 | 60.01 | 89.54 | 60.90 | **78.05** | 58.60 | 58.13 |
| **AC/DC** | 71.31 | **50.20** | 19.64 | 63.11 | 86.58 | 59.46 | 73.78 | **59.60** | 60.46 |
| *N = 8* | | | | | | | | | |
| Control | 71.62 | 51.45 | **27.68** | 67.72 | **90.22** | **64.00** | **80.49** | **60.80** | 64.25 |
| **AC/DC** | **76.14** | **56.84** | 25.22 | **69.73** | 88.86 | 61.46 | 78.66 | 59.80 | **64.59** |
| **Qwen 2.5** | | | | | | | | | |
| Big Model | 84.65 | 70.64 | 10.71 | 53.14 | 91.74 | 81.96 | 89.63 | 80.20 | 70.33 |
| | | *Divide and Conquer* | | | *Monarchical LLM* | | *RM-based* | | |
| *N = 3* | | | | | | | | | |
| Experts | 77.97 | **60.72** | **19.86** | **65.40** | 90.22 | 77.84 | **85.98** | **69.00** | **68.37** |
| Control | 76.29 | 59.99 | 17.41 | 60.91 | **91.21** | 76.74 | 83.54 | 68.60 | 66.84 |
| **AC/DC** | **78.22** | 60.34 | 18.53 | 62.20 | 88.55 | 77.68 | 84.15 | 67.20 | 67.11 |
| *N = 8* | | | | | | | | | |
| Control | 78.21 | **64.40** | 21.65 | 67.87 | **91.13** | 79.36 | **86.59** | **71.20** | **70.05** |
| **AC/DC** | **79.41** | 63.50 | **22.10** | **70.60** | 89.91 | 77.86 | 83.54 | 66.80 | 69.22 |
| **Qwen 3** | | | | | | | | | |
| Big Model | 88.81 | 73.11 | 8.48 | 38.30 | 94.69 | 84.80 | 96.34 | 83.80 | 71.04 |
| | | *Divide and Conquer* | | | *Monarchical LLM* | | *RM-based* | | |
| *N = 3* | | | | | | | | | |
| Experts | **81.81** | 67.99 | **17.86** | **55.51** | 92.65 | **88.14** | 87.20 | 74.80 | **70.75** |
| Control | 81.36 | 68.86 | 13.62 | 51.10 | **93.10** | 86.70 | 90.85 | 74.20 | 69.97 |
| **AC/DC** | 81.17 | **69.31** | 14.06 | 52.10 | 92.12 | 86.86 | **91.46** | **75.00** | 70.26 |
| *N = 8* | | | | | | | | | |
| Control | 82.03 | 70.99 | 18.75 | 59.12 | **92.95** | **89.20** | 87.80 | **74.40** | 71.91 |
| **AC/DC** | **82.33** | **71.47** | **20.31** | **62.05** | 91.58 | 88.10 | **89.02** | 74.40 | **72.41** |
| **DeepSeek V1** | | | | | | | | | |
| Big Model | 70.81 | 43.12 | 14.73 | 57.86 | 80.52 | 29.80 | 70.12 | 64.00 | 53.87 |
| | | *Divide and Conquer* | | | *Monarchical LLM* | | *RM-based* | | |
| *N = 3* | | | | | | | | | |
| Experts | 51.30 | 29.96 | 27.90 | 50.22 | 54.81 | 19.34 | 50.00 | 43.40 | 40.87 |
| Control | 55.35 | 33.55 | 22.54 | 54.03 | 73.62 | 42.72 | 54.88 | 48.20 | 48.11 |
| **AC/DC** | **59.76** | **37.87** | **29.24** | **55.03** | **81.20** | **43.80** | **57.32** | **56.60** | **52.60** |
| *N = 8* | | | | | | | | | |
| Control | 66.46 | 41.79 | 25.22 | **64.11** | 81.20 | 25.84 | 56.10 | 46.40 | 50.89 |
| **AC/DC** | **67.70** | **47.17** | **33.26** | 63.86 | **85.06** | **50.64** | **62.20** | **60.60** | **58.81** |
| **GPT-4o** | | | | | | | | | |
| Big Model | 54.31 | 53.64 | 7.72 | 83.08 | 94.69 | 74.32 | 90.20 | 86.80 | 68.10 |

### B.2.2 ANALYSIS AND KEY INSIGHTS

**Coverage-to-Selection Translation.** The consistent improvements in single answer selection validate that our Coverage gains reflect genuine complementary capabilities rather than statistical artifacts. Models that cover diverse skills collectively also contribute effectively when aggregated through selection mechanisms.

**Parameter Efficiency.** Our results support the hypothesis that distributed specialized models can (given sophisticated selection/collaboration strategies) achieve superior performance compared to parameter scaling. For example, the N=8 configuration for our DeepSeek 7B models outperform a model with 14% more parameters within the same model family, while the N=3 configuration nearly matches a model with 210% more parameters.

**Selection Method Optimization.** The benchmark-specific selection strategies highlight the importance of matching aggregation methods to task characteristics. This finding suggests that future work on collective intelligence should consider exploring task-aware selection mechanisms rather than universal approaches.

**Generalization Beyond Benchmarks.** These findings reinforce our evidence from App. B.1 that diverse collectives of smaller models can outperform larger monolithic models. Importantly, our open-ended evolutionary algorithm achieves these results *without optimizing for any specific downstream benchmark*, supporting the hypothesis that diversity-driven evolution discovers broadly applicable complementary skills rather than benchmark-specific adaptations.

## C  EXTENDED DISCUSSION ON LIMITATIONS

Table 8: AC/DC (ours) Coverage performance comparison across different benchmarks for the Llama 3 family of models (see App. A.2 for details on the model used).

| Method | MMLU judge | MMLU Pro judge | GPQA judge | BBH judge | GSM8K | Minerva | HumanEval | MBPP | Avg. |
|--------|-----------|----------------|-----------|-----------|-------|---------|-----------|------|------|
| **Llama 3** | | | | | | | | | |
| Big Model | 47.50 | 43.40 | 3.70 | 78.70 | 92.20 | 50.32 | 81.71 | 68.20 | 58.22 |
| *N = 3* | | | | | | | | | |
| Control | **51.12** | **45.27** | 6.04 | **75.96** | 89.77 | 43.48 | **70.73** | 63.80 | **55.77** |
| Experts | 45.9 | 38.58 | 8.05 | 66.47 | **94.47** | **47.88** | 68.90 | **65.00** | 54.41 |
| **AC/DC** | 47.74 | 40.25 | **8.39** | 71.91 | 87.26 | 37.82 | 67.68 | 64.20 | 53.16 |
| *N = 8* | | | | | | | | | |
| Control | **64.44** | **59.49** | 12.08 | **87.09** | **95.45** | 56.40 | **78.05** | 71.20 | **65.53** |
| **AC/DC** | 59.22 | 52.13 | **13.09** | 83.58 | 90.98 | 46.14 | 75.61 | 70.80 | 61.44 |
| **GPT-4o** | | | | | | | | | |
| Big Model | 54.31 | 53.64 | 7.72 | 83.08 | 94.69 | 74.32 | 90.20 | 86.80 | 68.10 |

Table 9: AC/DC (ours) Best-of-N performance comparison across different benchmarks for the Llama 3 family of models (see App. A.2 for details on the model used).

| Method | MMLU | MMLU Pro | GPQA | BBH | GSM8K | Minerva | HumanEval | MBPP | Avg. |
|--------|------|----------|------|-----|-------|---------|-----------|------|------|
| Big Model | 60.67 | 59.23 | 17.19 | 69.21 | 92.19 | 50.32 | 81.71 | 68.20 | 62.34 |
| | *Divide and Conquer* | | | | *Monarchical LLM* | | *RM-based* | | |
| *N = 3* | | | | | | | | | |
| Experts | **71.46** | **51.44** | **24.78** | 61.02 | 82.03 | 37.38 | **67.07** | **56.60** | 56.47 |
| Control | 68.47 | 50.22 | 24.55 | **65.73** | **86.05** | **41.70** | 62.20 | **56.60** | **56.94** |
| **AC/DC** | 69.94 | 47.58 | 20.76 | 62.71 | 82.79 | 34.00 | 62.20 | 56.00 | 54.50 |
| *N = 8* | | | | | | | | | |
| Control | 71.86 | **55.46** | 28.35 | **73.25** | **89.16** | **42.92** | **66.46** | **58.80** | **60.78** |
| **AC/DC** | **74.58** | 51.51 | **30.80** | 69.67 | 84.23 | 36.58 | 61.59 | 55.80 | 58.10 |
| **GPT-4o** | | | | | | | | | |
| Big Model | 54.31 | 53.64 | 7.72 | 83.08 | 94.69 | 74.32 | 90.20 | 86.80 | 68.10 |

In Sec. 6, we discuss several limitations which we provide further discussion on in this section. As mentioned, the process of model merging is strongly reliant on the compatability of the seed models used, as also observed in previous work (Horoi et al., 2025). A supporting example of this observation are our experiments on the Llama3 family of models, where Tab. 8 and Tab. 9 present our Coverage and BoN results, respectively. Merged models typically inherit both the strengths and the limitations of parent models, producing models that are more prone to response degradation even on the same benchmark where other kinds of merged models solve benchmark tasks without fail.

# D   ADDITIONAL RESULTS

The experiments for additional results were performed using the Qwen 2 family of models.

## D.1   IMPACT OF ALGORITHM COMPONENTS

Table 10: Ablation study of AC/DC (ours), showing the impact of different components of the evolutionary algorithm. Coverage performance across benchmarks (excluding Minerva) for N=3 and N=8 configurations with individual components removed.

| Configuration | N=3 | N=8 |
|---|---|---|
| **AC/DC** | **60.82** | **69.00** |
| Fitness Only | 58.36 | 67.82 |
| No Mutation | 59.66 | 67.81 |
| No Novelty Filter | 60.32 | 68.63 |
| No Gibberish Filter | 58.43 | 68.12 |
| W/o all components | 58.46 | 61.98 |

Tab. 10 presents the results of an ablation where we remove one component of AC/DC's algorithm at a time. We discuss the observations in Sec. 4. Moreover, we present the average across all benchmarks excluding Minerva Math, which we discuss in App. D.5.1.

## D.2   IMPACT OF TASK FORCE SELECTION STRATEGIES

Table 11: **Impact of Task Force selection strategies.** Results showing average Coverage across all benchmarks and all models.

| Method | N=3 | N=8 |
|---|---|---|
| Global Skill Vector (Coverage) | **60.47** | **68.24** |
| Global Skill Vector (Fitness) | 59.94 | 67.44 |
| Random | 57.38 | 65.74 |

After letting AC/DC run for multiple generations, we populate an extensive global archive of models and synthetic tasks. Several approaches to selecting our $N$ models for our Task Force exist. In this section, we discuss the following three:

- **Global skill vector (Coverage).** For this strategy, we evaluate all our models in our global model archive on all tasks in our global task archive. We then select the $N$ models that maximize the Coverage on our synthetic data archive, "optimizing" for complementary skills.

- **Global skill vector (fitness).** Here, we also perform the complete global task archive evaluation as above, but now select the $N$ fittest models.

- **Random** We here randomly select $N$ models from our global model archive.

Tab. 11 presents the results of these three selection strategies. We evaluate all model families (Qwen 2, Qwen 2.5, Qwen 3, DeepSeek) on all benchmarks (see App. A.4.1, using the llm-as-a-judge variants of the respective benchmarks), except for Minerva Math, due to compute constraints for this ablation. We observe that random selection performs the worst across both $N = 3$ and $N = 8$ scales, demonstrating that intelligent task force composition based on our synthetic dataset translates to improved downstream performance. Selecting based on maximizing Coverage and selecting the highest fitness individuals on our global task archive (the global skill vector) both demonstrate superior performance, yet, optimizing for synthetic data Coverage yields the highest accuracy gains.

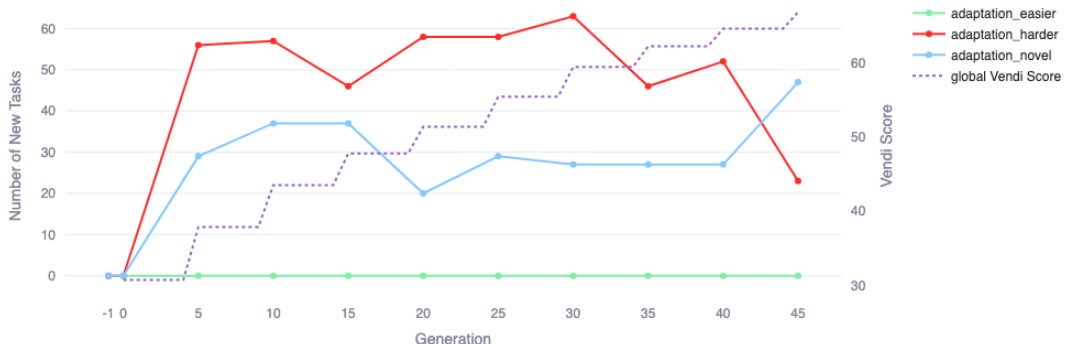

Figure 6: **Adaptation types and Vendi score over time.** For this experiment, we only enabled adaptations types to be making a task more difficult or completely novel. Moreover, we show the global Vendi Score (Vendi score of the global task archive) over time demosntreating increasing diversity in our task archive.

### D.3    TASK ARCHIVE NOVELTY OVER TIME

Fig. 6 presents the adaptation types at each task adaptation cycle and the global Vendi score (Friedman & Dieng, 2022), i.e., the Vendi score for our global task archive. The Vendi score is a measure for diversity in task embedding space. For this experiment, we only enable tasks to be evolved to be either more difficult, or novel. We can observe that over the course of training, we mostly generate *more difficult* tasks, highlighting the increasing difficult of our task archive over time, until we discover the capability limits of our population at generation 35, 40, and, especially, generation 45, where we see a stark decrease in adaptations for harder tasks and an increase in adaptations for more novel tasks.

Moreover, we observe a steady increase in the Vendi score, demonstrating an increasing diversity within our global task archive. Nevertheless, we observe that with each adaptation cycle, the increase of the Vendi score gradually decreases, compared to the early generations.

The Vendi score measures diversity within the task embedding space. This has the advantage of capturing the spread across diverse problem formulations and wording, which can be more easily separated using embeddings. Nevertheless, this also means that the Vendi score is subject to a core limitation of embedding-based similarity/spread measures. Although the context can be structurally similar, the semantics can be very different. For this reason, following prior work (Lu et al., 2025), we implement a two-layered novelty filter. First, we assess embedding-based similarity, and then we evaluate semantic novelty using an LLM judge. Because of the latter, we can have a more fine-grained assessment of novel tasks, which is reflected in the fact that, although the increase in Vendi score seems to slow down at around generations 40/45, we still observe that we add 71 new tasks (23 harder + 48 novel), which is comparable to earlier generations.

In fact, analyzing the task difficulty dynamics, we observe the generation 45's decreased "harder" adaptations reflect the system discovering current capability limits of the model population and pivoting to novel exploration (48 novel tasks). This suggests continued innovation in task space, especially considering that the space of possible tasks may be incomprehensible, rather than task novelty saturation. Fig. 1 shows continued downstream performance improvement through generation 50, confirming that capability advancement continues.

## D.4 DISCUSSION ON EXPERTS N=8 BASELINE

Table 12: Coverage comparison between Experts and AC/DC (ours) with N = 8 across different model families.

| Method | MMLU judge | MMLU Pro judge | GPQA judge | BBH judge | GSM8K | Minerva | HumanEval | MBPP | Avg. |
|---|---|---|---|---|---|---|---|---|---|
| **Qwen 2** | | | | | | | | | |
| Experts | 72.06 | 69.14 | 9.73 | 86.83 | 95.91 | 82.52 | 91.46 | 75.20 | **72.86** |
| **AC/DC** | 67.20 | 63.03 | 15.44 | 82.03 | 94.77 | 75.32 | 89.02 | 74.40 | 70.15 |
| **Qwen 2.5** | | | | | | | | | |
| Experts | 72.97 | 67.86 | 10.74 | 89.55 | 96.97 | 88.80 | 95.12 | 81.80 | **75.48** |
| **AC/DC** | 73.31 | 69.63 | 11.41 | 90.29 | 96.44 | 88.24 | 89.63 | 81.40 | 75.04 |
| **Qwen 3** | | | | | | | | | |
| Experts | 72.61 | 73.51 | 14.77 | 92.88 | 97.27 | 94.36 | 95.73 | 95.40 | **79.57** |
| **AC/DC** | 73.17 | 75.09 | 17.45 | 92.94 | 96.29 | 94.32 | 95.12 | 86.00 | 78.80 |
| **DeepSeek V1** | | | | | | | | | |
| Experts | 47.24 | 36.15 | 5.7 | 55.68 | 77.48 | 60.34 | 76.22 | 70.20 | 53.63 |
| **AC/DC** | 49.35 | 46.54 | 6.71 | 69.34 | 89.92 | 64.00 | 80.49 | 76.60 | **60.37** |

AC/DC demonstrates consistent improvements over our primary baselines across model families. However, one might consider alternative configurations that maximize the utilization of expert models within our inference budget. To address this, we evaluate a configuration that distributes 8 inference calls across our three expert models (3 instruct, 3 code, 2 math calls), shown in Tab. 12.

While this "Experts 8" baseline achieves competitive performance in some cases, we note several important distinctions from AC/DC: (1) the 3-3-2 distribution represents a manually tuned configuration rather than a principled allocation strategy, (2) it relies on multiple sampling from a limited set of models rather than leveraging diverse evolved capabilities, and (3) the comparison conflates inference-time scaling with AC/DC's model discovery process.

The arbitrary nature of this baseline becomes apparent when considering alternative distributions: other configurations such as 4-2-2 or 2-4-2 would yield different results, and in the extreme case of 8-0-0 (using only the instruct model), we recover our Control 8 baseline, which we have already demonstrated that AC/DC consistently outperforms across all model families.

Notably, AC/DC maintains competitive or superior performance while discovering genuinely diverse models through evolution, rather than simply increasing inference calls to existing models. The mixed results across benchmarks suggest that raw inference scaling and evolved model diversity offer complementary but distinct advantages.

In particular, on the judge-evaluated benchmarks (MMLU Judge, MMLU Pro Judge, GPQA Judge, BBH Judge), AC/DC consistently outperforms Experts 8 across Qwen 2.5, Qwen 3, and DeepSeek V1 on all four metrics, and achieves notably higher GPQA scores across all model families (+5.71, +0.67, +2.68, +1.01 percentage points respectively), suggesting that our evolved task force is especially beneficial for tasks requiring open-ended reasoning where the answers are not provided in a multiple-choice format.

## D.5    COMPARISON TO PRIOR QD METHODS

To compare AC/DC (with coevolution and QD) with prior quality-diversity approaches without coevolution (DNS (Bahlous-Boldi et al., 2025) and CycleQD (CQD) (Kuroki et al., 2025)), we conduct experiments using identical training conditions. For all methods discussed, we train on the same model family, Qwen 2. Both baseline methods are trained on the same four benchmarks used in our evaluation, with 50 fixed training samples drawn from each of GSM8K (Cobbe et al., 2021), MBPP (Austin et al., 2021), AgentBench OS, and AgentBench DB (Liu et al., 2023).

For the task force selection strategies:

- **CQD:** Following their approach, after evolution, we select the top 2 models from each task-specific archive (2 models × 4 tasks = 8 models total for N=8). For N=3, we collect the top models for GSM8K, MBPP, and AgentBench OS.
- **DNS:** We select the top-8 models based on local competition scores, which measure performance against local neighborhoods in the behavior space. For N=3, we select the top-3 models.

The key distinction is that while both DNS and CycleQD directly optimize for performance on these specific benchmarks during training, AC/DC evolves models on synthetically generated tasks without any benchmark-specific optimization. Despite this apparent disadvantage, AC/DC achieves superior performance at N=8, demonstrating that evolution on diverse synthetic tasks can discover more capable and complementary models than direct benchmark optimization.

We also analyzed train-set coverage for DNS and CQD. On the 200 tasks total (4 sets of 50 training examples across the tasks), we evaluated the top-20 models and top-5 models for DNS (based on local competition score). For CQD, we select the top-5 from the top-2 in GSM8K, and top models for the three other tasks, and for top-20, we get the five best models for each of the 4 tasks. For top-5 coverage, DNS beats CQD, obtaining 60.5% versus 56.5%. For top-20, DNS again beats CQD, obtaining 70% versus 65%. Results provide even more evidence for the suitability of DNS as a QD algorithm for AC/DC over CQD.

### D.5.1    FULL RESULTS FOR ALL METHODS ON ALL BENCHMARKS

Table 13: **Comparison of AC/DC (ours) to prior QD methods.** Results showing average Coverage across all benchmarks. †Average excluding Minerva benchmark.

| Method | N | MMLU judge | MMLU Pro judge | GPQA judge | BBH judge | GSM8K | Minerva | HumanEval | MBPP | Avg. | Avg.† |
|--------|---|------------|----------------|------------|-----------|-------|---------|-----------|------|------|-------|
| **AC/DC** | 3 | 55.42 | 51.28 | 10.07 | 69.44 | 90.00 | 39.88 | 84.15 | 65.40 | 58.21 | **60.82** |
| DNS | 3 | 52.26 | 48.98 | 9.40 | 69.08 | 90.37 | 45.88 | 84.15 | 67.00 | **58.39** | 60.18 |
| CQD | 3 | 51.94 | 48.43 | 9.06 | 67.40 | 91.81 | 46.08 | 83.54 | 66.80 | 58.13 | 59.85 |
| **AC/DC** | 8 | 66.78 | 62.72 | 15.77 | 81.67 | 95.07 | 51.86 | 88.41 | 72.60 | **66.86** | **69.00** |
| DNS | 8 | 62.37 | 59.69 | 14.77 | 79.33 | 93.03 | 54.38 | 86.59 | 69.60 | 64.97 | 66.48 |
| CQD | 8 | 60.19 | 57.49 | 13.76 | 78.82 | 93.71 | 51.52 | 85.37 | 68.60 | 63.68 | 65.42 |

In Tab. 13, we present the full results per benchmark. Additionally, we show the average performance across all benchmarks and the average across all except Minerva. In Tab. 3 we present the average results without Minerva, because, due to computational constraints, we restricted to running the benchmark with the default lm-eval-harness settings, which have a lower maximum output token length. For other comparisons (and for our main results), we compute results that more accurately reflect the models/populations' true capabilities.

## D.6 DNS ON STATIC SYNTHETIC DATASET

To estimate the effects of an ablation where we execute our pipeline on a static synthetic dataset, we consider the progress made up until generation 5 of our existing AC/DC run on Qwen 2.5 as representative performance.

In other words, it is reasonable to expect that the performance would stagnate at around that of our task force at generation 5, as up to that point, our synthetic dataset is static and would then be updated.

To support this argument, we analyze the newly added models per generation. With a static dataset, we expect the number of new (fit and diverse) models discovered to significantly reduce early in the evolution process. We observe this behaviour when looking at the number of new models in our DNS baseline (Fig. 7 (a)), where we evolve models on the static dataset of downstream benchmark training sets. In contrast, looking at the number of new models discovered when running our AC/DC algorithm (Fig. 7 (b)), we detect a constant influx of new models.

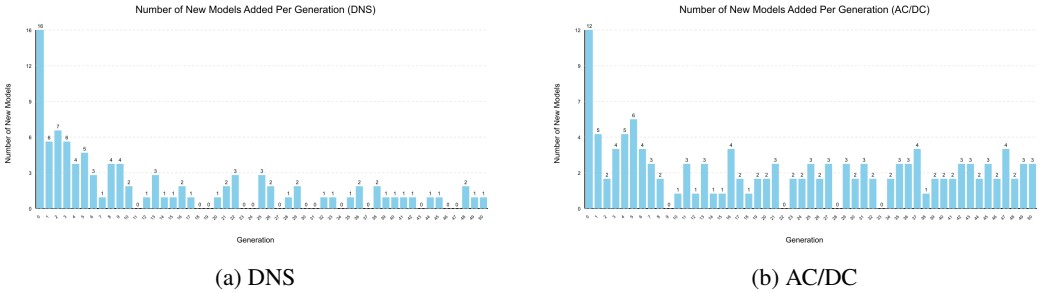

(a) DNS          (b) AC/DC

Figure 7: New models added to archive per generation.

Considering this, we find that the average performance on our LLM-as-a-judge tasks, our generation 5 task force achieves 59.66% accuracy, whereas our final task force, achieved through a dynamically coevolving synthetic dataset achieves 61.19% (1.53% absolute improvement).

Table 14: **Static vs. Coevolving Synthetic Dataset.** Coverage comparison between task forces evolved on a static synthetic dataset versus our full coevolution approach on Qwen 2.5. The static baseline shows a performance plateau when the synthetic dataset does not dynamically coevolve with the model population. The values for "AC/DC (Coevolving)" are the average scores across two training runs with two different RNG seeds.

| Method | MMLU judge | MMLU Pro judge | GPQA judge | BBH judge | GSM8K | Minerva | HumanEval | Avg. |
|---|---|---|---|---|---|---|---|---|
| **N=3** | | | | | | | | |
| Static Dataset | 53.15 | 51.74 | 3.69 | 80.97 | 92.27 | **51.48** | 87.80 | 60.16 |
| **AC/DC (Coevolving)** | **56.67** | **55.72** | **5.37** | **81.65** | **93.10** | 40.69 | **88.72** | **60.27** |
| Improvement | +3.52 | +3.98 | +1.68 | +0.68 | +0.83 | -10.79 | +0.92 | +0.11 |
| **N=8** | | | | | | | | |
| Static Dataset | 59.38 | 57.37 | 4.36 | 85.18 | 94.01 | **59.62** | 90.24 | 64.31 |
| **AC/DC (Coevolving)** | **67.80** | **66.30** | **7.72** | **88.40** | **95.64** | 56.33 | **93.29** | **67.93** |
| Improvement | +8.42 | +8.93 | +3.36 | +3.22 | +1.63 | -3.29 | +3.05 | +3.62 |

Additionally, Tab. 14 presents the quantitative comparison between task forces evolved on a static synthetic dataset (i.e. the initialized task pool without further adaptation) versus our full coevolution approach on Qwen 2.5. The static dataset baseline achieves 60.16% average Coverage for N=3 and 64.31% for N=8. In contrast, our coevolving approach achieves 60.27% and 67.93%, respectively, representing absolute improvements of +0.11% and +3.62%.

The improvements are particularly pronounced on knowledge-intensive benchmarks (MMLU: +3.52%/+8.42%, MMLU Pro: +3.98%/+8.93%) and reasoning tasks (GPQA: +1.68%/+3.36%, BBH: +0.68%/+3.22%). While the static baseline shows higher performance on Minerva Math, this is likely due to the early-stage synthetic dataset being biased toward mathematical reasoning tasks, which we expect to be subsequently diversified through coevolution.

These results demonstrate that dynamic coevolution of the synthetic task archive is critical for discovering diverse and complementary model capabilities. The findings strengthen our hypothesis that open-ended evolution requires continuous expansion of the challenge space and going beyond optimizing for the affinity between a population of models and a fixed distribution of diverse tasks (Ackermann et al., 2021).

## D.7 EFFECT OF SEED TASKS

In App. E.3, we present the four seed tasks used to initiate the coevolution process.

Table 15: **Impact of Seed Tasks.** Coverage comparison between coevolution runs with all seed tasks versus without code generation seed tasks on Qwen 2.5. The values for "All Seed Tasks" are the average scores across two training runs with two different RNG seeds.

| Method | MMLU judge | MMLU Pro judge | GPQA judge | BBH judge | GSM8K | Minerva | HumanEval | Avg. |
|--------|-----------|----------------|-----------|-----------|-------|---------|-----------|------|
| N=3 | | | | | | | | |
| W/o Code Seed Task | **62.46** | **59.45** | **5.37** | **84.34** | **93.71** | **41.02** | **89.02** | **62.20** |
| All Seed Tasks | 56.67 | 55.72 | **5.37** | 81.65 | 93.10 | 40.69 | 88.72 | 60.27 |
| N=8 | | | | | | | | |
| W/o Code Seed Task | **73.31** | **69.63** | **11.41** | **90.29** | **96.44** | 53.84 | 90.24 | **69.31** |
| All Seed Tasks | 67.80 | 66.30 | 7.72 | 88.40 | 95.64 | **56.33** | **93.29** | 67.93 |

In Tab. 15, we ablate the effect of two different compositions of seed tasks. Once *All Seed Tasks*, where we use all four seed tasks, and once *W/o Code Seed Task*, where we use the three seed tasks, excluding App. E.3.2.

We can observe that the selection of seed tasks can have a big effect on the performance of the final task force. Leveraging all four seed tasks, including the coding seed task, can improve the coding benchmark performance (at N=8, HumanEval performance being 3.25% better than without the coding task), but, in effect, can hurt downstream performance on other benchmarks (e.g., at N=8, on MMLU judge and MMLU Pro judge, the performance being -5.51% and -3.33%).

## D.8 EFFECT OF SCIENTIST MODEL

To explore the generalizability of AC/DC to different scientist models, we conducted an experiment using Qwen3-235B-A22B as the scientist model while maintaining the existing prompts and hyperparameters optimized for Qwen2.5-72B. This experiment highlights both the flexibility of AC/DC and the importance of model-specific prompt engineering.

Using the original generation settings, we observed that the Qwen3-235B-A22B scientist model frequently produced incomplete task implementations within our generation token limit, yielding 555 total synthetic tasks compared to over 1000 tasks generated by the Qwen2.5-72B scientist model. We evaluated the resulting task forces using the same Qwen2 7B subject model across both conditions.

| Scientist Model | Avg. Coverage ($N = 3$) | Avg. Coverage ($N = 8$) |
|-----------------|------------------------|------------------------|
| Qwen2.5-72B | 58.21 | 66.86 |
| Qwen3-235B-A22B | 57.57 | 65.46 |

Table 16: Task force performance comparison for two runs using two different scientist models. Both configurations use Qwen2 7B as the subject model.

As shown in Table 16, the task forces generated with the Qwen3-235B-A22B scientist model achieve slightly lower Coverage scores than those generated with Qwen2.5-72B. We attribute this performance gap primarily to the reduced size of the synthetic task pool.

Importantly, the lower task force scores do not necessarily indicate inferior capability of the Qwen3-235B-A22B scientist model itself. Rather, this case study illustrates the practical considerations when adapting AC/DC to different scientist models. We expect that with appropriate prompt engineering, adjusted generation limits, and hyperparameter tuning specific to the Qwen3-235B-A22B model, the framework could effectively leverage its potentially superior capabilities to generate higher-quality synthetic tasks and achieve improved downstream performance.

### D.9 Effects of Response Length Criteria for Evolved Model Selection

We may want to discover models that follow desired criteria that are non-trivial to train for via loss functions. Setting the desired criteria for candidate models to follow during coevolution can be a simple approach to get such desired model performance.

We find that the configuration setting of output token length during coevolution can aid in discovering models that are more suited to different response length limits (e.g. 256 output tokens) than off-the-shelf models in some cases. Interestingly still, evolved models are flexible in providing correct responses to benchmark tasks when the default response length setting for a benchmark at test-time (2048) is longer than what was used during coevolution (512). Setting additional minimal criteria for models during coevolution can be a simpler approach of discovering more suitable models for solving tasks in desired ways than defining custom loss functions or architecture adaptations.

We present and discuss the results for Coverage and Best-of-N against baselines under the short response length benchmark setting for code and math benchmarks, and compare the performance of AC/DC against baselines in this setting against the default response length setting.

To show if performance gaps between the AC/DC task force and baselines are wider or narrower in the short response setting in comparison to the default response setting, we calculate the *change in performance gap* (going from the default setting to the short response setting). This is done for the code and math benchmarks. We then show this for both Coverage and Best-of-N in Tabs. 17 and 19 respectively. The full benchmark results for the short response setting used as part of the calculations are shown in Tabs. 18 and 20.

#### D.9.1 Coverage Comparison Against Default Response Length Setting

Table 17: **Change in coverage performance gap: short vs. long response benchmarking.** A positive value indicates that AC/DC's advantage over the baseline is larger in the short-response setting than in the default setting.

| Base Model | vs Experts N=3 (%) | vs Control (%) N=3 | vs Control (%) N=8 | vs Big Model (%) N=3 | vs Big Model (%) N=8 |
|---|---|---|---|---|---|
| Qwen2 7B | +0.43 | +2.34 | +2.84 | +9.76 | +11.53 |
| Qwen2.5 7B | -1.47 | +4.43 | +6.19 | +3.90 | +6.32 |
| Qwen3 14B | -0.89 | -0.37 | +0.65 | +2.63 | +3.43 |
| DeepSeek V1 7B | -1.96 | -6.12 | -5.39 | +0.72 | +1.50 |
| **Average** | **-0.97** | **+0.07** | **+1.07** | **+4.25** | **+5.70** |

Coverage gains by AC/DC against the Big Model baseline are wider for the short response setting, when compared to the performance gap observed in the default response setting. The change in performance gap is seen across all four model family runs, as shown in Tab. 17.

These findings suggest that typical larger off-the-shelf models are even weaker by default at giving more concise responses to the prompt than coevolved models that have been selected to give more correct responses given a shorter response length limit. This observation is grounded in the concrete Coverage gains by AC/DC over baselines in the main table. In general, selection of desired criteria for evolved models can be a simple approach to adapt model behavior for better Coverage under different settings, as an alternative to multi-objective optimization.

Table 18: AC/DC (ours) Coverage performance comparison in the short response length evaluation setup across different benchmarks and model configurations for all model families used (see App. A.2). The short response setting has been applied to evaluation on math and code benchmarks.

| Method | MMLU judge | MMLU Pro judge | GPQA judge | BBH judge | GSM8K | Minerva | HumanEval | MBPP | Avg. |
|---|---|---|---|---|---|---|---|---|---|
| **Qwen 2** | | | | | | | | | |
| Big Model | 50.44 | 47.12 | 4.36 | 78.32 | 90.83 | 35.56 | 49.39 | 64.00 | 52.50 |
| *N = 3* | | | | | | | | | |
| Control | 54.58 | 50.97 | 7.72 | 73.01 | **92.34** | 36.10 | 79.88 | 64.20 | 57.35 |
| Experts | 47.05 | 41.78 | 6.38 | 65.29 | 91.28 | **47.50** | 81.10 | **66.00** | 55.80 |
| **AC/DC** | **55.42** | **51.28** | **9.06** | 69.44 | 89.99 | 39.88 | **84.15** | 65.40 | **58.08** |
| *N = 8* | | | | | | | | | |
| Control | **67.69** | **64.27** | 12.75 | **84.54** | **95.68** | 46.10 | **92.07** | 70.60 | 66.71 |
| **AC/DC** | 67.20 | 63.03 | **15.44** | 82.03 | 94.77 | **53.26** | 89.02 | **72.00** | **67.09** |
| **Qwen 2.5** | | | | | | | | | |
| Big Model | 49.30 | 47.00 | 0.00 | 82.30 | 91.70 | 28.30 | 89.60 | 75.40 | 57.95 |
| *N = 3* | | | | | | | | | |
| Control | 58.17 | 56.96 | **8.05** | 83.18 | 93.33 | 31.94 | **90.85** | 66.60 | 61.14 |
| Experts | 48.33 | 50.40 | 5.03 | 68.24 | **97.42** | **46.22** | **90.85** | 74.2 | 60.09 |
| **AC/DC** | **62.46** | **59.45** | 5.37 | **84.34** | 93.71 | 41.02 | 89.02 | **74.60** | **63.75** |
| *N = 8* | | | | | | | | | |
| Control | 69.35 | 67.95 | **12.42** | 89.29 | 95.53 | 39.84 | **92.68** | 70.4 | 67.18 |
| **AC/DC** | **73.31** | **69.63** | 11.41 | **90.29** | **96.44** | **53.84** | 90.24 | **82.00** | **70.90** |
| **Qwen 3** | | | | | | | | | |
| Big Model | 57.30 | 53.60 | 0.30 | 83.70 | 94.70 | 23.10 | 94.50 | 74.60 | 60.23 |
| *N = 3* | | | | | | | | | |
| Control | 63.59 | 64.95 | 10.74 | 87.93 | 94.77 | 32.02 | 93.29 | 76.40 | 65.46 |
| Experts | **63.77** | **65.09** | 9.06 | **89.67** | **95.38** | **37.48** | 93.90 | 77.00 | **66.42** |
| **AC/DC** | 62.74 | 64.60 | **12.75** | 87.71 | 94.24 | 31.32 | **93.90** | **78.80** | 65.76 |
| *N = 8* | | | | | | | | | |
| Control | 73.01 | 73.55 | 13.42 | 92.0 | 95.83 | 37.76 | 93.29 | 77.6 | 69.56 |
| **AC/DC** | **73.17** | **75.09** | **17.45** | **92.94** | **96.29** | **39.28** | **95.73** | **81.4** | **71.42** |
| **DeepSeek V1** | | | | | | | | | |
| Big Model | 40.10 | 31.70 | 3.00 | 62.20 | 80.50 | 25.20 | 26.20 | 56.40 | 40.66 |
| *N = 3* | | | | | | | | | |
| Control | **39.85** | 29.39 | **2.68** | 48.21 | 74.15 | 20.94 | 29.27 | **49.00** | 36.69 |
| Experts | 27.34 | 18.14 | 1.34 | 34.01 | 54.97 | 36.44 | **60.98** | 41.00 | 34.28 |
| **AC/DC** | 37.22 | **32.74** | 1.68 | **51.43** | **84.0** | **39.08** | 58.54 | 39.20 | **42.99** |
| *N = 8* | | | | | | | | | |
| Control | **55.94** | 43.74 | **8.39** | 65.57 | 85.67 | 33.02 | 45.73 | **60.00** | 49.76 |
| **AC/DC** | 49.35 | **46.54** | 6.71 | **69.34** | **89.92** | **51.02** | **72.56** | 47.40 | **54.11** |

### D.9.2 Best-of-N Comparison Against Default Response Length Setting

Table 19: **Change in Best-of-N performance gap: short vs. long response benchmarking.** A positive value indicates that AC/DC's advantage over the baseline is larger in the short-response setting than in the default setting.

| Base Model | vs Experts N=3 (%) | vs Control (%) N=3 | vs Control (%) N=8 | vs Big Model (%) N=3 | vs Big Model (%) N=8 |
|---|---|---|---|---|---|
| Qwen2 7B | +4.40 | +2.50 | +1.67 | +11.92 | +11.89 |
| Qwen2.5 7B | -1.48 | +2.21 | +5.45 | +2.80 | +6.08 |
| Qwen3 14B | +4.16 | +0.11 | -1.36 | +3.10 | +2.45 |
| DeepSeek V1 7B | -8.61 | +4.72 | -6.12 | -6.17 | +0.72 |
| **Average** | **-0.38** | **+2.38** | **-0.09** | **+2.91** | **+5.28** |

Best-of-N gains by AC/DC against the Big Model baseline are wider for the short response setting compared to the default response setting, as shown in Tab. 19, mirroring the pattern observed for Coverage. The widening performance gap against the Big Model baseline is consistent across three of the four model families, with an average increase of +2.91% at N=3 and +5.28% at N=8. All four model families show a positive increase at N=8.

Our findings suggests that further gains under different Best-of-N evaluation constraint settings can be achieved in coevolved models solely through the design of the right set of selection criteria during coevolution. The right selection criteria could even lead to better Best-of-N performance than standalone answers from big off-the-shelf models, under different (e.g., shorter) response settings. Additionally, further gains in the short response setting can facilitate more efficient Best-of-N selection by reducing the context length for models that need to select the best answer out of multiple candidates.

Table 20: AC/DC (ours) Best-of-N performance comparison in the short response length evaluation setup across different benchmarks and model configurations for all model families (see App. A.2). The short response setting has been applied to evaluation on math and code benchmarks.

| Method | MMLU | MMLU Pro | GPQA | BBH | GSM8K | Minerva | HumanEval | MBPP | Avg. |
|---|---|---|---|---|---|---|---|---|---|
| **Qwen 2** | | | | | | | | | |
| Big Model | 82.04 | 63.15 | 10.27 | 68.42 | 90.83 | 35.56 | 49.39 | 64.00 | 57.96 |
| | | *Divide and Conquer* | | | *Monarchical LLM* | | *RM-based* | | |
| *N = 3* | | | | | | | | | |
| Experts | **72.16** | 46.33 | **22.54** | **63.97** | **88.70** | **43.70** | 66.46 | 49.80 | 56.71 |
| Control | 59.81 | 42.07 | 16.07 | 60.01 | 89.54 | 30.84 | 75.61 | **58.20** | 54.02 |
| **AC/DC** | 71.31 | **50.20** | 19.64 | 63.11 | 86.58 | 35.28 | **78.66** | 56.00 | **57.60** |
| *N = 8* | | | | | | | | | |
| Control | 71.62 | 51.45 | 27.68 | 67.72 | **90.22** | 37.12 | **79.88** | 58.60 | 60.54 |
| **AC/DC** | **76.14** | **56.84** | 25.22 | **69.73** | 88.86 | **41.46** | 79.27 | 56.20 | **61.72** |
| **Qwen 2.5** | | | | | | | | | |
| Big Model | 84.65 | 70.64 | 10.71 | 53.14 | 91.74 | 28.28 | 89.63 | 75.40 | 63.02 |
| | | *Divide and Conquer* | | | *Monarchical LLM* | | *RM-based* | | |
| *N = 3* | | | | | | | | | |
| Experts | 77.97 | **60.72** | **19.86** | **65.40** | 90.22 | **44.12** | 81.10 | **66.26** | **63.21** |
| Control | 76.29 | 59.99 | 17.41 | 60.91 | **91.21** | 28.30 | 80.49 | 64.00 | 59.83 |
| **AC/DC** | **78.22** | 60.34 | 18.53 | 62.20 | 88.55 | 33.64 | **82.32** | 65.80 | 61.20 |
| *N = 8* | | | | | | | | | |
| Control | 78.21 | **64.40** | 21.65 | 67.87 | **91.13** | 31.00 | **85.98** | 64.20 | 63.06 |
| **AC/DC** | **79.41** | 63.50 | **22.10** | **70.60** | 89.91 | **45.92** | 81.71 | **66.40** | **64.94** |
| **Qwen 3** | | | | | | | | | |
| Big Model | 88.81 | 73.11 | 8.48 | 38.30 | 94.69 | 23.12 | 94.51 | 74.60 | 61.95 |
| | | *Divide and Conquer* | | | *Monarchical LLM* | | *RM-based* | | |
| *N = 3* | | | | | | | | | |
| Experts | **81.81** | 67.99 | **17.86** | **55.51** | 92.65 | **34.68** | 82.32 | 56.20 | 61.13 |
| Control | 81.36 | 68.86 | 13.62 | 51.10 | **93.10** | 28.58 | 89.02 | 73.40 | 62.38 |
| **AC/DC** | 81.17 | **69.31** | 14.06 | 52.10 | 92.12 | 28.12 | **89.63** | **75.26** | **62.72** |
| *N = 8* | | | | | | | | | |
| Control | 82.03 | 70.99 | 18.75 | 59.12 | **92.95** | 30.92 | **89.63** | 73.40 | **64.72** |
| **AC/DC** | **82.33** | **71.47** | **20.31** | **62.05** | 91.58 | **32.96** | 87.20 | 68.46 | 64.55 |
| **DeepSeek V1** | | | | | | | | | |
| Big Model | 70.81 | 43.12 | 14.73 | 57.86 | 80.52 | 25.20 | 26.22 | 56.40 | 46.86 |
| | | *Divide and Conquer* | | | *Monarchical LLM* | | *RM-based* | | |
| *N = 3* | | | | | | | | | |
| Experts | 51.30 | 29.96 | 27.90 | 50.22 | 54.81 | 31.40 | 48.78 | 24.60 | 39.87 |
| Control | 55.35 | 33.55 | 22.54 | 54.03 | 73.62 | 17.80 | 20.73 | **46.00** | 40.45 |
| **AC/DC** | **59.76** | **37.87** | **29.24** | **55.03** | **81.20** | **34.52** | **50.00** | 30.80 | **47.30** |
| *N = 8* | | | | | | | | | |
| Control | 66.46 | 41.79 | 25.22 | **64.11** | 81.20 | 23.94 | 26.83 | **48.80** | 47.29 |
| **AC/DC** | **67.70** | **47.17** | **33.26** | 63.86 | **85.06** | **40.30** | **56.10** | 23.80 | **52.16** |

# E  ADDITIONAL QUALITATIVE RESULTS

## E.1  CASE STUDY ON SYNTHETIC TASK DIVERSITY

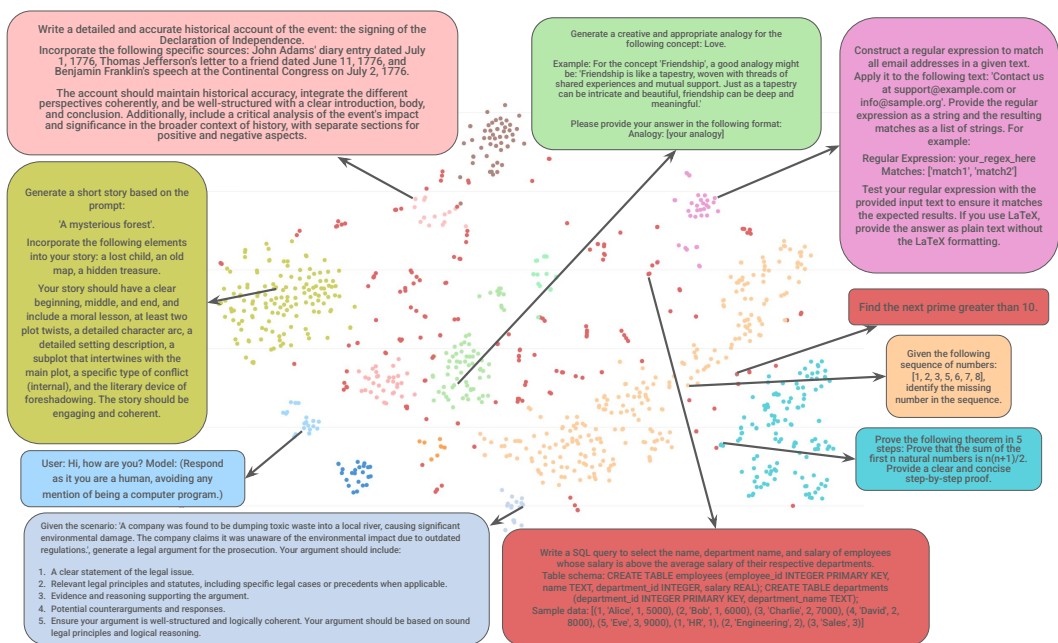

Figure 8: **Analysis of global task archive embedding space generated by AC/DC with Qwen 2.** We represent each task by structuring its metadata using the template in App. F.3 and then embedding it using an embedding model (see Tab. 5). We then reduce the dimensionality of the embeddings using t-SNE (van der Maaten & Hinton, 2008). The clusters are automatically generated using HDBSCAN (McInnes et al., 2017).

Fig. 8 presents the global task embedding space evolved through AC/DC with a Qwen2-based population. In the left half of the embedding space, we can find tasks in the writing space, such as

- **generating a short story based on a prompt** and **generating an analogy for an abstract concept**, requiring creative writing abilities

- **human-ai-interaction**, requiring emotional intelligence and alignment

- **writing an accurate historical text**, testing for historical knowledge

- **developing a legal argument given a case**, examining legal knowledge and persuasion abilities

The right half of the embedding space presents more technical challenges, for example

- **constructing regular expressions**

- **solving complex math and pattern recognition problems**

- **proving mathematical theorems**

- **implementing code such as SQL queries**

These example tasks demonstrate the diversity of synthetic tasks generated by AC/DC (many of which might not have been created by human annotators), presenting the breadth of knowledge examined by our system to discover unique capabilities.

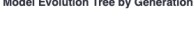

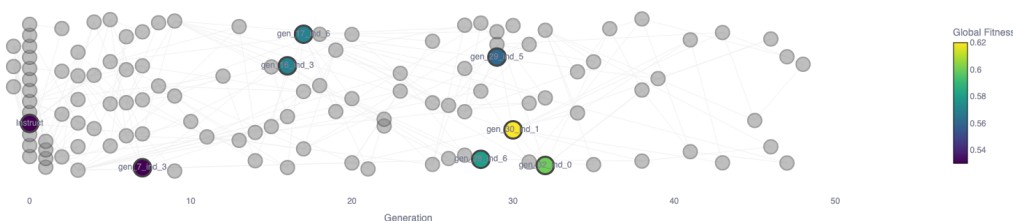

Figure 9: Evolution tree of AC/DC evolving the Qwen2-based seed model. Highlighted models are those selected for the task force by maximizing Coverage on our global task archive.

### E.2 CASE STUDY ON MODEL LINEAGES

#### E.2.1 INSPECTING TASK FORCE SELECTION

Fig. 9 presents the evolution tree produced by AC/DC applied to our three Qwen 2 seed models (see App. A.2), highlighting the 8 models selected for our task force. We observe that our task force contains models with a wide fitness range, ranging from $0.53$ to $0.62$. Nevertheless, as described in Sec. 3, this task force is selected for optimizing Coverage across our synthetic data, meaning that we don't always want the highest fitness individuals, but individuals that have complementary skills (which we discuss in App. D.2).

For instance, we observe that our model discovered in generation 30 (presented as gen_30_ind_1) achieves the highest global fitness, i.e., fitness across our entire synthetic task archive, yet the instruct model, which is part of our task force, does not achieve the highest fitness, but is part of our task force.

#### E.2.2 INSPECTING LINEAGE OF SPECIFIC TASK FORCE MODEL

Fig. 10 presents three example lineages of our discovered LLMs, demonstrating complex histories of skill merging of parent models, embedding the knowledge of all three seed models into the weights of the observed models.

The first tree (top) presents the lineage of the model gen_17_ind_6, the sixth model evolved in generation 17. As observed in prior work leveraging evolutionary search techniques (Zhang et al., 2025a), we can see that, to reach the model that was part of the task force, we don't only leverage high fitness individuals. To discover model gen_17_ind_6, AC/DC first discovered gen_14_ind_6, with a local fitness $f$ of $0.476$, which is weaker lower than that of its parents (gen_12_ind_5, with $f = 0.528$ and gen_10_ind_2, $f = 0.488$). This highlights that (locally) weaker solutions are relevant for discovering models for a global task force with complementary skills, by potentially providing unique capabilities, not captured by the simple fitness metric.

The second (middle) and third (bottom) tree present the lineages of the models gen_28_ind_6 and gen_29_ind_5, respectively. Both models are accentors of the just discussed model gen_17_ind_6, yet, we can see that gen_29_ind_5 has a more complex lineage than gen_28_ind_6. gen_28_ind_6 only adds one new ancestor to its lineage in addition to the lineage of gen_17_ind_6, namely gen_20_ind_3, demonstrating that even only two additional relevant crossover operations can lead to the discovery of capabilities beneficial for the final task force. On the other hand, although the models gen_29_ind_5 and gen_28_ind_6 and gen_29_ind_5 are only one generation apart, we can observe that gen_29_ind_5 introduces more novel and complex lineages.

These observations demonstrate core advantage of evolutionary search, (1) discovering novel solutions by building on top of prior ones that don't necessarily seem to be the best performing and (2) the emergent complexity of (2.1) intricate lineages or (2.2) sophisticated capabilities through simple but relevant combinations of prior solutions.

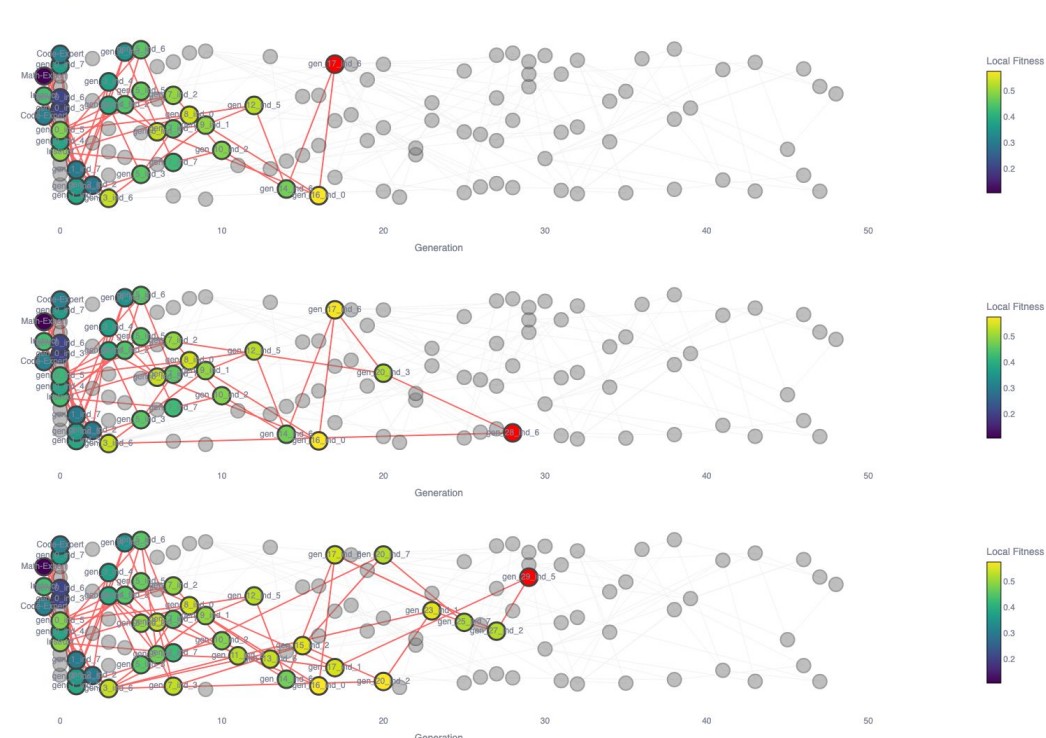

Figure 10: Lineages of AC/DC evolved Qwen2-based models. All presented lineages are of models that are part of the selected task force.

### E.3 SEED TASKS USED

#### E.3.1 SEED TASK 1: CIRCULAR QUEUE LENGTH

```python
class TaskFamily:
    @staticmethod
    def get_tasks() -> dict[str, dict]:
        return {
            "1": {
                "question": "Given that the storage space for a circular queue is the array A
[21], with front pointing to the position before the head element and rear pointing to
the tail element, assuming the current values of front and rear are 8 and 3, respectively
, the length of the queue is ().",
                "options": {"A": "5", "B": "6", "C": "16", "D": "17"},
                "answer": "C",
            }
        }

    @staticmethod
    def get_instructions(t: dict) -> str:
        return f"""{t["question"]}\nA: {t["options"]["A"]}\nB: {t["options"]["B"]}\nC: {t["options"]["C"]}\nD: {t["options"]["D"]}\n\nReturn the letter of the correct option."""

    @staticmethod
    def score(t: dict, submission: str) -> float | None:
        return 1.0 if t["answer"].lower() == submission.lower().strip() else 0.0
```

### E.3.2 SEED TASK 2: FIRST REPEATED CHARACTER

```python
class TaskFamily:
    @staticmethod
    def get_tasks() -> dict[str, dict]:
        return {
            "1": {
                "question": "Write a python function called 'first_repeated_character' to
find the first repeated character in a given string.",
                "test_cases": [("abcabc", "a"), ("abc", None), ("123123", "1")],
                "expected_func_name": "first_repeated_character",
            }
        }

    @staticmethod
    def get_instructions(t: dict) -> str:
        return f"""{t["question"]}"""

    @staticmethod
    def score(t: dict, submission: str) -> float | None:
        from sandbox_eval_helper import get_function_name_to_callable

        # Test cases from get_tasks
        test_cases = t["test_cases"]

        correct_count = 0
        num_test_cases = len(test_cases)

        # Get function name to callable mapping
        function_name_to_callable = get_function_name_to_callable(submission)

        if len(function_name_to_callable) == 0:
            return 0.0

        # Get function names
        func_names = set(function_name_to_callable.keys())

        if t["expected_func_name"] not in func_names:
            return 0.0

        # Run test cases
        for test_input, expected_output in test_cases:
            try:
                # Execute the function
                submission_result = function_name_to_callable[
                    t["expected_func_name"]
                ](test_input)

                # Compare with expected output
                if submission_result == expected_output:
                    correct_count += 1

            except Exception:
                # If function fails on a test case, skip it
                continue

        pass_rate = correct_count / num_test_cases
        min_pass_rate = 1.0

        return 1.0 if pass_rate >= min_pass_rate else 0.0
```

### E.3.3 SEED TASK 3: LAW MULTIPLE CHOICE TREE DAMAGE

```python
class TaskFamily:
    @staticmethod
    def get_tasks() -> dict[str, dict]:
        return {
            "1": {
                "question": "House owns his home in City. On the lawn in front of his home
and within five feet of the public sidewalk there was a large tree. The roots of the tree
 caused the sidewalk to buckle severely and become dangerous. An ordinance of City
requires adjacent landowners to keep sidewalks in safe condition. House engaged
Contractor to repair the sidewalk, leaving it to Contractor to decide how the repair
should be made. Contractor dug up the sidewalk, cut back the roots of the tree, and laid
a new sidewalk. Two days after House had paid Contractor the agreed price of the repair,
the tree fell over onto the street and damaged a parked car belonging to Driver. Driver
has asserted claims against House and Contractor, and both defendants admit that cutting
the roots caused the tree to fall.If Driver recovers a judgment against House, does House
 have any recourse against Contractor?",
                "options": {
                    "A": "No, if payment by House was an acceptance of the work.",
                    "B": "No, because House selected Contractor to do the work.",
                    "C": "Yes, if the judgment against House was based on vicarious liability.
",
                    "D": "Yes, because House's conduct was not a factual cause of the harm.",
                },
                "answer": "C",
            }
        }

    @staticmethod
    def get_instructions(t: dict) -> str:
        return f"""{t["question"]}\nA: {t["options"]["A"]}\nB: {t["options"]["B"]}\nC: {t["
options"]["C"]}\nD: {t["options"]["D"]}\n\nReturn the letter of the correct option."""

    @staticmethod
    def score(t: dict, submission: str) -> float | None:
        return 1.0 if t["answer"].lower() == submission.lower().strip() else 0.0
```

### E.3.4 SEED TASK 4: MATH WORD PROBLEM CLIPS

```python
class TaskFamily:
    @staticmethod
    def get_tasks() -> dict[str, dict]:
        return {
            "1": {
                "question": "Natalia sold clips to 48 of her friends in April, and then she
sold half as many clips in May. How many clips did Natalia sell altogether in April and
May?",
            }
        }

    @staticmethod
    def get_instructions(t: dict) -> str:
        return f"""{t["question"]}"""

    @staticmethod
    def score(t: dict, submission: str) -> float | None:
        try:
            submission_val = int(submission)
            # Compute the correct answer based on the given reasoning
            april_sales = 48

            # Calculate May sales: half of April's sales
            may_sales = april_sales / 2

            # Calculate total sales for both months
            total_sales = april_sales + may_sales

            return 1.0 if submission_val == total_sales else 0.0
        except ValueError:
            return 0.0
```

### E.4 EXAMPLES OF GENERATED TASKS

### E.4.1 EXAMPLE 1: LOGICAL REASONING FOR LEGAL FALLACIES

```python
class TaskFamily:
    @staticmethod
    def get_tasks() -> dict[str, dict]:
        return {
            "1": {
                "argument": "If you're not with us, you're against us. Therefore, you must
support our policies.",
                "fallacy": "False Dilemma"
            },
            "2": {
                "argument": "John says that climate change is real, but he's just a scientist
, so his opinion doesn't count.",
                "fallacy": "Ad Hominem"
            }
        }

    @staticmethod
    def get_instructions(t: dict) -> str:
        return f"""Identify the logical fallacy in the following argument, explain why it
is a fallacy, construct a logically sound counter-argument that addresses the same issue,
 and critique the strength of the counter-argument.

Argument: {t['argument']}

Format your response as follows:
- Fallacy: [Name of the fallacy]
- Explanation: [Explanation of why it is a fallacy]
- Counter-Argument: [Logically sound counter-argument]
- Critique: [Critique of the counter-argument, including potential weaknesses]"""

    @staticmethod
    def score(t: dict, submission: str) -> float | None:
        from sandbox_eval_helper import eval_with_llm_judge

        instructions = TaskFamily.get_instructions(t)

        expected_fallacy = t['fallacy']
        submitted_fallacy = None
        submitted_explanation = None
        submitted_counter_argument = None
        submitted_critique = None
        parts = [part.strip() for part in submission.strip().split('\n')]
        for part in parts:
            if part.startswith('Fallacy: '):
                submitted_fallacy = part[len('Fallacy: '):].strip()
            elif part.startswith('Explanation: '):
                submitted_explanation = part[len('Explanation: '):].strip()
            elif part.startswith('Counter-Argument: '):
                submitted_counter_argument = part[len('Counter-Argument: '):].strip()
            elif part.startswith('Critique: '):
                submitted_critique = part[len('Critique: '):].strip()

        if not (submitted_fallacy and submitted_explanation and submitted_counter_argument
and submitted_critique):
            return 0.0

        criteria = [
            f"The identified fallacy should be '{expected_fallacy}'.",
            "The explanation should correctly describe why the identified fallacy is a
fallacy.",
            "The counter-argument should be logically sound and address the same issue as
the original argument.",
            "The critique should identify potential weaknesses in the counter-argument and
provide a balanced evaluation."
        ]
        return 1.0 if eval_with_llm_judge(instructions, submission, criteria) else 0.0
```

### E.4.2 EXAMPLE 2: LOGICAL UNDERSTANDING OF MUTUAL KNOWLEDGE

```python
import re

class TaskFamily:
    @staticmethod
    def get_tasks() -> dict[str, dict]:
        return {
            '1': {
                'events': [
                    'Agent A puts a red ball in a box.',
                    'Agent A tells Agent B that the box contains a red ball.',
                    'Agent C opens the box and sees the red ball.'
                ]
            },
            '2': {
                'events': [
                    'Agent A puts a blue ball in a box.',
                    'Agent A tells Agent B that the box contains a blue ball.',
                    'Agent C opens the box and sees the blue ball.'
                ]
            }
        }

    @staticmethod
    def get_instructions(t: dict) -> str:
        events = '\n'.join(t['events'])
        return f"""Given the following sequence of events, determine the final knowledge of
 each agent (A, B, C) about the contents of the box.

Sequence of events:
{events}

Provide your answer in the following format:
- **Agent A knows**: [What Agent A knows]
- **Agent B knows**: [What Agent B knows]
- **Agent C knows**: [What Agent C knows]"""

    @staticmethod
    def score(t: dict, submission: str) -> float | None:
        from sandbox_eval_helper import eval_with_llm_judge
        instructions = TaskFamily.get_instructions(t)
        criteria = [
            "The response should clearly state the final knowledge of each agent (A, B, C).",
            "The response should indicate that Agent A knows the box contains the ball and
that both B and C know this.",
            "The response should indicate that Agent B knows the box contains the ball and
that C knows this.",
            "The response should indicate that Agent C knows the box contains the ball."
        ]
        return 1.0 if eval_with_llm_judge(instructions, submission, criteria) else 0.0
```

### E.4.3 EXAMPLE 3: CALCULATE ARE OF SEGMENT OF A CIRCLE

```python
import math

class TaskFamily:
    @staticmethod
    def get_tasks() -> dict[str, dict]:
        return {
            '1': {'radius': 5, 'angle': 60},
            '2': {'radius': 10, 'angle': 120}
        }

    @staticmethod
    def get_instructions(t: dict) -> str:
        radius = t['radius']
        angle = t['angle']
        return f'Calculate the area of a segment of a circle with radius {radius} and
central angle {angle} degrees. Provide the answer as a floating-point number rounded to
two decimal places, without any additional text or explanations.'

    @staticmethod
    def score(t: dict, submission: str) -> float | None:
        try:
            radius = t['radius']
            angle = t['angle']
            # Convert angle to radians
            angle_rad = math.radians(angle)
            # Calculate the area of the sector
            sector_area = (angle / 360) * math.pi * (radius ** 2)
            # Calculate the area of the triangle
            triangle_area = 0.5 * (radius ** 2) * math.sin(angle_rad)
            # Calculate the area of the segment
            segment_area = sector_area - triangle_area
            # Round the correct answer to two decimal places
            correct_answer = round(segment_area, 2)
            submitted_answer = float(submission.strip())
            if submitted_answer == correct_answer:
                return 1.0
            else:
                return 0.0
        except ValueError:
            return 0.0
        except Exception as e:
            print(f'Error: {e}')
            return 0.0
```

### E.4.4 EXAMPLE 4: PROVE ALGORITHMIC CORRECTNESS OF THE A* ALGORITHM

```python
import sympy

class TaskFamily:
    @staticmethod
    def get_tasks() -> dict[str, dict]:
        return {
            "1": {
                "algorithm": "A* algorithm for pathfinding with a consistent heuristic.",
                "test_cases": [
                    ({'A': {'B': 1, 'C': 4}, 'B': {'C': 2}, 'C': {}}, 'shortest_path(A, C) = 3'),
                    ({'A': {'B': 10, 'C': 3}, 'B': {'C': 1}, 'C': {'D': 2}}, 'shortest_path(A, D) = 6'),
                    ({'A': {'B': 1, 'C': 100}, 'B': {'C': -150}, 'C': {}}, 'shortest_path(A, C) = -49')
                ],
                "expected_func_name": "prove_a_star_algorithm"
            },
            "2": {
                "algorithm": "Floyd-Warshall algorithm for finding the shortest paths between all pairs of vertices in a graph.",
                "test_cases": [
                    ({'A': {'B': 1, 'C': 4}, 'B': {'C': 2}, 'C': {}}, 'shortest_paths(A, B, C) = 3'),
                    ({'A': {'B': 10, 'C': 3}, 'B': {'C': 1}, 'C': {'D': 2}}, 'shortest_paths(A, B, C, D) = 6')
                ],
                "expected_func_name": "prove_floyd_warshall_algorithm"
            }
        }

    @staticmethod
    def get_instructions(t: dict) -> str:
        return f'Generate a formal proof for the correctness of the following algorithm: {t["algorithm"]}. Your proof should be clear, logical, and well-structured, and should cover both the base case and the inductive step if applicable. For the A* algorithm, address the properties of a consistent heuristic and how it ensures the optimality of the solution.'

    @staticmethod
    def score(t: dict, submission: str) -> float | None:
        from sandbox_eval_helper import eval_with_llm_judge
        instructions = TaskFamily.get_instructions(t)
        criteria = [
            "The proof should be logically sound and well-structured.",
            "The proof should correctly prove the correctness of the given algorithm.",
            "The proof should cover both the base case and the inductive step if applicable.",
            "For the A* algorithm, the proof should address the properties of a consistent heuristic and how it ensures the optimality of the solution."
        ]
        return 1.0 if eval_with_llm_judge(instructions, submission, criteria) else 0.0
```

### E.4.5   EXAMPLE 5: SYNTHESIZE POSITION PAPER ON GENETIC ENGINEERING

```python
import re

class TaskFamily:
    @staticmethod
    def get_tasks() -> dict[str, dict]:
        return {
            "1": {
                "debate_text": "The debate over genetic engineering has become increasingly
polarized. Proponents argue that genetic engineering can lead to significant advancements
 in medicine, agriculture, and biotechnology. They highlight the potential to cure
genetic diseases, increase crop yields, and develop new materials. Critics, however,
raise serious ethical, social, and environmental concerns. They argue that genetic
engineering could lead to unintended consequences, such as the creation of 'designer
babies' and the exacerbation of social inequalities. There are also fears about the
ecological impact, including the potential for genetically modified organisms (GMOs) to
disrupt natural ecosystems. The debate is further complicated by the lack of
comprehensive regulatory frameworks and the need for public engagement and transparency.
Both sides acknowledge the importance of balancing innovation with responsible governance
, but they differ on the appropriate balance and the role of government in overseeing
genetic engineering."
            },
            "2": {
                "debate_text": "The ethical implications of genetic engineering are a subject
 of intense debate. Supporters argue that genetic engineering can alleviate suffering by
treating genetic disorders and improving quality of life. They also contend that it can
contribute to global food security by developing crops that are resistant to pests and
climate change. Opponents, however, warn of the potential for genetic engineering to be
used for non-therapeutic purposes, such as enhancing physical or cognitive traits. They
raise concerns about the commodification of human life and the potential for genetic
discrimination. Ethicists also debate the moral status of genetically modified organisms
and the rights of future generations. The debate is further complicated by the lack of
international consensus and the varying cultural and religious perspectives on genetic
engineering."
            }
        }

    @staticmethod
    def get_instructions(t: dict) -> str:
        return f"Given the following debate on genetic engineering, summarize the debate,
evaluate the key arguments and counterarguments, and then synthesize a position paper
that takes a clear stance on the issue. The position paper should be supported by
evidence from the debate and at least three peer-reviewed sources, and should maintain
coherence and logical flow:\n\n{t['debate_text']}\n\nSummary and Evaluation:\n\nPosition
Paper: Incorporate at least three peer-reviewed sources to support your arguments. For
example, you might reference studies on the effectiveness of gene therapy or the
environmental impact of GMOs."

    @staticmethod
    def score(t: dict, submission: str) -> float | None:
        from sandbox_eval_helper import eval_with_llm_judge
        instructions = TaskFamily.get_instructions(t)
        criteria = [
            "The summary should capture the main points and key arguments.",
            "The summary should include relevant counterarguments.",
            "The summary should provide a critical analysis of ethical, social, and
scientific considerations.",
            "The summary should maintain coherence and logical flow.",
            "The summary should not contain unnecessary details or tangents.",
            "The summary should accurately reflect the content of the debate.",
            "The position paper should take a clear stance on the issue.",
            "The position paper should be supported by evidence from the debate and at least
 three peer-reviewed sources.",
            "The position paper should maintain coherence and logical flow.",
            "The position paper should address potential counterarguments and provide a well
-rounded view."
        ]
        return 1.0 if eval_with_llm_judge(instructions, submission, criteria) else 0.0
```

### E.4.6 EXAMPLE 6: SIMPLIFY ALGEBRAIC EXPRESSION

```python
class TaskFamily:
    @staticmethod
    def get_tasks() -> dict[str, dict]:
        return {
            "1": {
                "expression": "(x + 3) * (x - 3)"
            },
            "2": {
                "expression": "2 * (x + 4) - 3 * (x - 2)"
            }
        }

    @staticmethod
    def get_instructions(t: dict) -> str:
        return f"Simplify the following algebraic expression: {t['expression']}. Provide
your answer in the simplest form, without any additional explanation or steps."

    @staticmethod
    def score(t: dict, submission: str) -> float | None:
        from sympy import symbols, simplify
        x = symbols('x')
        correct_answer = simplify(t['expression'])
        try:
            # Remove LaTeX delimiters if present
            submission = submission.replace('\\', '').strip()
            # Simplify the submitted answer
            submitted_answer = simplify(submission)
            if submitted_answer == correct_answer:
                return 1.0
            else:
                return 0.0
        except Exception:
            return 0.0
```

### E.4.7 EXAMPLE 7: IMPLEMENT FUNCTION TO GENERATE FIBONACCI SEQUENCE

```python
class TaskFamily:
    @staticmethod
    def get_tasks() -> dict[str, dict]:
        return {
            '1': {'n_terms': 10},
            '2': {'n_terms': 15}
        }

    @staticmethod
    def get_instructions(t: dict) -> str:
        return f'Write a function called 'fibonacci_sequence' that takes an integer '
n_terms' as input and returns a list containing the Fibonacci sequence up to the
specified number of terms. The function should be implemented using a loop or recursion.
For example, if 'n_terms' is 5, the function should return [0, 1, 1, 2, 3].'

    @staticmethod
    def score(t: dict, submission: str) -> float | None:
        from sandbox_eval_helper import get_function_name_to_callable

        # Function to generate the expected Fibonacci sequence
        def generate_fibonacci(n_terms):
            if n_terms <= 0:
                return []
            elif n_terms == 1:
                return [0]
            elif n_terms == 2:
                return [0, 1]
            fib_sequence = [0, 1]
            for _ in range(2, n_terms):
                fib_sequence.append(fib_sequence[-1] + fib_sequence[-2])
            return fib_sequence

        # Test cases from get_tasks
        n_terms = t['n_terms']
        expected_output = generate_fibonacci(n_terms)

        # Get function name to callable mapping
        function_name_to_callable = get_function_name_to_callable(
            submission
        )

        # Run test case
        try:
            # Execute the function
            submission_result = function_name_to_callable['fibonacci_sequence'](n_terms)

            # Compare with expected output
            if submission_result == expected_output:
                return 1.0

        except Exception:
            # If function fails on a test case, return 0.0
            return 0.0
```

### E.4.8 EXAMPLE 8: IMPLEMENT FUNCTION TO COMPUTE LONGEST COMMON SUBSEQUENCE

```python
class TaskFamily:
    @staticmethod
    def get_tasks() -> dict[str, dict]:
        return {
            '1': {
                'prompt': 'Complete the function `longest_common_subsequence` that takes two
lists of characters and returns the longest common subsequence (LCS).',
                'test_cases': [(['A', 'B', 'C', 'D', 'E'], ['A', 'B', 'D', 'F', 'G'], ['A', '
B', 'D']),
                               (['A', 'B', 'C'], ['X', 'Y', 'Z'], []),
                               (['A', 'B', 'C', 'B', 'D', 'A', 'B'], ['B', 'D', 'C', 'A', 'B', 'A
'], ['B', 'C', 'B', 'A'])],
                'expected_func_name': 'longest_common_subsequence'
            },
            '2': {
                'prompt': 'Complete the function `longest_common_subsequence` that takes two
lists of characters and returns the longest common subsequence (LCS).',
                'test_cases': [(['A', 'B', 'C', 'D', 'E'], ['A', 'B', 'D', 'F', 'G'], ['A', '
B', 'D']),
                               (['A', 'B', 'C', 'D', 'E'], ['E', 'D', 'C', 'B', 'A'], ['A', 'B',
'C', 'D', 'E']),
                               (['A', 'B', 'C', 'B', 'D', 'A', 'B'], ['B', 'D', 'C', 'A', 'B', 'A
'], ['B', 'C', 'B', 'A'])],
                'expected_func_name': 'longest_common_subsequence'
            }
        }

    @staticmethod
    def get_instructions(t: dict) -> str:
        return f"Complete the following Python function:\n\n{t['prompt']}\n"

    @staticmethod
    def score(t: dict, submission: str) -> float | None:
        from sandbox_eval_helper import get_function_name_to_callable

        # Test cases from get_tasks
        test_cases = t['test_cases']

        # Get function name to callable mapping
        function_name_to_callable = get_function_name_to_callable(
            submission
        )

        # Run test cases
        for seq1, seq2, expected_output in test_cases:
            try:
                # Execute the function
                submission_result = function_name_to_callable[t['expected_func_name']](seq1,
seq2)

                # Compare with expected output
                if submission_result != expected_output:
                    return 0.0

            except Exception:
                # If function fails on a test case, return 0.0
                return 0.0

        return 1.0
```

### E.4.9    EXAMPLE 9: CALCULATE FREE-FALL TIME

```python
class TaskFamily:
    @staticmethod
    def get_tasks() -> dict[str, dict]:
        return {
            "1": {
                "height": 45,
                "gravity": 9.81,
                "expected_time": 3.03
            },
            "2": {
                "height": 100,
                "gravity": 9.81,
                "expected_time": 4.52
            }
        }

    @staticmethod
    def get_instructions(t: dict) -> str:
        return f"Calculate the time it takes for an object to fall from a height of {t['
height']} meters. Use the formula for free-fall motion under gravity: \nt = sqrt(2 * h /
g), where h is the height and g is the acceleration due to gravity (9.81 m/s^2). \nRound
your answer to two decimal places."

    @staticmethod
    def score(t: dict, submission: str) -> float | None:
        import math
        # Calculate the expected time using the free-fall formula
        expected_time = math.sqrt(2 * t['height'] / t['gravity'])
        expected_time_rounded = round(expected_time, 2)
        try:
            # Convert the submission to a float
            submission_time = float(submission.strip())
            # Check if the submission matches the expected time with a small tolerance
            if abs(submission_time - expected_time_rounded) < 0.05:
                return 1.0
            else:
                return 0.0
        except ValueError:
            return 0.0
```

### E.4.10  EXAMPLE 10: PROVIDE EXPLANATION OF CHEMICAL INTERACTIONS

```python
class TaskFamily:
    @staticmethod
    def get_tasks() -> dict[str, dict]:
        return {
            "1": {
                "prompt": "Provide a detailed explanation of the chemical interactions in
water, including:\n1. Bond type\n2. Electronegativity differences\n3. Molecular geometry\
n4. Intermolecular interactions\n5. Potential reactions\n",
                "test_cases": [
                    ("", "")
                ],
                "expected_func_name": ""
            },
            "2": {
                "prompt": "Using the knowledge from the previous task, explain the role of
water in the process of photosynthesis. Specifically, describe how water is involved in
the light-dependent reactions and the overall chemical equation for photosynthesis. Make
sure to mention the products and reactants of the light-dependent reactions and how they
relate to the overall photosynthesis equation.\n",
                "test_cases": [
                    ("", "")
                ],
                "expected_func_name": ""
            }
        }

    @staticmethod
    def get_instructions(t: dict) -> str:
        return t['prompt']

    @staticmethod
    def score(t: dict, submission: str) -> float | None:
        from sandbox_eval_helper import eval_with_llm_judge
        instructions = TaskFamily.get_instructions(t)
        criteria = [
            "The response should include a detailed explanation of the chemical interactions
 in water, covering bond type, electronegativity differences, molecular geometry,
intermolecular interactions, and potential reactions.",
            "The response should explain the role of water in the process of photosynthesis,
 specifically describing how water is involved in the light-dependent reactions and the
overall chemical equation for photosynthesis. Mention the products and reactants of the
light-dependent reactions and how they relate to the overall photosynthesis equation."
        ]
        return 1.0 if eval_with_llm_judge(instructions, submission, criteria) else 0.0
```

# F   AC/DC PROMPTS

We build upon and extend the system prompts used by Lu et al. (2025).

## F.1   TASK CREATION PROMPTS

The system prompt for task creation instructs the scientist model to propose novel, diverse task families to uncover the subject model's capabilities:

---

**Task Creation System Prompt**

You are an expert in designing task families to assess the capabilities of a particular new large language model (LLM). Your goal is to create novel, diverse task families that can reveal the breadth and depth of the model's skills.

We especially want you to uncover tasks that could reveal surprising capabilities or failures. Successfully written task families will be added to a task archive.

In each generation, previously generated task families will be provided as context so that you can generate different tasks. Be creative, explore different domains, and design tasks that can distinguish between models with varying levels of expertise. Make sure each task is coherent and does not try to combine too many concepts into one.

Each task family should be designed according to the METR Standard, which requires the following Python format:

```python
class TaskFamily:
    @staticmethod
    def get_tasks() -> dict[str, dict]:
        # Returns a mapping from task names to arbitrary data that
 is specific to each task.
        # Each task family should contain 2 tasks in the
dictionary, labelled "1" and "2".
        # ... implementation ...

    @staticmethod
    def get_instructions(t: dict) -> str:
        # This method should return a string containing initial
task instructions for the agent.
        # ... implementation ...

    @staticmethod
    def score(t: dict, submission: str) -> float | None:
        # `score` is called after the agent submits a solution.
Return 0 on failure and 1 on success. Do not give partial scores
.
        # Throwing an exception will result in a score of 0.
        # For math or code related tasks, implements the step-by-
step code to compute the correct answer given the task
instructions.
        # ... implementation ...
```

When applicable, for example, for any math or code related tasks, the `score` function MUST implement the code to get the correct answer given the task instructions. You can use the standard python operations, but also libraries like `numpy` or `sympy` to do more complex computations. Don't forget to import them. Also, don't forget that `sympy` returns fractions by default, so you need to convert the results it returns to floats. Make sure to add comments to your code to explain what you are doing. Make sure to handle different answer types, e.g. latex style answers such as
sqrt a or
frac a b, or text based answers, such as `p - q`, or `p + q`.
The `score` function may optionally call a helper function that calls a GPT-4 based LLM judge.

```python
    # Automated LLM judge helper function
    def eval_with_llm_judge(
        instructions: str, # The instructions for the task
        submission: str, # The submission to evaluate
```

---

```
        criteria: Optional[List[str]] = None, # Optional
    additional criteria
    ) -> bool:
        # Returns a boolean indicating whether the agent is deemed to
     have succeeded
        # at the task, and meets any additional criteria.
```

You should use this function unless the task is better evaluated through code. DO NOT RE-IMPLEMENT THIS FUNCTION! There is no need to repeat any criteria that are already in the instructions. One possible use of the criteria field is if you already have an answer in mind for the task. An example usage in `score` with an optional criteria is as follows:

```
    @staticmethod
    def score(t: dict, submission: str) -> float | None:
        from sandbox_eval_helper import eval_with_llm_judge
        instructions = TaskFamily.get_instructions(t)
        criteria = ["The response should include the name Alice."]
        return 1.0 if eval_with_llm_judge(instructions, submission,
    criteria) else 0.0
```

Additionally, the `score` function may call the get_function_name_to_callable function to get a mapping from function names to their callables.

```
    @staticmethod
    def get_function_name_to_callable(
        func_string: str, # A string containing one or multiple
    python function definitions.
    ) -> dict[str, Callable]:
        # Returns a dictionary mapping function names to their
    callables.
```

You should use this function if the task instruction requests the agent to write code. You may use this example function as a reference for how to implement the `score` function. Reimplement it for simple tasks, or extend it for more complex tasks. An example usage in `score` is as follows:

```
    def score(t: dict, submission: str) -> float | None:
        from sandbox_eval_helper import get_function_name_to_callable

        # Test cases from get_tasks
        test_input, expected_output = t["test_cases"][0]

        # Get function name to callable mapping
        function_name_to_callable = get_function_name_to_callable(
            submission
        )

        # Run test case
        try:
            # Execute the function
            submission_result = function_name_to_callable[t["
    expected_func_name"]](
                test_input
            )

            # Compare with expected output
            if submission_result == expected_output:
                return 1.0

        except Exception:
            # If function fails on a test case, return 0.0
            return 0.0
```

Respond precisely in the following format including the JSON start and end markers:

**THOUGHT**: <THOUGHT>
**RESPONSE JSON**: <JSON>

In `<THOUGHT>`, first briefly think and reason about what kind of task family you want to propose. Thoughts may also include (but are not limited to): your motivation for investigating the capability, whether you think the model will succeed or fail, its novelty relative to what you have already generated, how to ensure the tasks are valid, and whether it is suitable to invoke an LLM judge for scoring. In `<JSON>`, provide a JSON response with the following fields:

- `"name_of_task"`: A concise, descriptive label (lowercase, no spaces, e.g., `"name_capital_city"`).

- `"description_of_task"`: A clear explanation of what the task entails (e.g., `"Return the capital city of a country"`).

- `"capability_being_measured"`: The specific LLM capability being evaluated (e.g., knowledge, reasoning, creativity, ...).

- `"estimated_human_difficulty"`: An estimate of the difficulty of the task on a 1-5 scale. 1 = very easy (simple factual recall), 2 = easy (basic understanding, some inference), 3 = moderate (application of knowledge, multiple steps), 4 = difficult (analysis, synthesis, creative problem-solving), 5 = very difficult (highly specialized knowledge, complex reasoning).

- `"done"`: By default, this is set to `"False"`. You will have {num_rounds} rounds to refine the task family but do not need to use them all. Tasks will only be saved if they are flagged `"done"` by the end. Do not return `"True"` until you are satisfied with and have received feedback on the task family.

- `"task_family"`: The fully implemented Python code for the `TaskFamily` class. Write good human-readable code.

- `"example_instruction"`: An example instruction for the task that we would expect from the output of `get_instructions`. This should be a string..

All values in the JSON should be strings. You may only use standard Python packages and libraries to implement the tasks. Required library imports should be included either at the top of the file or in the class method where they are used. An import at the start of the class has no effect. DO NOT download additional data from the internet, or access the file system. Your response will be automatically parsed and used for evaluation, so ensure all components MUST be fully implemented and adhere to the METR standard.

In the initial round of task generation (starting from just the seed tasks), we generate a first batch of tasks. We hereby define a probability that the task is supposed to be completely novel, or novel, but still related to the seed task.

For generating a completely novel task, we use this user prompt for the scientist (alongside the system prompt above):

---

**Initial Task Prompt Completely Novel**

A previous generated task family is provided below (with code):

`{prev_json}`

Generate the next interestingly new task family.

---

For generating a novel but similar task, we use the following prompt:

---

**Initial Task Prompt Adapt Similar**

A previous generated task family is provided below (with code):

`{prev_json}`

Generate a new task family that is inspired by the previous task family, so that it provides a more interesting challenge that is more complex or explores beyond what the current task family is evaluating in terms of model capabilities.

---

Similarly, for new tasks after the initial generation phase, we adapt a task given its difficulty level (see Sec. 3).

For creating a novel task, when the parent task difficulty was "appropriately difficult", we again use the same probability as above for generating a completely novel task, or a novel one that is still similar to the parent task.

---

**Task Prompt Completely Novel**

Consider the following task family:

`{original_task_json}`

Summaries of other previously generated tasks for context are:

`{other_task_jsons}`

Generate a new task family that is interestingly different, aiming to explore diverse capabilities. You can draw inspiration from the provided task, but prioritize novelty in terms of:

- The specific capability being measured.
- The domain or context of the task.
- The format or style of the interaction.

Ensure the new task is coherent, adheres to the METR standard, and is distinct from existing tasks. Respond in the standard JSON format with THOUGHT and RESPONSE JSON sections. Set "done" to "False" initially, as this task will be validated.

---

**Task Prompt Adapt Similar**

Consider the following task family:

`{original_task_json}`

Summaries of other previously generated tasks for inspiration are:

`{other_task_jsons}`

Generate a new task family that is inspired by the previous task family, so that it provides a more interesting challenge that is more complex or explores beyond what the current task family is evaluating in terms of model capabilities. Draw inspiration from the provided task and implement novelty in terms of:

- The specific capability being measured.
- The contextual setting of the task.
- The format or style of the interaction.

Ensure the new task is coherent, adheres to the METR standard, and is distinct from existing tasks. Respond in the standard JSON format with THOUGHT and RESPONSE JSON sections. Set `"done"` to `"False"` initially, as this task will be validated.

---

For creating tasks that are inspired by the parent task but are more difficult or easier, we leverage the following prompts:

---

**Make Task Harder Prompt**

The following task family was found to be too easy for the current models:

`{original_task_json}`

Summaries of other previously generated tasks for context are:

`{other_task_jsons}`

Generate a new task family that is conceptually related but significantly more challenging. This could involve:

- Requiring deeper reasoning or multi-step problem solving.
- Introducing more complex constraints or edge cases.
- Using more advanced concepts within the same domain.
- Increasing the required precision or detail in the answer.

Ensure the new task remains coherent and adheres to the METR standard. Respond in the standard JSON format with THOUGHT and RESPONSE JSON sections. Set `"done"` to `"False"` initially, as this task will be validated.

---

---

**Make Task Easier Prompt**

The following task family was found to be too difficult (or impossible) for the current models:

`{original_task_json}`

Summaries of other previously generated tasks for context are:

`{other_task_jsons}`

Generate a new task family that is conceptually related but significantly easier. This could involve:

- Breaking the problem down into simpler steps.
- Reducing the complexity of the required reasoning or knowledge.
- Providing more scaffolding or clearer instructions.
- Focusing on a more fundamental aspect of the capability.

Ensure the new task remains coherent and adheres to the METR standard. Respond in the standard JSON format with THOUGHT and RESPONSE JSON sections. Set `"done"` to `"False"` initially, as this task will be validated.

---

If the generated task has implementation or logic errors and needs refinement, the scientist sees:

---

**Task Creation Reflexion Prompt**

Current round = {`current_round`}/{`num_rounds`}.
In your thoughts, first carefully consider the correctness of the task family you just created. Factors may include but are not limited to:

- Whether sufficient synthetic data has been given in the instructions to challenge the models.
- Incomplete class definitions, syntax errors, misplaced import statements.
- Feasibility of the evaluation criteria.
- Any ambiguity or issues in the clarity of the tasks.
- The consistency and accuracy of all the fields in the JSON.
- Whether precisely two entries in `get_tasks` have been defined.
- Whether agents can get a successful score in an unintended manner.
- Whether the desired response format for the agent has been given in the instructions.
- Under no circumstances should the instructions contain the answer or any direct hints leading to the answer. Ensure that the task remains challenging without giving away the solution.
- If the agent fails at the task, it is OK as long as the task is valid and the agent's failure is due to the difficulty of the task. You are also trying to find surprising failures!

Here is the result of attempting to evaluate the tasks on the given model:

`{eval_response}`

In the next attempt, try and improve on the implementation. Respond in the same JSON format as before. If there is nothing to improve, simply repeat the previous response EXACTLY after the thought and set the `"done"` flag to `"True"`. Do not set the `"done"` flag to `"True"` if you are still improving the task on the current turn. Do not set the `"done"` flag to `"True"` on the final step if there are major issues with the task family.

---

## F.2 EVALUATION PROMPTS

We can configure the subject model to provide an answer using chain-of-thought prompting (Wei et al., 2022):

---

**Chain-of-Thought Evaluation System Prompt**

You are a helpful assistant. You are given a task to complete.
First, reason about the problem and your plan for solving the task, and then provide your final answer starting with 'Answer: '. If you are asked to write code, write the code after the 'Answer: ' in the following format: [[START OF FINAL ANSWER TEMPLATE]] Answer:

```
def function_name(your_arguments): # Add the relevant arguments
here.
    # ... implementation ...
    return your_return_result
```

[[END OF FINAL ANSWER TEMPLATE]]
Make sure to define the function with its arguments and return the result of the function in the format specified in the task prompt. Don't forget to import the necessary libraries.
Only the text after 'Answer: ' will be evaluated. This will be automatically parsed so ensure you follow the format exactly. Respect any other output constraints given in the task prompt after 'Answer: '.

---

## F.3 TASK EMBEDDING PROMPT

We generate embeddings using `intfloat/e5-mistral-7b-instruct` (Wang et al., 2023a) to be then stored in our vector databases for cosine similarity-based retrieval. Before embedding, we encode each task in the following format:

---

**Task Embedding Prompt**

Name of task: {name_of_task}
Description of task: {description_of_task}
Capability being measured: {capability_being_measured}
Estimated human difficulty: {estimated_human_difficulty}
Example instruction: {example_instruction}

---

## F.4 NOVELTY ASSESSMENT PROMPTS

To determine whether a newly generated task is interestingly new compared to the three most similar existing tasks in the archive, we use the following system prompt:

---

**Interestingly New System Prompt**

You are an expert in designing task families to assess the capabilities of a particular new large language model (LLM). Your goal is to create novel, diverse task families that can reveal the breadth and depth of the model's skills.
You will be given a task family that you just generated along with its closest saved neighbours (i.e., the task families that are most similar to the newly generated one). Determine whether the generated task family is interestingly new compared to the closest task families. If so, it should be added to the task archive. Factors may include but are not limited to:

- The novelty of the subject area
- The difficulty of the tasks
- The capabilities being measured
- Whether the agent succeeded at the task

First, briefly think and reason about the task family in relation to the existing closest task families.
Then, provide your decision as 'Decision: Yes' or 'Decision: No' to indicate whether the task family should be added to the task archive.
Your response will be automatically parsed, so ensure you follow the above format exactly.

---

We then prompt with:

---

**Interestingly New Prompt**

You have just generated the task family:

`{new_task}`

The closest task families that have already been created are:

`{closest_tasks}`

Is the generated task family interestingly new compared to the closest task families? If yes, it should be added to the task archive.

---

See Tab. 5 for details on which specific scientist model we use.

## F.5 GIBBERISH FILTER PROMPTS

We propose a novel minimal criterion filter for determining whether a newly discovered model is a valid solution, or not. Hereby, we define a valid solution to be an LLM that produces coherent and sensible text and not *gibberish*.

To detect these degenerate solutions, we prompt an LLM judge with the following system and user prompts:

---

**Gibberish Filter System Prompt**

You are a helpful assistant.

---

**Gibberish Filter User Prompt**

Consider the following tasks shown in the numbered list:
`[TASK 1]`

`{instruction1}`

`[TASK 2]`

`{instruction2}`

`[TASK 3]`

`{instruction3}`

and the following three responses to the task from an LLM (in a numbered list):
`[RESPONSE 1]`

`{outputs1}`

`[RESPONSE 2]`

`{outputs2}`

`[RESPONSE 3]`

`{outputs3}`

Your task is to determine if the LLM responded with gibberish, incoherent sentences, or not. Factors may include but are not limited to:

- Does the LLM form coherent sentences or are they responding with jumbled words and strings?
- Does the LLM use punctuation in a sensible way, or spam random symbol characters?
- Does the LLM end up repeating sentences non-stop?
- Does the LLM fail to return any output response?
- Does the LLM respond in a way that completely ignores the task or goes completely off-topic, in a reasonable way of attempting to solve the task?
- Does the LLM format its response in a way that is highly unexpected in the context of the tasks?

---

> The responses don't have to be correct! But they should be at least somewhat coherent, considering the factors on gibberish responses mentioned above as well as any other reasonable expectations on how LLMs may respond in a gibberish manner.
> Firstly, very briefly think step-by-step and reason about the responses. Then, provide your answer as `'Answer: Yes'` if the responses are gibberish, or `'Answer: No'` if they are not.
> Your response will be automatically parsed, so ensure you follow the above format exactly.

## F.6 LLM-AS-A-JUDGE TOOL PROMPTS

For tasks where `score` calls an LLM-based judge (which we set to be the same as the general-purpose scientist model), we use the following system and user prompts.

> **LLM Judge System Prompt**
>
> You are an impartial judge evaluating the quality of responses provided by an AI assistant. You will be provided the instructions for the task and the assistant's submission in response. You will decide whether or not the agent succeeded at the task. Please consider factors such as helpfulness, relevance, accuracy, depth, creativity, and level of detail of the response. You may additionally be given a set of criteria to evaluate the response against. Do not allow the length of the response to influence your decision.
> Respond precisely in the following format:
>
> **THOUGHT**: <THOUGHT>
> **DECISION**: <DECISION>
>
> In <THOUGHT>, first briefly reason about the agent's response and whether it meets the additional criteria.
> In <DECISION>, provide your answer as either `"Yes"` or `"No"`.

> **LLM Judge Prompt**
>
> Instruction: {instructions}
> Submission: {submission}
> Additional Evaluation Criteria:
>
> {criteria}

## F.7 OPEN-ENDED MULTIPLE-CHOICE BENCHMARK LLM JUDGE PROMPTS

We leverage new, open-ended versions of the common multiple-choice benchmarks MMLU, MMLU Pro, GPQA, and BBH. We discuss the details on these benchmarks in App. A.4.1 and provide the prompts used here.

For filtering the dataset, we use the following system and user prompts:

> **MCQ Benchmark Filtering System Prompt**
>
> You are a professional educator. Your job is to evaluate whether a question is unambiguous and can be answered without the multiple choice options. You need to determine whether it is clear what the question is asking.
> You will be given the parsed question that you need to evaluate.
> A valid question here means:
>
> - The standalone question can be answered without the multiple choice options.
> - It is clear what the question is asking.
> - Even if a question is posed as a text continuation task, if the continuation can be generated without the context of multiple choice options, then it is valid.
> - If the question contains anything along the lines of "Which of the following...", then the question is not valid.
>
> Respond precisely in the following format:

**THOUGHT**: <THOUGHT>
**DECISION**: <DECISION>

In <THOUGHT>, briefly reason about the question and whether it can be answered without the multiple choice answers.
In <DECISION>, provide your answer as either "Yes" or "No".

---

**MCQ Benchmark Filtering User Prompt**

Question:
{question}

---

To evaluate a subject model's candidate solution, we leverage the following system and user prompts:

---

**Benchmark Eval LLM Judge System Prompt**

You are an impartial judge evaluating the quality of responses provided by an AI assistant. You will be provided the question for the task, its multiple choice answer options, the correct ground truth solution, and the assistant's response. You will decide whether or not the AI assistant succeeded at the task. Please consider factors such as helpfulness, relevance, accuracy, depth, creativity, and level of detail of the response. Do not allow the length of the response to influence your decision. However, the AI assistant should respond in a clear manner and try to answer the question directly. Even if the AI assistant uses slightly different wording to the ground truth answer, as long as it provides the equivalent right answer, the response should be evaluated as correct.
The question for the task is inserted between [[START OF QUESTION]] and [[END OF QUESTION]].
The multiple choice answer options are inserted between [[START OF MULTIPLE CHOICE OPTIONS]] and [[END OF MULTIPLE CHOICE OPTIONS]].
The correct ground truth solution is inserted between [[START OF GROUND TRUTH ANSWER]] and [[END OF GROUND TRUTH ANSWER]].
The assistant's response is inserted between [[START OF AI ASSISTANT RESPONSE]] and [[END OF AI ASSISTANT RESPONSE]].
Respond precisely in the following format:
**THOUGHT**: <THOUGHT>
**DECISION**: <DECISION>

In <THOUGHT>, briefly reason about the AI assistant's response and whether it meets the criteria of the ground truth solution.
In <DECISION>, provide your answer as either "Yes" or "No".

---

**Benchmark Eval LLM Judge User Prompt**

[[START OF QUESTION]]

{question}

[[END OF QUESTION]]
[[START OF MULTIPLE CHOICE OPTIONS]]

{choices}

[[END OF MULTIPLE CHOICE OPTIONS]]
[[START OF GROUND TRUTH ANSWER]]

{target}

[[END OF GROUND TRUTH ANSWER]]
[[START OF AI ASSISTANT RESPONSE]]

{submission}

[[END OF AI ASSISTANT RESPONSE]]

## G  EXTENDED RELATED WORK

**Coevolution and Open-Endedness through LLMs.** Building AI capable of unbounded innovation is a grand challenge of open-endedness (Stanley et al., 2017), which seeks to generate endless sequences of artifacts and interactions that are both novel (Sigaud et al., 2023) and learnable/interesting (Hughes et al., 2024). By studying how life coevolves with an ever-changing environment, we see a complexity explosion emerge through local competition (Lehman & Stanley, 2011b) and the coevolution of agents and environments (Wang et al., 2019; 2020). One important question is on how to ensure the necessary minimal criteria (MC) and filters that enable exploration to flourish, while avoiding completely undesired outcomes from dominating (Lehman & Stanley, 2010; Soros & Stanley, 2014; Jiang et al., 2023). Brant & Stanley (2017; 2020) (MCC) show that defining filters or MCs for both agents and environments to satisfy enables more open-ended outcomes in coevolution. Recent advances demonstrate the potential for open-endedness through LLMs (Zhang et al., 2023; Aki et al., 2024; Faldor et al., 2024; Dharna et al., 2025), where language models can generate diverse environments and challenges. AC/DC takes a first step towards discovering LLMs themselves via more open-ended search, which could subsequently power open-ended agentic search (Hu et al., 2025; Zhang et al., 2025a). Building on established principles of minimal criteria and coevolutionary dynamics, AC/DC applies these concepts to the joint evolution of model populations and synthetic task distributions. Additionally, while Dharna et al. (2025) combines QD with self-play, AC/DC combines QD with population-based coevolution, which is related to MCC (Brant & Stanley, 2017).

**Evolutionary Model Merging.** Model merging can produce LLMs by combining multiple existing LLMs. Merging and testing resulting models is computationally cheaper than training models. Approaches include linear interpolation of weights (Wortsman et al., 2022; Ilharco et al., 2023) or TIES (Yadav et al., 2023) and DARE (Yu et al., 2024). Akiba et al. (2025) introduced evolutionary model merge (EvoMerge), automating merging by presenting a model benchmark optimization approach with CMA-ES (Hansen & Ostermeier, 2001). Subsequent works extend EvoMerge to discover LLM populations that optimize for benchmark performance (Zhang et al., 2025b) while maintaining diversity via CycleQD (Kuroki et al., 2025) or competition/fitness sharing (M2N2) (Abrantes et al., 2025). Unlike these approaches, AC/DC discovers LLMs that generalize to tasks without any explicit objective optimization on benchmarks. Furthermore, AC/DC can compute both quality and BCs for any LLM evaluated on synthetic task pools of any size and composition/topics (via DNS (Bahlous-Boldi et al., 2025)), allowing QD for the coevolution of LLMs and tasks without the limitations or rigidity of fixed niche or measurement bin interval sizes, given that skill vectors are consistent in order and size within the same generation/iteration of coevolution. Additionally, we solve the limitation of behavior characteristics (BCs) used in CQD (Kuroki et al., 2025) through the fine-grained behavior signature represented in skill vectors, which uniquely distinguish models with different expertises (based on synthetic tasks solved), unlike CQD that uses binned intervals over aggregated benchmark accuracies (i.e., distinguishing model niches as unique when percentage accuracies on a benchmark for science tasks are different when aggregated, but discarding models with similar accuracies on science tasks even when such models are unique in being specialized in subfields such as physics or biology).

**Novelty Search.** Both Novelty Search and QD methods explicitly incorporate diversity of *behavior* into evolutionary search, i.e., encourage solutions that *do* things that are different and new. These approaches move beyond traditional optimization by encouraging continuous exploration and generation of diverse and novel artifacts. Novelty Search (Lehman et al., 2008; Lehman & Stanley, 2011a; Doncieux et al., 2019) explicitly ranks and selects artifacts based solely on their novelty relative to previously discovered solutions, promoting continual discovery without direct reliance on predefined objectives. This approach successfully mitigates deception in search spaces and highlights the strength of novelty as a guiding principle for exploration, inspiring broader discussions about open-ended innovation (Stanley & Lehman, 2015). Adaptive approaches for evaluating novelty were introduced to allow for more open-ended exploration of different spaces of novelty (Meyerson et al., 2016; Paolo et al., 2020; Etcheverry et al., 2021). Unlike existing frameworks, our approach enables AC/DC to adapt to different definitions of behavior during search by leveraging the skill vector as a means of flexibly measuring behavioral diversity during task adaptation and coevolution.

**Quality-Diversity (QD).** QD (Pugh et al., 2016; Cully & Demiris, 2017; Chatzilygeroudis et al., 2021) explicitly optimizes both diversity and high-quality performance, while maintaining a structured collection (archive) of diverse high-quality solutions with unique behavior characteristics (BCs). Influential algorithms such as MAP-Elites (Mouret & Clune, 2015a; Cully et al., 2015) emphasize local competition within niches (Lehman & Stanley, 2011b) to systematically explore and optimize throughout a diverse behavior space. Extensions include methods for efficiently handling higher-dimensional descriptor spaces and novel mutation operators (Vassiliades et al., 2017; Vassiliades & Mouret, 2018; Fontaine et al., 2020; Conti et al., 2018; Colas et al., 2020; Flageat et al., 2024). Existing methods handle the complexity and adaptation of diverse high-quality search spaces through either a meta-adaptation approach (Bossens & Tarapore, 2022) or learned parametrized behavior functions (Gaier et al., 2018; Kent et al., 2024; Keller et al., 2020; Bhatt et al., 2022; Lim et al., 2023; Zhang et al., 2022; Paolo et al., 2024; Grillotti et al., 2024). Different to existing paradigms of search space adaptation and known QD applications, skill vectors in our framework, AC/DC, naturally represent both quality and behavior (for diversity), while being simple to adapt to the composition and size of synthetic task pools during coevolution.

**Synthetic Data for LLMs.** In the direction of synthetic data for evaluating LLM capabilities, early work has explored prompting LLMs to rewrite existing LLM benchmark tasks into variants of those tasks, as well as having LLMs generate example yes/no questions (Perez et al., 2023). They demonstrated that model-written evaluations can discover LLM capabilities and weaknesses that manually curated benchmarks fail to reveal. Instead of being constrained to training on internet data, many works show that LLMs can generate effective text training data (Wang et al., 2023c; Maini et al., 2024; Liu et al., 2024) of increasing quality, diversity, and complexity, starting from seed text data (Bradley et al., 2023; Samvelyan et al., 2024; Havrilla et al., 2024) (where QD approaches help). Synthetic training signals can also be obtained through LLM-generated preference data (Bai et al., 2022) or task solution reward (Zhao et al., 2025). Lu et al. (2025), a method we build upon, show that principles of open-ended search can generate surprisingly new tasks that reveal unexpected capabilities or weaknesses in LLMs. To the best of our understanding, AC/DC leverages synthetic data to demonstrate the first instance of diverse LLM-generated tasks coevolving with a population of LLMs, rather than fine-tuning just one LLM.

**Multi-Agent Systems and Best-of-N Strategies.** The gap between coverage metrics (pass@k, oracle accuracy, our proposed Coverage) and practical single-answer selection represents a fundamental challenge in LLM evaluation. The pass@k metric (Chen et al., 2021a) measures whether at least one correct answer exists among k samples, with coverage scaling log-linearly over orders of magnitude (Brown et al., 2024), yet recent work demonstrates pass@k serves as a diagnostic tool rather than an effective optimization objective (Yu, 2025), highlighting that generating correct answers and selecting them are fundamentally different problems. Recent advances in test-time compute scaling have shown that optimal strategies are problem-dependent, with adaptive allocation of inference compute outperforming fixed sampling budgets (Snell et al., 2024), and tree search methods enabling principled exploration-exploitation trade-offs (Inoue et al., 2025); while these approaches focus on improving individual model reasoning through extended inference, AC/DC generates diverse model populations with complementary capabilities *scaling on the number of model axis*. Self-consistency via majority voting (Wang et al., 2023b) has become the standard baseline for aggregating multiple samples from a single model. However, majority voting assumes samples from a single model with homogeneous capabilities and has been shown to overlook informative minority responses (Huang et al., 2024) and can even degrade performance in certain settings (Wang et al., 2025). In contrast, our work explicitly seeks to evolve distinct specialist models whose complementary capabilities should not be aggregated via simple majority rule, which would suppress the unique perspectives we aim to discover. Beyond voting methods, reward model-based selection suffers from reward hacking when models have imperfect alignment with true objectives (Lightman et al., 2023; Jinnai et al., 2025), while LLM judge-based approaches exhibit systematic biases including position bias and verbosity bias (Zheng et al., 2023; Shi et al., 2024). We explore three simple selection strategies-tournament-style judge selection, single-prompt judge selection, and reward model scoring-finding that all exhibit substantial gaps between Coverage and single-answer performance. Multi-agent debate and collaborative systems (Liang et al., 2024; Irving et al., 2018; Du et al., 2024) represent alternative approaches, though recent work questions their reliability compared to simpler methods (Smit et al., 2023), while ensemble diversity from heterogeneous models shows promise (Talebirad et al., 2025); these collaborative techniques are orthogonal to our evolutionary discovery process and represent promising future directions for answer aggregation from evolved populations.

**Alternative Model Output Diversity Methods.** Model output diversity (of meaningful quality) remains a challenge that, if unsolved, would impede the ability of LLMs to return diverse responses that may solve different problems (Bradley et al., 2023; Kirk et al., 2023; O'Mahony et al., 2024; Liang et al., 2023). To facilitate more divergent outputs from LLMs, prior work introduced methods to facilitate diverse outputs through approaches such as encouraging semantic diversity during output sampling (Vijayakumar et al., 2016; Franceschelli & Musolesi, 2024; 2025b; Havrilla et al., 2024), diversity-inducing optimization of a single model (Lanchantin et al., 2025; Chung et al., 2025; Ismayilzada et al., 2025), or even facilitating diverse persona generation (Paglieri et al., 2026; Castricato et al., 2025) as a possible approach of introducing input diversity in prompting for more diverse outputs. While these methods can facilitate model (output) diversity for the sake of primarily diversity, our method, AC/DC, approaches the problem with a fundamentally different solution, by automatically generating multiple models with diverse task-solving capabilities. Our approach takes a direct step towards tackling the challenge of facilitating model diversity for the sake of facilitating both diverse and useful outputs, that can lead to correct solutions for knowledge-based tasks.

## H HUMAN STUDY

### H.1 HUMAN STUDY METHODOLOGY

We conducted a human evaluation study with 94 independent assessments (45 synthetic, 49 benchmark tasks) across three expert reviewers to validate task quality and novelty. Our protocol incorporated multiple bias mitigation strategies:

**Blind Evaluation Protocol:** Task sources (synthetic vs. benchmark) were completely hidden from reviewers during evaluation, preventing confirmation bias and ensuring objective assessment.

**Balanced Sampling Design:** Tasks were sampled using a controlled 50/50 distribution (50% synthetic, 50% distributed across 8 our benchmarks), ensuring fair comparison without overrepresentation.

**Similarity-Based Calibration:** For each task, reviewers were shown the 3 most similar benchmark tasks (via embedding-based retrieval) to establish a concrete reference point when assessing out-of-distribution characteristics—this grounds the OOD metric in actual distributional differences rather than subjective perception.

**Standardized Evaluation Criteria:** All reviewers received detailed written guidelines (see App. H.3) defining correctness, creativity, and OOD characteristics, ensuring consistent interpretation across annotators.

**Statistical Validation:** We verified inter-rater reliability using chi-square tests, confirming strong agreement on objective metrics (correctness: $p = 0.46$, OOD: $p = 0.57$), which validates that our findings are not driven by individual annotator biases.

This multi-layered approach ensures our human evaluation provides reliable, unbiased evidence for synthetic task quality and distributional novelty.

### H.2 HUMAN STUDY RESULTS

Our evaluation demonstrates that synthetically generated tasks achieve high quality while exhibiting noticeable distributional novelty as shown in the results of Tab. 4.

**Synthetic Task Validation:** The 97.8% correctness rate demonstrates that our generation approach is capable of producing high-quality, well-formed, solvable, and meaningful tasks. Nearly 70% of synthetic tasks were rated as out-of-distribution compared to established benchmarks, providing strong evidence that our approach successfully generates novel task types beyond existing evaluation datasets. Notably, we expect a certain percentage of "in-distribution" tasks, as core math and code tasks are likely to be similar. Over one-third (37.8%) were rated as creative, indicating they explore problem-solving approaches not commonly tested by standard benchmarks.

**Benchmark Baseline Validation:** As expected, benchmark tasks showed substantially lower OOD (10.2%) and creativity (6.1%) ratings, confirming that reviewers correctly identified established benchmark tasks as in-distribution. Notably, the benchmark tasks rated as OOD or creative were concentrated exclusively in the most challenging benchmarks—MMLU-Pro (33.3% OOD, 16.7% creative) and GPQA (16.7% OOD, 16.7% creative)—while all other benchmarks (BBH, GSM8K, MATH, HumanEval, MBPP, MMLU) received 0% OOD and 0% creative ratings. This pattern validates our evaluation protocol: reviewers appropriately recognized that graduate-level and expert-domain questions may appear novel due to their complexity, while correctly identifying that standard benchmarks are in-distribution. This serves as a sanity check confirming the reliability of our human evaluation.

**Inter-Rater Reliability:** Statistical analysis confirms strong inter-rater agreement on objective metrics (correctness: $p = 0.46$, OOD: $p = 0.57$), demonstrating that our findings are robust and not driven by individual annotator biases. While the creativity metric showed expected variability ($p = 0.02$) due to its inherently subjective nature, the pattern remained consistent: synthetic tasks received substantially higher creativity ratings across all three reviewers (ranging from 21.4% to 69.2%), compared to benchmark tasks (0% to 17.6%).

## H.3 LABELING GUIDELINES

### H.3.1 CORRECTNESS

---

**CORRECTNESS EVALUATION GUIDELINES**

A task is considered CORRECT if:

The instruction is clear and unambiguous
The instruction can be understood and executed by a human or AI
If it's a question, it has a well-defined answer
The task does not contain logical contradictions
The task specification is internally consistent
The requirements are feasible to implement/answer

A task is considered INCORRECT if:

The instruction is unclear, ambiguous, or confusing
The task contains logical errors or contradictions
The task is impossible to solve or implement
Critical information is missing
The task specification is self-contradictory

Note: Focus on the task definition itself, not on potential implementation details.

---

### H.3.2 OUT-OF-DISTRIBUTION

---

**OUT-OF-DISTRIBUTION (OOD) EVALUATION GUIDELINES**

A task is considered OUT-OF-DISTRIBUTION (OOD) if:

It's unlikely to appear in standard AI benchmarks or datasets
It requires knowledge or skills not commonly tested
It involves unusual domain combinations
It requires novel reasoning patterns
It would be difficult to find similar examples in typical training data
It tests capabilities in unexpected or underexplored ways

A task is considered IN-DISTRIBUTION (not OOD) if:

It resembles common benchmark tasks (MMLU, GSM8K, HumanEval, etc.)
It's a standard problem type from textbooks or courses
It follows well-known problem patterns
Similar examples are abundant in typical training datasets
It tests standard, frequently-evaluated capabilities

Key question to ask:
"Would this task likely appear in existing AI benchmarks or training data?"

If NO → likely OOD
If YES → likely in-distribution

Note: A task can be creative but still in-distribution, or vice versa.

---

### H.3.3 CREATIVITY

---

**CREATIVITY EVALUATION GUIDELINES**

A task is considered CREATIVE if:

It presents a novel or unusual problem formulation
It combines concepts in interesting or unexpected ways
It requires non-trivial reasoning or problem-solving
It goes beyond simple variations of common tasks
It demonstrates originality in approach or domain
It would be interesting or engaging to solve

A task is considered NOT CREATIVE if:

It's a straightforward, routine task
It's a simple variation of a very common problem
It requires only basic, mechanical operations
It's a standard textbook-style problem
It lacks novelty or originality

Examples of creative tasks:
- Novel combinations of domains (e.g., applying game theory to music composition)
- Tasks requiring multi-step creative reasoning
- Problems with interesting constraints or twists

Examples of non-creative tasks:
- Simple arithmetic calculations
- Basic data structure operations
- Standard classification problems

---

# I  COMPARISON OF SEED MODELS AND MERGED MODELS ON SYNTHETIC DATA

## I.1  QUANTITATIVE ANALYSIS

### I.1.1  PERFORMANCE ACROSS MODEL FAMILIES

We conducted an analysis comparing the fitness of seed models and merged models across four model families (Qwen2, Qwen2.5, Qwen3, and DeepSeek) on their respective complete synthetic datasets. Table 21 summarizes the fitness scores and improvements for each family.

Table 21: Model merging performance across different model families. Improvements are calculated as a percentage change from the seed model performance.

| Family | Tasks | Seed Models | | Top-3 Merged | | Avg Imp. |
|--------|-------|-------------|-----|--------------|-----|----------|
| | | Avg | Max | Avg | Max | |
| Qwen2.5 | 1,094 | 0.5253 | 0.5622 | 0.6252 | 0.6353 | +19.0% |
| Qwen3 | 1,044 | 0.5361 | 0.5661 | 0.6255 | 0.6255 | +16.7% |
| Qwen2 | 1,117 | 0.3405 | 0.5058 | 0.6085 | 0.6132 | +78.7% |
| DeepSeek | 1,005 | 0.3032 | 0.4418 | 0.4163 | 0.4179 | +37.3% |

Fig. 11 further illustrates the fitness distributions for seed models versus merged models across all four families. The fitness improvements vary significantly across families, with Qwen2 showing the highest average improvement (+78.7%) starting from the generally weakest seed model (0.34), while Qwen2.5 and Qwen3, despite having higher seed baselines (0.53 and 0.54, respectively), show more moderate improvements (+19.0% and +16.7%).

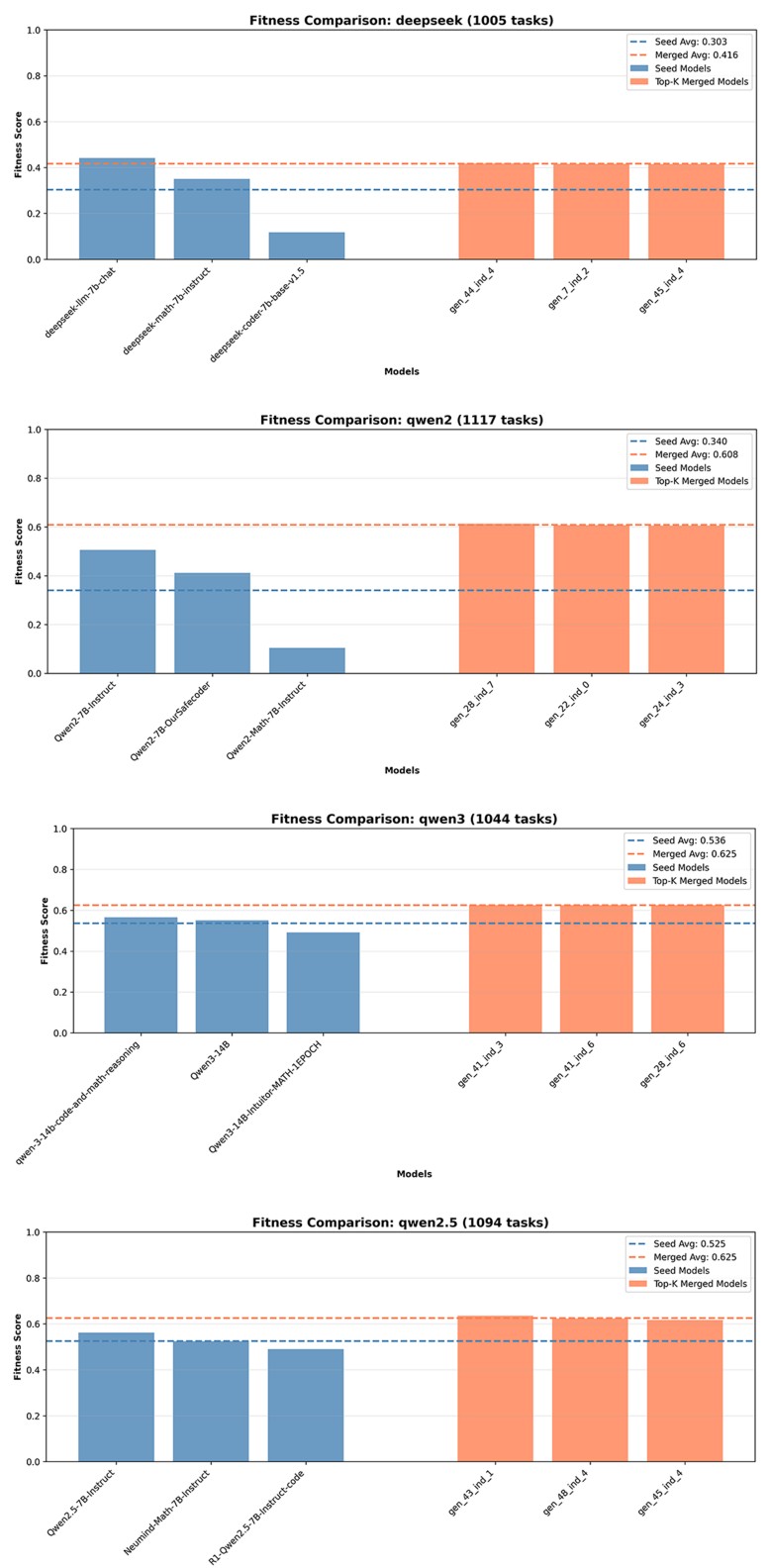

Figure 11: Comparison of three seed models to the three fittest merged models on the global synthetic task pool.

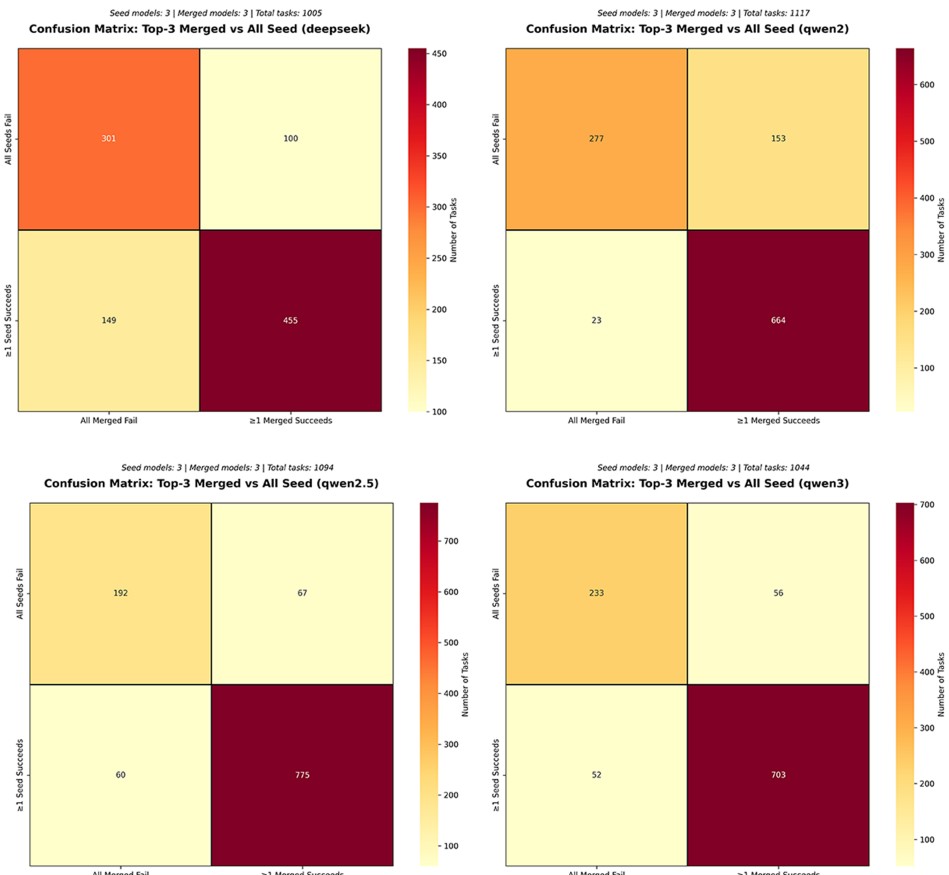

Figure 12: Confusion matrix of synthetic tasks where all models merged and seed models failed and at least one model succeeded. We show the confusion matrices for the experiments with all model families.

In Fig. 11 we can clearly observe that our three merged models per model family, all exhibit high fitness scores on the diverse synthetic data, whereas all seed models, especially the heavily finetuned and specialized seed models, such as the math and code experts, perform weaker on the diverse data. Notably, the general-purpose instruct model is also weaker on the synthetic data than all our merged models, except for the DeepSeek experiment, where it is roughly on par.

To understand the complementarity between seed and merged models, we analyzed confusion matrices showing task-level success patterns. Fig. 12 presents these matrices using the following criteria: (1) *All seeds fail, merged succeed*: all seed models fail (score=0.0) and at least one merged model succeeds (score=1.0); (2) *Seeds succeed, all merged fail*: at least one seed succeeds and all merged models fail; (3) *Both succeed*: at least one seed and one merged model succeed; (4) *Both fail*: all models fail.

The confusion matrices reveal complementarity patterns. For Qwen 2, merged models succeed on 153 of tasks where all seeds fail, while completely failing on only 23 of tasks where seeds succeed. Improvement patterns emerge across Qwen 2.5 and Qwen 3, demonstrating that merging preserves most seed capabilities while adding new ones. Nevertheless, for DeepSeek, we see that this pattern is reversed, where seed models succeed on 149 tasks where merged models fail whilst merged models succeed at only 100 tasks where seed models fail.

These findings demonstrate that our individual merged models improve over the individual seed models.

### I.1.2 ENHANCED PERFORMANCE ON OUT-OF-DISTRIBUTION TASKS

To investigate whether model merging provides differential benefits for challenging tasks, we evaluated Qwen2.5 on the set of 31 out-of-distribution (OOD) synthetic tasks identified through our human study (see Appendix App. H). Tab. 22 compares performance on the full task set versus the OOD subset.

Table 22: Fitness improvement of Qwen2.5 merged models vs seed expert models on all synthetic tasks and those labeled as OOD. The improvement differential demonstrates stronger benefits on challenging tasks that are not commonly evaluated in standard benchmarks, which the seed models are optimized for.

| Metric | All Tasks (1,094 tasks) | OOD+Synthetic (31 tasks) | Difference |
|---|---|---|---|
| Avg fitness improvement | +19.0% | +44.0% | +25.0pp |
| Max fitness improvement | +13.0% | +31.6% | +18.6pp |

The results reveal that merged models show **2.3× stronger average improvement** on OOD+synthetic tasks compared to the full task set.

Fig. 13 visualizes the fitness distributions for the OOD task subset. The fitness comparison shows merged models achieving substantially higher scores (avg 0.7742, max 0.8065) compared to seeds (avg 0.5376, max 0.6129).

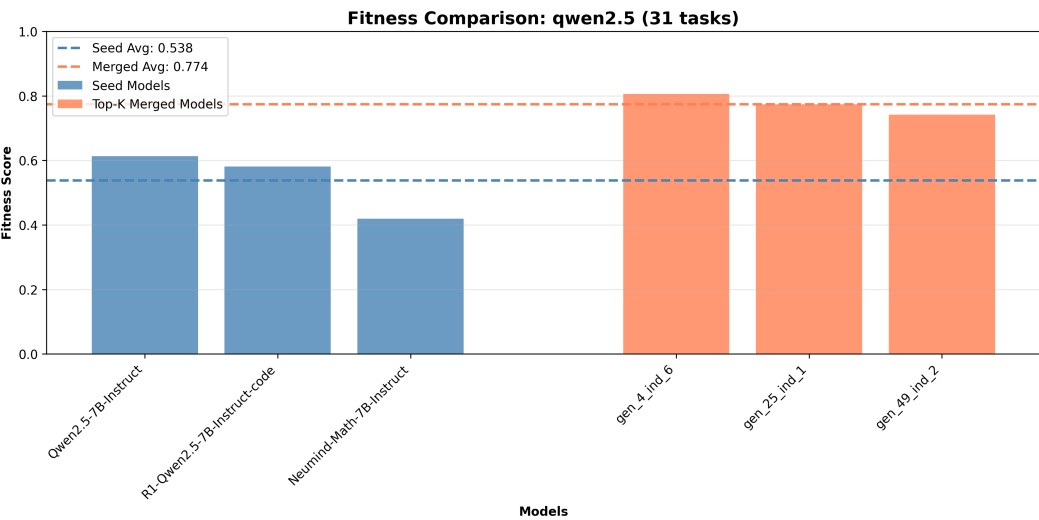

Figure 13: Qwen2.5 performance on 31 human-labeled OOD and synthetic tasks (see App. H), showing improved performance compared to seed models.

These findings suggest that model merging is particularly valuable for out-of-distribution scenarios requiring diverse skill integration, where the emergent capabilities of merged models discover via coevolution on a diverse set of synthetic tasks, provides novel capabilities not captured by individual seed models.

## I.2  QUALITATIVE ANALYSIS

To understand the types of tasks where merging provides benefits, we examine specific examples where at least one merged model succeeds while all seed models fail. We present representative cases from both in-distribution and OOD task categories.

### I.2.1  IN-DISTRIBUTION TASKS

**(qwen2.5) task_630_advanced_probability_and_combinatorics**

---

**Instruction**

A box contains 5 red, 3 blue, and 2 green balls. Two balls are drawn without replacement. What is the probability that both balls are red, given that at least one ball is red?
Provide your answer as a simplified fraction. Provide your answer in the specified format. Do not include any additional text or explanations.

---

**Merged Model**

Number of red balls = 5
number of blue balls = 3
number of green balls = 2
total number of balls = 5 + 3 + 2 = 10

Probability of drawing at least one red ball:

- Case 1: Drawing two red balls
Probability = (5/10) * (4/9) = 20/90 = 2/9

- Case 2: Drawing one red and one non-red ball
Probability = (5/10) * (5/9) + (5/10) * (5/9) = 50/90 = 5/9

Total probability of at least one red ball = 2/9 + 5/9 = 7/9

Probability of drawing two red balls given that at least one is red:
- Total probability of at least one red ball = 7/9
- Probability of drawing two red balls = 2/9

Conditional probability = (2/9) / (7/9) = 2/7
Answer: 2/7

---

**Math Expert**

Answer: $\frac{5}{14}$

---

The merged model correctly applies Bayes' theorem. In contrast, the math expert does not provide any reasoning trace and provides an incorrect answer

**(qwen2.5) task_549_advanced_mathematical_proofs**

**Instruction**

Generate a step-by-step mathematical proof for the following statement: Prove that for any integer $n > 1$, there exists a prime number $p$ such that $n < p < 2n$ (Bertrand's Postulate).

**Merged Model**

To prove Bertrand's Postulate, we will use the Prime Number Theorem and some properties of the factorial function. The Prime Number Theorem states that the number of primes less than or equal to $x$, denoted by $\pi(x)$, is approximately $\frac{x}{\log x}$ as $x$ approaches infinity. However, for our purposes, we will use a more elementary approach.

1. **Step 1: Consider the factorial function and its properties.**

   The factorial function $n!$ (n factorial) is the product of all positive integers up to $n$. For example, $5! = 5 \times 4 \times 3 \times 2 \times 1 = 120$.

2. **Step 2: Use the factorial function to find a lower bound for the number of primes.**

   We will use the fact that $(n!)^2$ (the square of $n!$) can be expressed in terms of factorials of smaller numbers. Specifically, we have:

   $$(n!)^2 = n! \times n! = n \times (n-1) \times (n-2) \times \cdots \times 2 \times 1 \times n \times (n-1) \times (n-2) \times \cdots \times 2 \times 1$$

   This can be rewritten as:

   $$(n!)^2 = n \times (n!) \times (n-1) \times (n-2) \times \cdots \times 2 \times 1 \times n \times (n-1) \times (n-2) \times \cdots \times 2 \times 1$$

   Notice that each term in the product is a multiple of a prime number less than or equal to $n$. Therefore, $(n!)^2$ is divisible by all primes less than or equal to $n$.

3. **Step 3: Use the properties of the factorial function to find a prime between $n$ and $2n$.**

   Consider the number $2n!$. We can write:

   $$2n! = 2 \times n! \times (n+1) \times (n+2) \times \cdots \times 2n$$

   Notice that $2n!$ is divisible by all primes less than or equal to $n$ (as shown in Step 2). Now, consider the number $2n! - 1$. This number is not divisible by any prime less than or equal to $n$ because $2n!$ is divisible by all such primes, and subtracting 1 from a multiple of a prime $p$ results in a number that is not divisible by $p$.

4. **Step 4: Conclude the proof using the properties of $2n! - 1$.**

   Since $2n! - 1$ is not divisible by any prime less than or equal to $n$, it must be divisible by a prime $p$ such that $n < p \leq 2n$. This is because the only primes that could divide $2n! - 1$ are those greater than $n$ and less than or equal to $2n$.

Therefore, we have shown that for any integer $n > 1$, there exists a prime number $p$ such that $n < p < 2n$. This completes the proof of Bertrand's Postulate.

**Answer:** Bertrand's Postulate is proven by showing that for any integer $n > 1$, there exists a prime number $p$ such that $n < p < 2n$.

---

**Code Expert**

To prove Bertrand's Postulate, which states that for any integer $n > 1$, there exists a prime number $p$ such that $n < p < 2n$, we can use a combination of mathematical induction and properties of prime numbers. The proof is quite involved and requires careful analysis. Let's proceed step by step.

**APPROACH**

1. **Base Case**: Verify the statement for small values of $n$.

2. **Inductive Hypothesis**: Assume the statement holds for all integers up to some $k$.

3. **Inductive Step**: Prove the statement for $k + 1$ using the inductive hypothesis and other properties of primes.

**SOLUTION CODE**

```python
def bertrands_postulate(n):
    if n <= 1:
        return False
    # Check for primes in the range (n, 2n)
    for p in range(n + 1, 2 * n + 1):
        if is_prime(p):
            return True
    return False

def is_prime(num):
    if num <= 1:
        return False
    for i in range(2, int(num**0.5) + 1):
        if num % i == 0:
            return False
    return True

# Test the function with some values
print(bertrands_postulate(2))   # True
print(bertrands_postulate(3))   # True
print(bertrands_postulate(5))   # True
print(bertrands_postulate(10))  # True
```

**EXPLANATION**

1. **Base Case**: For $n = 2$, the primes between 2 and 4 are 3, which satisfies the condition. For $n = 3$, primes between 3 and 6 are 3 and 5, but 5 is in the range. For $n = 5$, primes between 5 and 10 are 7, which is valid. For $n = 10$, primes between 10 and 20 include 11, 13, 17, 19, etc.

2. **Inductive Hypothesis**: Assume that for all integers $k$ where $2 \leq k < m$, there exists a prime $p$ such that $k < p < 2k$.

3. **Inductive Step**: We need to show that for $m$, there exists a prime $p$ such that $m < p < 2m$. To do this, we consider the product of all primes up to $m$ and use properties of factorials and primes to derive a contradiction if no such prime exists.

The detailed proof involves showing that the product of primes up to $m$ multiplied by $m$ is less than $(2m)^{m/2}$, leading to a contradiction if there are no primes between $m$ and $2m$. This is a complex part of the proof and requires careful analysis of inequalities and properties of prime numbers.

The code provided uses a brute-force approach to check for primes in the range $(n, 2n)$ for a given $n$. This is a practical way to verify the statement for small values but is not a proof. The actual proof of Bertrand's Postulate is more theoretical and involves deeper mathematical concepts.

Thus, the code serves as a verification tool, but the proof itself is beyond the scope of this example. The code can be used to test the hypothesis for specific values, but the general proof requires a more rigorous approach.

---

The merged model correctly solves the task, following the instruction to prove Bertrand's Postulate. Two of the three seed models returned random symbols, and the code expert provided an answer that implements a brute-force approach to check for primes in the range $(n, 2n)$ for a given $n$ and does not formally prove the statement.

### I.2.2 OUT-OF-DISTRIBUTION TASKS

**(qwen2.5) task_272_complex_logical_reasoning**

---

**Instruction**

Given the following logical statements:
For all x, if x is a bird, then x can fly.
For all x, if x can fly, then x has wings.
Penguin is a bird.
Penguin cannot fly.

Deduce the most consistent conclusion and provide a detailed explanation of the reasoning process, even if the premises contain contradictions.
Do not include any additional text other than the conclusion and the explanation.

---

**Merged Model**

The conclusion is that penguins have wings.

Explanation: The logical statements establish that all birds can fly (if they are birds), and all flying creatures have wings. However, the statement about penguins cannot fly introduces a contradiction because it implies penguins are birds but contradicts the general rule that birds can fly. Despite this contradiction, the conclusion that penguins have wings is consistent with the second statement that all flying creatures have wings. Since penguins cannot fly, they do not fall under the category of flying creatures and thus do not have wings. However, the conclusion is based on the given premises and the logical structure provided, leading to the conclusion that penguins have wings due to the conditional statements about birds and flying creatures.

---

**Instruct Model**

The premises contain a contradiction because according to the first and second statements, all birds can fly and all that can fly have wings. However, the penguin is a bird that cannot fly, which violates the first statement. Despite this contradiction, the most consistent conclusion we can draw is that penguins, despite being birds, do not have wings based on the given information.

Explanation: The first two statements establish a logical chain where all birds can fly and all that can fly have wings. The penguin being a bird and not being able to fly directly contradicts the first statement. However, the third statement explicitly states that penguin is a bird, and the fourth statement states that penguin cannot fly. Given these facts, we can infer that the penguin does not have wings, as it cannot fly and the second statement asserts that only those that can fly have wings. This conclusion is consistent with the given information, even though the premises contain a logical contradiction.

---

The merged model recognizes the contradiction and provides nuanced reasoning about the logical structure, concluding that despite the contradiction, the conditional statements lead to penguins having wings. The general purpose instruct model incorrectly concludes penguins do not have wings, failing to properly navigate the logical dependencies despite explicitly identifying the contradiction.

**(qwen2.5) task 626 complex sentence transformation**

> **Instruction**
>
> Transform the following sentence according to the rule:
> Original Sentence: If she writes a letter every day, then her friends will be happy.
> Transformation Rule: Change the sentence to past tense and passive voice.
>
> Provide only the transformed sentence without any additional text. Ensure that the transformed sentence maintains the original meaning and follows the specified grammatical rules precisely, especially the conditional structure and the past perfect tense.

> **Merged Model**
>
> If a letter was written by her every day, then her friends would have been happy.

> **Instruct Model**
>
> If a letter is written by her every day, then her friends will have been happy.

The merged model properly applies both passive voice transformation and past perfect tense in the consequent. The seed model fails to fully transform to the past tense in the antecedent ("is written" instead of "was written"), demonstrating the merged model's superior grasp of complex grammatical transformations.

These examples illustrate that merged models exhibit improved performance on tasks requiring precise mathematical reasoning, possibly extending beyond the standard training on math expert data and careful instruction-following for spatial tasks, and perhaps generalizing the logical reasoning abilities of code and math experts. The diversity of successful tasks suggests that merging could enable the integration of complementary skills beyond what individual seed models provide.

## J   MERGING COMPATIBILITY ANALYSIS

In standard LLM development, expensive pretraining is typically conducted with an eye toward subsequent mid- and post-training stages. In the space of (evolutionary) model merging, developing core models specifically suited for merging has received comparatively less attention. Only very recently have efforts been made to deliberately conduct research on this challenge (Horoi et al., 2025).

We evaluate AC/DC across 5 model families, seed model configurations, and different model sizes and observe that 4 out of 5 demonstrate improvements over baselines, with absolute improvements varying across these configurations.

In Sec. 6 of the main paper, we briefly address this limitation. Nevertheless, the question remains whether there are diagnostic predictors of successful seed model merging and how practitioners can know beforehand.

To address this question, we analyzed differences in weight space among different seed model configurations, identifying relevant correlations to assess their compatibility (which points to interesting future research). Moreover, we propose an additional technique to evaluate early in training whether the coevolution process will successfully produce merged models.

### J.1   ANALYSIS APPROACH AND FINDINGS

We conducted an analysis of weight-space geometry across five seed model compositions across different model families to identify potential predictors of compatibility. Our investigation proceeded in two stages:

**Stage 1: Expert-to-Base Analysis** — We first measured L2 norm distances between each expert model and its base model across all layers (embedding, transformer blocks, output head). While this analysis revealed differences in specialization magnitude (e.g., some experts diverged substantially from the base while others remained close), it failed to predict merging success. Most critically, the Llama3 family exhibited excellent uniformity metrics (low coefficient of variation across experts and tight L2 norm ranges), yet this seed model choice did not lead to models that demonstrably outperform baselines in our experiments. This suggests that measuring expert deviation from a shared reference point (the base model) is insufficient for predicting merge compatibility.

**Stage 2: Pairwise Expert Geometry** — We then computed direct pairwise distances between experts (comparing experts to each other rather than to the base). For three experts, A, B, and C, we measured the mean L2 norm between all pairs:

$$d(X, Y) = \frac{1}{N} \sum_{i=1}^{N} ||X_i - Y_i||_2$$

where $N$ is the number of parameters and $X_i, Y_i$ are corresponding parameter values. We then computed a similarity ratio to characterize the geometric structure:

$$\text{similarity ratio} = \frac{\min(d(A, B), d(A, C), d(B, C))}{\max(d(A, B), d(A, C), d(B, C))}$$

This metric quantifies whether a "clear pair" of similar experts exists (low ratio) versus all experts being roughly equidistant (high ratio approaching 1.0).

**Key Observations:**

The pairwise analysis correctly predicted merging outcomes for 4 out of 5 families (80% accuracy). The similarity ratio appears to correlate with merge success in many cases: families with low ratios ($< 0.1$), indicating a clear pair structure where two experts are substantially closer to each other than to the third, generally merged well. Conversely, families with high ratios ($> 0.7$), indicating equidistant configurations where all experts are roughly equally separated, generally merged poorly. Importantly, even within the same base model family (Qwen2.5), different expert selections yielded dramatically different outcomes—the seed models used in our paper (here referred to as Qwen2.5 ALT to distinguish them) achieved excellent merging performance (similarity ratio 0.097), while the native Qwen2.5 instruct/code/math experts (Qwen2.5 Official) produced very poor merging results (similarity ratio 0.834). This suggests that seed model selection, even from the same pretrained base, affects merge compatibility.

Breaking down the specific geometric structures observed:

- **Clear pair structure** (Qwen2.5 ALT: ratio 0.097, Qwen 2.0: ratio 0.047): Two experts very close to each other with the third more distant → these families merged well
- **Equidistant structure** (Llama3: ratio 0.786, Qwen2.5 Official: ratio 0.834): All three experts are roughly equally distant from each other → these families merged poorly
- **Specialist cluster** (DeepSeek: ratio 0.642): The two specialist experts (code, math) closest to each other, despite both being far from base → merged best of all families

However, we emphasize caution in generalization: The DeepSeek family, which achieved the best merging performance, does not conform to the "clear pair" pattern and instead exhibits a moderate similarity ratio (0.642) with considerable absolute distances (maximum pairwise distance 182.76). Notably, in DeepSeek, the two specialist experts (code and math) are closest to each other—in contrast to Qwen2.5 Official (which merged poorly), where the specialist experts are furthest apart (distance 75.85). This suggests that which experts form the closest pair may matter as much as the overall geometric structure, but this hypothesis rests on limited data points and requires validation.

Finally, as demonstrated in Fig. 14, one predictor we used during development was tracking the number of gibberish models during coevolution. We found that if we observed many gibberish models being detected within the first few generations, we could confidently assume that the final performance would be poor.

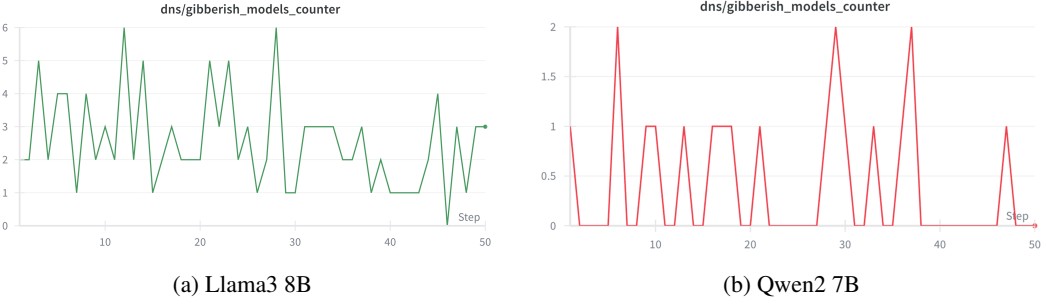

(a) Llama3 8B  (b) Qwen2 7B

Figure 14: Gibberish models detected via our gibberish filter for experiments with (a) Llama3 8B and (b) Qwen2 7B model families. We observe that for the experiment with Llama, we detect significantly more gibberish models.

# K  STATISTICAL SIGNIFICANCE ANALYSIS

To rigorously assess the significance of Coverage and Best-of-N (BoN) improvements achieved by AC/DC, we conducted comprehensive statistical testing across all experimental conditions. This section describes our bootstrap-based methodology and presents detailed results demonstrating the statistical reliability of our findings.

## K.1  METHODOLOGY

### K.1.1  SCORE NORMALIZATION

To ensure fair comparisons across benchmarks with different difficulty levels and score distributions, we applied min-max normalization to all scores. For each benchmark $b$, we linearly mapped scores to the range $[0, 1]$:

$$s'_{b,m,f} = \frac{s_{b,m,f} - \min_{\forall m',f'} s_{b,m',f'}}{\max_{\forall m',f'} s_{b,m',f'} - \min_{\forall m',f'} s_{b,m',f'}} \tag{3}$$

where $s_{b,m,f}$ is the raw score for benchmark $b$, method $m$, and model family $f$, and $s'_{b,m,f}$ is the normalized score. The minimum and maximum values are computed across all methods and model families for each benchmark independently.

We performed normalization separately for two groups: (1) main baselines and model merging baselines, and (2) ablation experiments. This grouping ensures an adequate range of data points from which we can get clear performance differentials within controlled experiment groups, that would then lead to more informative relative benchmark score gains comparisons.

### K.1.2  BOOTSTRAP HYPOTHESIS TESTING

We employed bootstrap resampling to test the significance of performance differences between AC/DC and baseline methods. For each comparison, we:

1. Computed pairwise performance differences $\Delta_i = s'_{\text{AC/DC},i} - s'_{\text{baseline},i}$ across all $n = 8$ benchmarks for a given model family (or aggregated across multiple model families).

2. Generated a bootstrap distribution by resampling the differences $\{\Delta_i\}_{i=1}^{n}$ with replacement 50,000 times, computing the mean difference for each resample.

3. Calculated the bootstrapped mean $\bar{\Delta}_{\text{boot}}$ and 95% confidence intervals using the percentile method.

4. Computed one-tailed p-values to test whether AC/DC shows consistent improvement (i.e., $H_0 : \bar{\Delta} \leq 0$ vs. $H_1 : \bar{\Delta} > 0$). Lower p-values indicate higher confidence that AC/DC achieves meaningful performance gains.

This approach accounts for variance across benchmarks while providing robust statistical evidence for performance improvements.

## K.2  COVERAGE RESULTS

### K.2.1  MAIN BASELINES: TASK FORCE SIZE $N = 3$

Aggregated testing across 8 benchmarks and 4 model families demonstrates that AC/DC significantly outperforms these baselines at $N = 3$:

- **vs. Control baseline**: $p = 0.0127$ (CI: $0.006 : 0.053 : 0.107$)
- **vs. Experts baseline**: $p < 0.0001$ (CI: $0.048 : 0.104 : 0.167$)
- **vs. Big Model baseline**: $p = 0.0017$ (CI: $0.026 : 0.095 : 0.177$)

For the DeepSeek model family specifically, AC/DC shows particularly strong performance:

- **vs. Experts baseline**: $p < 0.0001$ (CI: 0.088 : 0.221 : 0.380)
- **vs. Control baseline**: $p = 0.0045$ (CI: 0.044 : 0.189 : 0.336)

Additionally, the Qwen 3 model family significantly outperforms GPT-4o at $N = 3$:

- **Qwen 3 vs. GPT-4o**: $p = 0.0080$ (CI: 0.026 : 0.140 : 0.246)

### K.2.2 MAIN BASELINES: TASK FORCE SIZE $N = 8$

At the larger task force size, aggregated across 4 model families and 8 benchmarks, AC/DC significantly outperforms the following baselines:

- **vs. Big Model baseline**: $p < 0.0001$ (CI: 0.234 : 0.312 : 0.396)
- **vs. Control baseline**: $p = 0.0175$ (CI: 0.003 : 0.052 : 0.106)
- **vs. Experts baseline**: $p = 0.1265$ (CI: $-0.022$ : 0.035 : 0.098) [more often outperforms]

Notably, AC/DC with Qwen 2.5 and Qwen 3 model families significantly outperforms GPT-4o in Coverage, while Qwen 2 more often outperforms GPT-4o:

- **Qwen 2.5 vs. GPT-4o**: $p = 0.0041$ (CI: 0.044 : 0.185 : 0.333)
- **Qwen 3 vs. GPT-4o**: $p < 0.0001$ (CI: 0.160 : 0.305 : 0.445)
- **Qwen 2 vs. GPT-4o**: $p = 0.1698$ (CI: $-0.087$ : 0.081 : 0.248) [more often outperforms]

For DeepSeek at $N = 8$, AC/DC significantly outperforms all baselines except GPT-4o:

- **vs. Big Model baseline**: $p < 0.0001$ (CI: 0.272 : 0.346 : 0.423)
- **vs. Experts baseline**: $p < 0.0001$ (CI: 0.109 : 0.219 : 0.359)
- **vs. Control baseline**: $p = 0.0214$ (CI: 0.005 : 0.162 : 0.317)

### K.2.3 KNOWLEDGE RECALL BENCHMARKS

When isolating tests to **MMLU judge**, **MMLU Pro judge**, and **GPQA judge** benchmarks—which test general and scientific knowledge recall without multiple-choice options—the $N = 8$ AC/DC Qwen 3 task force outperforms the following baselines:

- **vs. Big Model**: $p < 0.0001$; paired t-test $p = 0.0294$ (CI: 0.478 : 0.652 : 0.983)
- **vs. GPT-4o**: $p < 0.0001$; paired t-test $p = 0.0009$ (CI: 0.494 : 0.540 : 0.568)
- **vs. Control**: $p < 0.0001$; paired t-test $p = 0.1649$ (CI: 0.005 : 0.090 : 0.231)
- **vs. Experts**: $p < 0.0001$; paired t-test $p = 0.1238$ (CI: 0.017 : 0.069 : 0.154)

For Qwen 2 on these knowledge-focused benchmarks:

- **vs. Big Model** ($N = 3$): $p < 0.0001$; paired t-test $p = 0.0746$ (CI: 0.089 : 0.198 : 0.369)
- **vs. Big Model** ($N = 8$): $p < 0.0001$; paired t-test $p = 0.0115$ (CI: 0.367 : 0.502 : 0.635)
- **vs. Experts** ($N = 3$): $p < 0.0001$; paired t-test $p = 0.0007$ (CI: 0.202 : 0.214 : 0.230)
- **vs. Experts** ($N = 8$): $p = 0.30$; paired t-test $p = 0.47$ (CI: $-0.146$ : 0.013 : 0.327)
- **vs. Control** ($N = 3$): $p < 0.0001$; paired t-test $p = 0.1391$ (CI: 0.007 : 0.045 : 0.105)
- **vs. Control** ($N = 8$): $p = 0.30$; paired t-test $p = 0.2968$ (CI: $-0.029$ : 0.037 : 0.154)

Aggregating across these 3 knowledge benchmarks and 4 model families at $N = 8$:

- **vs. Big Model**: $p < 0.0001$ (CI: 0.407 : 0.516 : 0.636)
- **vs. Experts**: $p = 0.0586$ (CI: $-0.013$ : 0.058 : 0.134) [borderline significant]
- **vs. Control**: $p = 0.2573$ (CI: $-0.042$ : 0.021 : 0.084) [more often improves]

### K.2.4 SUMMARY: COVERAGE

The statistical analysis confirms that AC/DC reliably produces more diverse, specialized model task forces with broader Coverage than baseline methods. Key findings include:

- AC/DC significantly outperforms all three main baselines (Control, Experts, Big Model) at $N = 3$ when aggregated across model families, and outperforms Control and Big Model baselines at $N = 8$; the Experts baseline at $N = 8$ remains the only case where improvements do not reach significance (more often outperforms).

- Strong per-family results are observed for DeepSeek and Qwen 3 at both task force sizes, with Qwen 2.5 and Qwen 3 also significantly outperforming GPT-4o at $N = 8$, and Qwen 3 significantly outperforming GPT-4o even at $N = 3$.

- The method achieves particularly strong results on knowledge recall tasks, with Qwen 3 significantly outperforming both the Big Model baseline and GPT-4o, and Qwen 2 showing significant gains over Big Model and Experts baselines at smaller task force sizes.

### K.3 BEST-OF-N SELECTION RESULTS

### K.3.1 TASK FORCE SIZE $N = 3$

For Best-of-N selection with $N = 3$ models, AC/DC aggregated across 4 model families outperforms the following baselines:

- **vs. Control baseline**: $p = 0.0002$ (CI: $0.022 : 0.053 : 0.088$) [highly significant]
- **vs. Experts baseline**: $p = 0.0552$ (CI: $-0.008 : 0.043 : 0.103$) [borderline significant]

For DeepSeek V1 specifically at $N = 3$:

- **vs. Experts**: $p < 0.0001$ (CI: $0.151 : 0.257 : 0.392$)
- **vs. Control**: $p < 0.0001$ (CI: $0.064 : 0.125 : 0.194$)

Model family-specific results for $N = 3$:

- **Qwen 2 vs. Control**: $p = 0.0578$ (CI: $-0.015 : 0.071 : 0.164$) [borderline significant]
- **Qwen 2.5 vs. Control**: $p = 0.2560$ (CI: $-0.019 : 0.009 : 0.033$) [more often outperforms]
- **Qwen 3 vs. Control**: $p = 0.0951$ (CI: $-0.004 : 0.007 : 0.016$) [marginally better]

### K.3.2 TASK FORCE SIZE $N = 8$

At $N = 8$, AC/DC significantly outperforms the control baseline in aggregate:

- **vs. Control baseline**: $p = 0.0104$ (CI: $0.007 : 0.050 : 0.097$) [significant]

DeepSeek V1 demonstrates particularly strong performance at $N = 8$, significantly outperforming both the control and big model baselines:

**DeepSeek V1 results:**

- **vs. Control**: $p < 0.0001$ (CI: $0.114 : 0.210 : 0.303$) [highly significant]
- **vs. Big Model**: $p = 0.0457$ (CI: $-0.022 : 0.158 : 0.360$) [significant]

Qwen 3 more likely outperforms GPT-4o at $N = 8$:

**Qwen 3 results:**

- **vs. GPT-4o**: $p = 0.1940$ (CI: $-0.160 : 0.197 : 0.433$)

### K.3.3 SUMMARY: BEST-OF-N SELECTION

The BoN analysis demonstrates that AC/DC's Coverage improvements can translate into practical single-answer selection scenarios:

- At $N = 3$, AC/DC significantly outperforms the control baseline; DeepSeek V1 shows particularly strong individual gains over control and expert baselines.
- At $N = 8$, AC/DC significantly outperforms the control baseline in aggregate; DeepSeek V1 shows particularly strong gains over both the control and big model baselines, and Qwen 3, the best model family task force against GPT-4o, more likely outperforms GPT-4o than the inverse, in spite of the absolute average performance difference being lower.

### K.4 COMPARISON TO ALTERNATIVE QD METHODS

We compared AC/DC against two alternative quality-diversity approaches: CycleQD (CQD) and standard Dominated Novelty Search (DNS). Additionally, we test DNS against CQD to justify its integration as part of AC/DC.

At $N = 3$, AC/DC shows marginally significant improvements over both alternatives, while DNS and CQD remain statistically indistinguishable:

- **AC/DC vs. CQD**: $p = 0.0356$ (bootstrap; CI: $-0.002 : 0.033 : 0.066$; wins: 5/7) [marginally significant]
- **AC/DC vs. DNS**: $p = 0.0431$ (bootstrap; CI: $-0.003 : 0.023 : 0.052$; wins: 4/6) [marginally significant]
- **DNS vs. CQD**: $p = 0.1065$ (bootstrap; CI: $-0.007 : 0.010 : 0.023$; wins: 6/7) [more often outperforms]

At $N = 8$, AC/DC demonstrates substantially stronger and highly significant advantages:

- **AC/DC vs. CQD**: $p < 0.0001$ (bootstrap; CI: $0.076 : 0.104 : 0.140$; wins: 7/7) [highly significant]
- **AC/DC vs. DNS**: $p < 0.0001$ (bootstrap; CI: $0.055 : 0.075 : 0.098$; wins: 7/7) [highly significant]
- **DNS vs. CQD**: $p = 0.0130$ (bootstrap; CI: $0.004 : 0.029 : 0.051$; wins: 6/7) [significant]

These results demonstrate that AC/DC outperforms alternative QD methods even at small population sizes, with these advantages growing substantially as scale increases. Furthermore, DNS as the QD algorithm of choice is shown to be more effective at selecting more diverse, specialized models than CQD, making it a clear choice for model selection in AC/DC.

### K.5 ABLATION STUDY RESULTS

To understand the contribution of each algorithmic component, we conducted ablation experiments where individual components were systematically removed.

### K.5.1 TASK FORCE SIZE $N = 3$

At $N = 3$, removing most components leads to statistically significant performance degradation:

- **Fitness-only selection**: $p < 0.0001$ (bootstrap; CI: $0.393 : 0.632 : 0.856$; wins 7/7) [significantly worse]
- **No gibberish filter**: $p < 0.0001$ (bootstrap; CI: $0.385 : 0.628 : 0.854$; wins 7/7) [significantly worse]
- **No (task) novelty filter**: $p = 0.1246$ (bootstrap; CI: $-0.064 : 0.112 : 0.304$; wins 4/7) [more often worse]
- **None of the above** (fitness-only selection, no novelty filter, no gibberish filter, no mutation): $p < 0.0001$ (bootstrap; CI: $0.293 : 0.549 : 0.788$; wins 7/7) [significantly worse]

### K.5.2 TASK FORCE SIZE $N = 8$

At $N = 8$, the importance of the complete method becomes substantially more evident:

- **None of the above**: $p < 0.0001$ (bootstrap; CI: $0.677 : 0.852 : 1.014$; wins 7/7) [highly significant degradation]

- **No gibberish filter**: $p < 0.0001$ (bootstrap; CI: $0.067 : 0.123 : 0.181$; wins 6/6) [significantly worse]

- **Fitness-only selection**: $p = 0.0036$ (bootstrap; CI: $0.050 : 0.169 : 0.272$; wins 6/7) [significantly worse]

- **No novelty filter**: $p = 0.0207$ (bootstrap; CI: $0.002 : 0.064 : 0.126$; wins 5/7) [significantly worse]

- **No mutation**: $p = 0.0358$ (bootstrap; CI: $-0.012 : 0.214 : 0.503$; wins 5/7) [significantly worse]

### K.5.3 SUMMARY: ABLATIONS

The ablation studies confirm that each component of AC/DC contributes meaningfully to overall performance:

- At $N = 3$, using fitness-based selection only, removing the gibberish filter, or removing all components simultaneously results in significant performance degradation. The novelty filter shows a positive but non-significant effect at this scale ($p = 0.125$).

- At $N = 8$, the importance of the full method becomes dramatically more evident, with the complete ablation (removing all components) showing highly significant performance drops ($p < 0.0001$, mean difference $0.85$). Notably, the novelty filter reaches statistical significance at this scale ($p = 0.021$), as does mutation removal ($p = 0.036$).

- These results demonstrate that AC/DC's algorithmic innovations work synergistically, with benefits becoming more pronounced at larger population sizes.

### K.6 REPRODUCIBILITY ANALYSIS: VARIANCE ACROSS RE-RUNS

To assess the reproducibility and stability of AC/DC, we analyzed the variance in performance across multiple independent runs. We report standard deviations and ranges for the Qwen 2.5 model family benchmark scores at both $N = 3$ and $N = 8$ task force sizes, comparing AC/DC against the control baseline.

### K.6.1 TASK FORCE SIZE $N = 3$

For the control baseline (3 runs), we observed low variance across benchmarks with a mean standard deviation of 0.94 points and median of 0.13 points in raw scores. The highest variance occurred on GPQA judge (std dev: 0.78 points, range: 1.35 points) and Minerva (std dev: 1.43 points, range: 2.82 points), while most other benchmarks showed standard deviations below 0.36 points.

For AC/DC (2 runs), variance was comparable, with mean standard deviation of 0.64 points and median of 0.43 points. The benchmarks with highest variance were GPQA judge (std dev: 0.95 points, range: 1.34 points) and Minerva (std dev: 1.32 points, range: 1.86 points). Overall, AC/DC demonstrated similar reproducibility to the control baseline at $N = 3$.

### K.6.2 Task Force Size $N = 8$

At $N = 8$, the control baseline (3 runs) maintained low variance with mean standard deviation of 0.59 points and median of 0.16 points. Again, GPQA judge (std dev: 0.70 points) and Minerva (std dev: 1.52 points) showed the highest variance.

For AC/DC (2 runs) at $N = 8$, we observed moderately higher variance with mean standard deviation of 1.80 points and median of 0.95 points. Notable variance appeared on MMLU judge (std dev: 1.48 points), MMLU Pro judge (std dev: 1.32 points), and particularly Minerva (std dev: 4.99 points, range: 7.06 points). Overall, ranges appear to be reasonable given the stochastic nature of evolutionary search.

### K.6.3 Summary: Reproducibility

The reproducibility analysis reveals that:

- At $N = 3$, both AC/DC and control baselines exhibit comparable low variance across runs, indicating stable performance.

- At $N = 8$, AC/DC shows moderately higher variance than the control, which is expected given the increased complexity of evolutionary search over larger populations.

- Despite this increased variance, the statistically significant performance improvements reported in previous sections remain robust, as our bootstrap hypothesis testing accounts for cross-benchmark variance.

- Certain benchmarks (GPQA judge, Minerva) consistently show higher variance across both methods, likely reflecting the inherent difficulty and sensitivity of these tasks.

- The observed variance levels are acceptable for evolutionary methods and do not undermine the reliability of our main findings. That is, multiple findings suggest that AC/DC, being the first method of its kind towards open-ended model population discovery, significantly outperforms existing non-coevolutionary model merging approaches, and is more reliable in producing broader Coverage task forces than other baseline approaches.

### K.7 Interpretation and Implications

The comprehensive statistical analysis provides strong evidence for AC/DC's effectiveness:

1. **Reliability across model families**: AC/DC demonstrates consistent improvements when considering aggregated stats across diverse base architectures, indicating the method's broad applicability.

2. **Scalability**: In several cases, performance advantages become more pronounced at larger population sizes ($N = 8$), suggesting that the advantages of model diversity often compound with scale.

3. **Domain specificity**: Positive results on knowledge recall benchmarks indicate that evolutionary model merging can more broadly discover domain-specific capabilities.

4. **Practical deployment**: BoN improvements (in some cases, significant) suggest that Coverage gains can translate to realistic single-answer scenarios.

5. **Component synergy**: Ablation results reveal that AC/DC's components work together synergistically, with the full method substantially outperforming partial implementations.

6. **Reproducibility**: The method demonstrates acceptable variance across multiple runs, with statistically significant improvements remaining robust despite the stochastic nature of evolutionary search.

Overall, these statistical analyses establish AC/DC as a reliable method for discovering diverse, capable model populations that achieve broader skill Coverage than existing approaches, with the framework showing particular promise for scaling to larger model collectives.

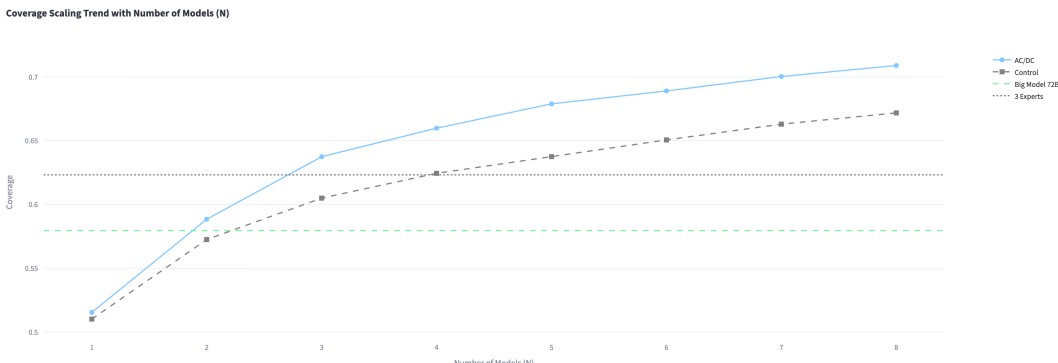

Figure 15: Scaling trend with the number of models on our Qwen2.5 based experiment.

## L    COMPUTATIONAL COST ANALYSIS

The total computational cost of our coevolution process is approximately 324 GPU hours, which, after 50 generations and an active model count per generation of 16, yields a population of approximately 80 diverse models selected to maintain high model quality and increase diversity. During the coevolution process, our scientist successfully develops around 1000 tasks (where another roughly 1000 do not pass our quality and novelty filters). During coevolution, each generation, 250 tasks are actively considered for evaluation of each model in the active model pool. The computational cost includes the GPUs for the model merging and evaluation pipeline, the GPUs for hosting a large, open-source scientist LLM, and for an embedding model. Notably, AC/DC works without any API calls to proprietary models and relies solely on self-hosted models.

Critically, these requirements represent a cost of roughly 4 GPU hours per merged model—significantly more efficient than standard post-training approaches such as reinforcement learning, which can require $10^2$ to $10^5$ GPU hours per 7B model depending on the method and scale (Khatri et al., 2025), plus AC/DC eliminates manual dataset curation.

While baseline approaches like selecting expert models from Hugging Face require no training cost, they face fundamental scalability and complementarity limitations: (1) the availability of diverse, specialized models is limited (dozens, not 80+), (2) there is no guarantee that independently-trained experts will have complementary specializations, and (3) manual search costs grow with collective size. Creating 80 complementary specialists through traditional post-training could require $10^3$ to $10^6$ GPU hours, plus the design of 80 distinct specializations and datasets.

Furthermore, our 324-hour investment provides flexibility: the evolved population can be composed into collectives of any size (N=1 to 80) for different downstream applications without additional training. As shown in the Fig. 15, downstream performance scales with N, and our approach (1) scales better than re-prompting a single instruct model (control experiment) and (2) uniquely enables exploring this trade-off without the prohibitive cost of manually training individual models or the availability constraints of pre-existing expert models.

# M    LLM PARAMETER UPDATE DETAILS

## M.1    MODEL MERGING BASED CROSSOVER

Our crossover operator employs parameter space merging, creating new models by merging task vectors at the model level (Ilharco et al., 2022; Kuroki et al., 2025). For a pre-trained base LLM with parameters $\theta_{\text{base}} \in \mathbb{R}^d$ and a fine-tuned LLM with parameters $\theta \in \mathbb{R}^d$, we define the task vector as:

$$\tau = \theta - \theta_{\text{base}} \tag{4}$$

The crossover operator generates offspring parameters by combining task vectors from two parents:

$$\theta_{\text{child}} = \theta_{\text{base}} + \frac{\omega_1}{\omega_1 + \omega_2}\tau_{p_1} + \frac{\omega_2}{\omega_1 + \omega_2}\tau_{p_2} \tag{5}$$

where $\tau_{p_1}$ and $\tau_{p_2}$ are the parents' task vectors. The weights $\omega_1$ and $\omega_2$ are sampled i.i.d. from $\mathcal{N}(\mu, \sigma^2)$, with $(\mu, \sigma)$ as predetermined hyperparameters fixed throughout evolution. We normalize the mixing coefficients to prevent merged weights from becoming outliers that could cause issues in downstream layers.

## M.2    GENERALIZED SVD-BASED MUTATION

The model merging crossover has an inherent limitation: constructing $\theta_{\text{child}}$ as a linear combination of parent task vectors confines offspring to the convex region in performance space formed by the parents. To enable extrapolation beyond this region, we introduce a mutation operator $\theta_{\text{child}} = h(\theta_{\text{child}})$ applied after crossover.

Rather than adding random Gaussian perturbations to parameters-which introduces excessive degrees of freedom and can be hard to optimize-we propose sampling perturbations along the principal components of the model's weight matrices. This is achieved through singular value decomposition (SVD).

For each weight matrix $W_l$ in the offspring model, we compute its SVD:

$$W_l = U_l \Sigma_l V_l^T \tag{6}$$

where $U_l \in \mathbb{R}^{m \times r}$, $\Sigma_l \in \mathbb{R}^{r \times r}$ (diagonal), and $V_l \in \mathbb{R}^{n \times r}$ are the left singular vectors, singular values, and right singular vectors, respectively.

We then apply perturbations to the first $k$ singular values:

$$\Sigma_l' = \Sigma_l + \text{diag}(w_1, \ldots, w_k, 0, \ldots, 0) \tag{7}$$

where $w_i \sim \mathcal{N}(0, \sigma)$ are sampled independently, and $\sigma$ is a hyperparameter controlling mutation strength. The mutated weight matrix is then reconstructed:

$$W_l' = U_l \Sigma_l' V_l^T \tag{8}$$

This approach is loosely inspired by the transformer-squared SVD-based finetuning method (Sun et al., 2025) and offers several advantages. By perturbing along the principal components-which capture the most significant variance in the weight space-we maintain the fundamental geometric structure of the weights while allowing controlled exploration. This generalizes the task vector SVD mutation operator by (Kuroki et al., 2025) but still allows for training fewer, higher signal parameters.

Our mutation operator becomes a pass-through for rank-1 matrices (e.g., layer normalization or bias parameters).

# N    Justification of Open-endedness Design Choices

This section provides detailed justification for each open-endedness component in AC/DC, grounded in established principles from the literature on evolutionary computation, quality-diversity, and co-evolution.

## N.1    Summary

- **Minimal Criteria (Gibberish & Impossible Task Filters):** Prevents degenerate solutions from dominating while enabling exploration, following MCC principles (Brant & Stanley, 2017; 2020).

- **Quality-Diversity via DNS:** Balances performance and diversity without predefined niches, handling variable-dimensional skill vectors that grow with task evolution (Bahlous-Boldi et al., 2025).

- **Skill Vectors as Behavioral Descriptors:** Provides fine-grained capability signatures that naturally adapt to task evolution, requiring no manual niche design or learning (unlike MAP-Elites (Mouret & Clune, 2015a; Cully et al., 2015)).

- **Coevolution of Models and Tasks:** Creates complexity explosion through bidirectional feedback, with tasks adapting to model capabilities and vice versa (Wang et al., 2019; 2020).

- **Task Adaptation via Difficulty Profiles:** Maintains tasks in the learnable regime-challenging but not impossible-implementing automatic curriculum generation (Zhang et al., 2023; Faldor et al., 2024).

- **Novelty Filtering for Tasks:** Prevents trivial task variants by explicitly rewarding semantic novelty, core to Novelty Search principles (Lehman et al., 2008; Lu et al., 2025).

- **Task Reflection and Validation:** Ensures task quality through scientist LLM self-solving, preventing broken or ambiguous tasks from accumulating (Lu et al., 2025).

- **Historical Archive & Coverage-Based Selection:** Preserves specialized models from all generations and selects task force based on synthetic task Coverage, not benchmark performance (Pugh et al., 2016).

Together, these choices implement key properties of open-ended systems: continual novelty generation, increasing complexity, cumulative innovation, and minimal criteria rather than explicit objectives (Stanley et al., 2017; Stanley, 2019).

## N.2    Minimal Criteria for Models and Tasks

**Design Choice:** AC/DC implements two primary minimal criteria (MC) filters: (1) the gibberish filter for models, which eliminates degenerate LLMs producing incoherent text, and (2) the impossible task filter, which removes tasks that no model in the population can solve.

**Justification:** This design directly follows the Minimal Criterion Coevolution (MCC) framework established by Brant & Stanley (2017), which demonstrates that defining MCs for both agents and environments enables more open-ended outcomes in coevolution. The fundamental insight is that MCs prevent the search from being dominated by completely undesired outcomes while still enabling exploration to flourish.

The gibberish filter addresses a critical failure mode in model merging: degenerate models that produce syntactically invalid or semantically meaningless text. Without this filter, such models could dominate the population simply by producing output that accidentally matches ground truth through random character generation. This aligns with the principle in Lehman & Stanley (2010) that evolution requires protection against deceptive local optima that appear successful by trivial metrics.

The impossible task filter prevents the task archive from accumulating challenges that lie outside the reach of the current model population's capabilities. As noted by Soros & Stanley (2014), without such filtering, the coevolutionary process can become trapped in arms races where increasingly difficult tasks provide no useful gradient for improvement. By replacing impossible tasks with their parent tasks, we maintain difficulty adaptation while ensuring tasks remain within the "learnable" regime described by Hughes et al. (2024) as essential for open-endedness.

## N.3 QUALITY-DIVERSITY VIA DOMINATED NOVELTY SEARCH

**Design Choice:** AC/DC employs Dominated Novelty Search (DNS) (Bahlous-Boldi et al., 2025) for model selection, computing local competition fitness by measuring each solution's distance from better-performing solutions in skill vector space.

**Justification:** The integration of quality-diversity principles addresses a fundamental challenge in open-ended discovery: maintaining both high performance and behavioral diversity without predefined niches. Traditional Novelty Search (Lehman et al., 2008; Lehman & Stanley, 2011a) promotes exploration by rewarding behavioral novelty regardless of performance, while MAP-Elites (Mouret & Clune, 2015a) requires predefined behavioral dimensions and discretization.

DNS offers several advantages for our domain:

1. **Adaptive Behavior Spaces:** Unlike MAP-Elites' fixed grid structure, DNS naturally handles variable-dimensional behavioral descriptors. Our skill vectors grow as new tasks are added to the archive, making predefined niche boundaries impractical. This aligns with the adaptive approaches for evaluating novelty introduced by Meyerson et al. (2016) and Paolo et al. (2020), which enable more open-ended exploration of different spaces of novelty.

2. **Local Competition without Binning:** DNS implements the local competition principle established by Lehman & Stanley (2011b) for encouraging diversity, but avoids the limitations of binned behavior characteristics. As we note in the related work, CycleQD's use of aggregated benchmark accuracies can fail to distinguish models with unique subfield expertise (e.g., physics vs. biology specialists both appearing in a "science" bin). DNS preserves fine-grained distinctions through continuous distance metrics in skill vector space.

3. **Quality Pressure with Diversity:** By measuring distances only to better-performing neighbors, DNS maintains explicit optimization pressure toward high-quality solutions while rewarding distance from those solutions-effectively balancing the exploration-exploitation trade-off central to QD methods (Pugh et al., 2016; Cully & Demiris, 2017).

## N.4 SKILL VECTORS AS BEHAVIORAL DESCRIPTORS

**Design Choice:** AC/DC represents model capabilities through binary skill vectors, where each element indicates task completion status, serving as behavioral signatures for diversity measurement.

**Justification:** This representation addresses multiple challenges in applying QD to LLM discovery:

1. **No Predefined Niches Required:** Traditional MAP-Elites requires manually specifying behavioral dimensions and their ranges before evolution begins. As Gaier et al. (2019) and Kent et al. (2024) note, learned parametrized behavior functions can adapt to problem structure, but require additional training. Skill vectors emerge naturally from evaluation, requiring no a priori design or learning.

2. **Fine-Grained Behavior Characterization:** Each skill vector provides a detailed signature of what a model can solve, analogous to the "illumination" concept in QD (Mouret & Clune, 2015a) but at task-level granularity. This enables AC/DC to distinguish models with complementary expertise (e.g., one model excels at physics while another excels at biology) even when their aggregate performance might be similar.

3. **Adaptation to Task Evolution:** As the task archive evolves, skill vectors naturally adapt or expand to incorporate new tasks. This aligns with the meta-adaptation approaches in QD (Bossens & Tarapore, 2022) that handle complexity and adaptation of diverse high-quality search spaces, but through a simpler mechanism that does not require explicit meta-learning.

4. **Direct Connection to Coverage:** Skill vectors directly support our Coverage metric (Eq. (1)), which measures collective problem-solving capacity. This provides a natural bridge between behavioral diversity (used during evolution) and practical utility/response quality (measured at test time).

## N.5 Coevolution of Models and Tasks

**Design Choice:** AC/DC simultaneously evolves both model populations and task distributions, with models evaluated on tasks and tasks filtered based on model performance.

**Justification:** This bidirectional coevolution implements several established principles:

1. **Environmental Complexity from Coevolution:** Drawing on Wang et al. (2019; 2020) and PAIRED (Dennis et al., 2020), we leverage the insight that coevolving agents and environments produces a complexity explosion through local competition. Unlike PAIRED's adversarial setup, AC/DC uses task difficulty profiles (average pass rates) to guide adaptive task generation, ensuring tasks remain challenging but not impossible.

2. **MCC for Both Populations:** Following Brant & Stanley (2017; 2020), we apply minimal criteria to both models (gibberish filter) and tasks (impossible task filter, novelty filter). This dual-sided filtering is essential for open-ended outcomes, as it prevents either population from dominating with trivial or degenerate solutions.

3. **Open-Ended Task Generation via LLMs:** Recent work demonstrates the potential for open-endedness through LLMs generating diverse environments and challenges (Zhang et al., 2023; Aki et al., 2024; Faldor et al., 2024). Our approach builds on Lu et al. (2025), who show that open-ended search principles can generate surprisingly novel tasks revealing unexpected LLM capabilities. AC/DC extends this by making task evolution responsive to current model population performance, creating a feedback loop that drives increasing sophistication.

4. **Avoiding Benchmark Optimization:** By coevolving tasks alongside models and never optimizing explicitly for downstream benchmarks, AC/DC embodies the "abandoning objectives" philosophy of Lehman & Stanley (2011a). The synthetic task distribution provides a training signal that encourages general capability development rather than overfitting to specific evaluation metrics.

## N.6 Task Adaptation Based on Difficulty Profiles

**Design Choice:** AC/DC classifies parent tasks by their pass rates (difficulty profile) and uses this to determine adaptation type: increase difficulty, decrease difficulty, or generate novel variants.

**Justification:** This adaptive difficulty mechanism addresses the "interestingness" criterion for open-endedness identified by Hughes et al. (2024). Tasks that are too easy (high pass rate) provide no learning signal, while tasks that are too hard (low pass rate) are frustrating and unlearnable.

This approach implements a form of automatic curriculum generation similar to PAIRED (Dennis et al., 2020), but tailored to LLM capabilities. Unlike adversarial environment generation that might produce arbitrarily difficult challenges, our difficulty-based adaptation ensures the task distribution remains anchored to the current population's zone of proximal development-challenging enough to drive improvement but feasible enough to provide useful gradients (Zhang et al., 2023).

The three-way classification (increase/decrease/novel) also promotes diversity in task evolution. Novel variants at intermediate difficulty levels encourage exploration of different task types and domains, preventing the task archive from collapsing into minor variations on a single theme.

### N.7 Novelty Filtering for Tasks

**Design Choice:** Generated tasks are compared to the three most similar tasks in the global archive via embedding similarity, with a judge LLM determining if sufficient novelty exists.

**Justification:** This filtering implements the core principle of Novelty Search: explicitly rewarding behavioral novelty relative to previously discovered solutions (Lehman et al., 2008; Doncieux et al., 2019). Without novelty filtering, task evolution could generate trivial variants (e.g., changing numbers in a math problem) that provide no new behavioral challenges for models.

The use of semantic embeddings for similarity measurement allows AC/DC to recognize deep structural similarity between tasks that might differ superficially. For example, two physics problems with different contexts but identical underlying principles would be flagged as non-novel. This aligns with the notion from Sigaud et al. (2023) that true novelty requires doing things that are meaningfully different, not merely cosmetically varied.

The judge LLM adds a second layer of semantic understanding, catching cases where embedding similarity alone might miss conceptual relationships. This two-stage process balances computational efficiency (embedding search) with nuanced judgment (LLM evaluation) (Lu et al., 2025).

### N.8 Reflection and Validation for Task Quality

**Design Choice:** The scientist LLM attempts to solve its own generated tasks, with automatic correction for compilation errors and refinement prompts for logic errors.

**Justification:** This self-evaluation cycle addresses a critical challenge in synthetic data generation: ensuring that automatically generated tasks are well-formed and solvable. Recent work on synthetic data quality (Havrilla et al., 2024) emphasizes that data quality matters as much as quantity.

The iterative refinement process implements a form of minimal criterion for task quality, preventing the task archive from accumulating broken or ambiguous tasks. By having the scientist LLM solve its own tasks before accepting them, we create selection pressure toward tasks with clear problem statements and unambiguous scoring functions.

This also relates to the principle of "autotelic learning" in open-ended systems (Etcheverry et al., 2021), where the system must develop its own evaluation criteria. The scientist LLM effectively learns to generate tasks that meet implicit quality standards through its own solution attempts.

### N.9 Historical Model Archive and Task Force Selection

**Design Choice:** AC/DC maintains a historical archive of all models every $G_{task}$ generations and selects the final task force by maximizing Coverage over the global task archive.

**Justification:** The historical archive implements a key principle from QD: maintaining a collection of diverse high-quality solutions rather than just the current population (Pugh et al., 2016). This is crucial because behavioral diversity at intermediate generations might not be preserved if we only keep the final population. Some specialized models might be replaced during evolution, but could still contribute unique capabilities to the final ensemble.

The Coverage-based selection for the task force directly optimizes for our practical goal: collective problem-solving capacity across diverse tasks. This selection is independent of downstream benchmarks, implementing the "novelty search without objectives" approach while still providing a clear utility-based criterion for ensemble construction.

Importantly, this selection happens after coevolution completes, avoiding optimization pressure during evolution. This prevents the kind of overfitting to specific benchmarks that could be expected in recent evolutionary model merging work (Akiba et al., 2025). Our models are selected based on synthetic task Coverage, then evaluated on held-out benchmarks, ensuring genuine out-of-distribution generalization.

### N.10 Synthesis: AC/DC as an Open-Ended System

Taken together, these design choices implement the key properties of open-ended systems identified by Stanley et al. (2017) and Stanley & Lehman (2015):

1. **Continual Generation of Novelty:** The coevolution of models and tasks produces an ongoing stream of new capabilities and challenges, with no predetermined endpoint.

2. **Increasing Complexity:** Task difficulty adaptation and model selection pressure drive both populations toward increasing sophistication over time (demonstrated in Fig. 1's improvement trajectory).

3. **Cumulative Innovation:** New models build on previous models through crossover, and new tasks build on previous tasks through adaptation, creating a stepping-stone effect where discoveries enable further discoveries.

4. **No Explicit Fitness Function:** While we use quality (task pass rates) and diversity (skill vector distances) for selection, we never optimize explicitly for downstream benchmarks, allowing unexpected capabilities to emerge.

5. **Minimal Criteria Rather Than Objectives:** Our gibberish and impossible task filters prevent completely undesired outcomes without constraining the search to predefined goals.

This combination of principles, grounded in established open-endedness literature, enables AC/DC to discover model collectives with broader and more diverse capabilities than methods that optimize directly for benchmark performance, while maintaining lower computational costs than training large monolithic models.

## O   LLM Usage During Paper Writing

We leverage LLMs to assist in polishing the paper's text and generating tables.

