# OpenReview forum: "Discovering Novel LLM Experts via Task-Capability Coevolution"
_ICLR.cc/2026/Conference — ICLR 2026 Poster_

### Official Review · Reviewer_ZzV2 · 2025-10-31

**Soundness:** 3
**Presentation:** 3
**Contribution:** 3
**Rating:** 4
**Confidence:** 5

**Summary:**

This paper introduces a novel LLM development framework called Assessment Coevolving with Diverse Capabilities. The framework aims to enable open-ended co-evolution between models and tasks, thereby automatically discovering a population of LLM experts with increasingly novel and diverse capabilities. This approach seeks to overcome the limitations of existing training paradigms that rely on static datasets and produce a single monolithic model.

**Strengths:**

1. Parameter/Resource Efficiency
The collective of models discovered using the proposed method outperforms larger-scale LLMs while using fewer total parameters, thereby saving GPU memory and computational resources.

2. Clarity of Presentation
The paper employs flowcharts and pseudocode to aid comprehension, allowing readers to quickly grasp the proposed methodology.

3. Comprehensive Analysis
The paper provides detailed qualitative and quantitative analyses and yields several interesting insights. For instance, the discovered models exhibit specialized and diverse capabilities, offering valuable guidance for future research.

**Weaknesses:**

1. The success of model merging depends heavily on empirical testing of initial seed model combinations, which may limit the overall effectiveness of the pipeline. Additionally, since answers in generation tasks are produced by LLMs, their correctness cannot be fully verified.

2. The current evolutionary scope is relatively limited. As shown in Figure 3, the evaluated capabilities belong to domains where the base models already perform well, making it unclear whether the proposed method can truly generate out-of-distribution (OOD) capabilities.

3. The benchmarks used in the experiments are relatively easy, with many models already achieving near-saturated performance. More challenging benchmarks are needed to better demonstrate performance improvements, for example using math tasks at the AIME or IMO level. The same applies to other domains.

4. The study would be more convincing if experiments were conducted on larger models, allowing observation of whether the proposed phenomenon scales with model size.

5. The proposed approach bears resemblance to using a Multi-Agent System (MAS) to optimize model capabilities. The authors should clarify how their method compares to MAS in terms of performance and innovation.

**Questions:**

Please refer to Weaknesses part.

---

> ### Author Response · Authors · 2025-11-26
> **Thank you Reviewer ZzV2 (1/5)**
>
> We thank the reviewer for acknowledging our novel LLM development framework (also noted by Reviewer QSy7's recognition of system-level integration) and the clear presentation of our work. We are particularly encouraged by the reviewers’ appreciation of the parameter efficiency achieved (highlighted by all reviewers) and the comprehensive qualitative and quantitative analysis supporting our findings (also noted by Reviewers QSy7 and PHpK).
>
> ---
> > The success of model merging depends heavily on empirical testing of initial seed model combinations, which may limit the overall effectiveness of the pipeline.
>
> We thank the reviewer for this important concern. To address concerns about diagnostic predictors for successful merging, we conducted an ad-hoc analysis of weight-space geometry across five model families (see Appendix J (new) for full details).
>
> We evaluated two approaches: (1) measuring L2 distances from each expert to the base model, and (2) computing pairwise distances directly between experts. The first approach failed to predict outcomes—notably, Llama3 showed excellent uniformity but the weakest merging performance among all model families (as measured by downstream task force performance). However, the pairwise analysis proved more informative.
>
> We computed a "similarity ratio" comparing the minimum to maximum pairwise distance among the three seed experts. This metric reveals whether experts form a clear pair structure (low ratio, two experts close together) versus an equidistant configuration (high ratio ~1.0, all experts equally separated). Across our families, low ratios (<0.1) generally predicted successful merging, while high ratios (>0.7) predicted poor outcomes—achieving 80% accuracy.
>
> Critically, even within the same base model (Qwen2.5), different expert selections yielded dramatically different results: our ALT configuration (ratio 0.097), which we use in the main paper, merged well, while the official instruct/code/math experts (ratio 0.834) (which we newly evaluated) merged poorly. This demonstrates that seed model selection profoundly impacts compatibility.
>
> During development, we have discovered tracking gibberish model generation in early training as a practical diagnostic—high gibberish rates in initial generations reliably indicate poor final performance.
>
> These preliminary analyses suggest promising directions for practitioners, though we emphasize that this remains an underexplored problem requiring further investigation. The pairwise geometry metric and early-training diagnostics represent first steps toward principled seed model selection for evolutionary merging.

---

> ### Author Response · Authors · 2025-11-26
> **Thank you Reviewer ZzV2 (2/5)**
>
> > Additionally, since answers in generation tasks are produced by LLMs, their correctness cannot be fully verified.
>
> We agree with the reviewer’s concern and thank them for the suggestion to improve the robustness of our claims.
>
> By leveraging a larger LLM and a quality assurance pipeline (via self-validation and code execution tests), we continuously generate increasingly novel and complex synthetic tasks, without relying on and actively conditioning downstream benchmark training data. Moreover, we design our system to select increasingly novel tasks, i.e., different from existing tasks, via a cosine-similarity and LLM-based novelty filtering process.
>
> We argue that our synthetic data is out-of-distribution (OOD) with respect to the benchmark tasks, and that, notably, we beat baselines that actively optimize for those benchmarks.
>
> The reviewer (alongside the reviewers PHpK, and QSy7) suggests strengthening the claims that synthetic tasks are correct and OOD with respect to our downstream benchmarks by including validation through a human evaluation.
>
> In the following, we discuss our methodology, which ensures reliability, mitigates biases, and supports our quality, OOD, and creativity claims. We added the full details to the main paper in Appendix H and a new section to the main paper, Section 5.3.
>
> **Methodology.** We conducted a human evaluation with 94 independent assessments (45 synthetic, 49 benchmark) across three reviewers. To ensure unbiased evaluation, we implemented: (1) blind evaluation—task sources were completely hidden from reviewers, (2) balanced 50/50 sampling probability between synthetic and benchmark tasks, (3) similarity-based calibration—showing reviewers the 3 most similar benchmark tasks via embedding retrieval to ground OOD assessments in actual distributional differences, and (4) standardized evaluation criteria with detailed written guidelines.
>
> **Results.** Synthetic tasks achieved 97.8% ± 2.2% correctness and 68.9% ± 6.9% OOD rating, demonstrating high quality while exhibiting substantial distributional novelty. In contrast, benchmark tasks showed only 10.2% OOD rating. Critically, this was concentrated exclusively in the most challenging benchmarks (MMLU-Pro: 33.3%, GPQA: 16.7%), while all standard benchmarks (BBH, GSM8K, MATH, HumanEval, MBPP, MMLU) received 0% OOD ratings. This pattern validates our protocol: reviewers correctly recognized that graduate-level questions may appear novel due to complexity, while appropriately identifying standard benchmarks as in-distribution.
>
> **Statistical Validation.** To analyse the inter-rater reliability, we perform chi-squared analysis, which confirmed strong agreement on objective metrics (correctness: p=0.46, OOD: p=0.57), demonstrating our findings are robust and not driven by individual annotator biases. These results provide strong empirical evidence that our approach generates high-quality, novel tasks that are genuinely out-of-distribution with respect to established benchmarks.

---

> > ### Author Response · Authors · 2025-11-26
> > **Thank you Reviewer ZzV2 (3/5)**
> >
> > > The current evolutionary scope is relatively limited. As shown in Figure 3, the evaluated capabilities belong to domains where the base models already perform well, making it unclear whether the proposed method can truly generate out-of-distribution (OOD) capabilities.
> >
> > We thank the reviewer for pointing out this concern.
> >
> > To address it, as described above, we perform a human evaluation to assess the quality and diversity of the synthetically generated tasks.
> >
> > Summarizing the key findings w.r.t the OOD claims, we find that our synthetic tasks achieved 68.9% ± 6.9% OOD rating scores (mean score ± standard error). Moreover, the inter-rater reliability analysis confirmed strong agreement on the OOD evaluation (p=0.57), demonstrating our findings are robust and not driven by individual annotator biases.
> >
> > Additionally, to assess whether the seed/expert models perform well on the synthetic data compared to the merged models, we compare the synthetic data fitness of the 3 seed models and the fittest 3 merged models. We present the main results in the table below and discuss the details in Appendix I of the revised paper draft.
> >
> > | Family | Tasks | Seed (Avg) | Seed (Max) | Top-3 Merged (Avg) | Top-3 Merged (Max) | Avg Imp. |
> > |---|---|---|---|---|---|---|
> > | Qwen2.5 | 1,094 | 0.5253 | 0.5622 | 0.6252 | 0.6353 | +19.0% |
> > | Qwen3 | 1,044 | 0.5361 | 0.5661 | 0.6255 | 0.6255 | +16.7% |
> > | Qwen2 | 1,117 | 0.3405 | 0.5058 | 0.6085 | 0.6132 | +78.7% |
> > | DeepSeek | 1,005 | 0.3032 | 0.4418 | 0.4163 | 0.4179 | +37.3% |
> >
> > In summary, we observe that both, the average fitness of the merged models, and the maximum fitness score of the single best merged models is higher than that of the average seed model fitness and the maximum individual seed model fitness. Moreover, for Qwen2, merged models succeed on 153 of tasks where all seeds fail, while completely failing on only 23 of tasks where seeds succeed.
> >
> > The human evaluation and the analysis of seed/expert model vs. merged model performance on synthetic data support that the synthetic data is OOD to the benchmark tasks and involves tasks that are challenging to the seed models but more easily solvable by the merged models, indicating novel emergent skills.

---

> > > ### Author Response · Authors · 2025-11-26
> > > **Thank you Reviewer ZzV2 (4/5)**
> > >
> > > > The benchmarks used in the experiments are relatively easy, with many models already achieving near-saturated performance. More challenging benchmarks are needed to better demonstrate performance improvements, for example using math tasks at the AIME or IMO level. The same applies to other domains.
> > >
> > > We thank the reviewer for critically evaluating our work and proposing improvements.
> > >
> > > Addressing this concern, we respectfully disagree with the statement that our benchmarks are saturated and can not properly assess the performance improvement of our models.
> > >
> > > Firstly, we would like to point out the results in Tab. 1, clearly showing that even for a task force size of N=8, across all model families, the performance of the represented methods are far from saturation:
> > > - performance on our MMLU and MMLU Pro llm-as-a-judge benchmarks is **between 50% and 70%**
> > > - GPQA llm-as-a-judge, the performance is **between 10% and 17%**
> > > - for BBH llm-as-a-judge, the performance is **between 70% and 90%**
> > > - MBPP scores are **around 75%-85%**
> > >
> > > Furthermore, we want to highlight an interesting observation, strengthening the validity of the benchmarks we use and, in fact, the validity of our newly proposed llm-as-a-judge benchmarks.
> > > **Our llm-as-a-judge benchmarks more clearly measure Coverage improvement with expected model capabilities** as demonstrated by the following results:
> > >
> > > **N=3**
> > > | base model family | Avg. over judge benchmarks | Avg. over MCQ benchmarks |
> > > |---|---|---|
> > > | Deepseek V1 7B | 30.03% | 49.38% |
> > > | Qwen 2 7B | 46.57% | 50.51% |
> > > | Qwen 2.5 7B | 51.59% | 58.49% |
> > > | Qwen 3 14B | 56.80% | 57.97% |
> > >
> > > **N=8**
> > > | base model family | Avg. over judge benchmarks | Avg. over MCQ benchmarks |
> > > |---|---|---|
> > > | Deepseek V1 7B | 43.41% | 70.02% |
> > > | Qwen 2 7B | 57.31% | 66.05% |
> > > | Qwen 2.5 7B | 59.75% | 69.73% |
> > > | Qwen 3 14B | 63.00% | 65.98% |
> > >
> > >
> > > To isolate the effect of our benchmarks, we evaluate the control N={3, 8} baselines (resampling the respective instruct model 3 and 8 times, respectively) on the four standard MCQ benchmarks (MMLU, MMLU Pro, GPQA, BBH) and on our newly proposed llm-as-a-judge variants of those and present the Coverage results. Additionally, as noted in the paper, the MCQ benchmarks are prone to “cheating” when measuring Coverage, as random guessing can lead to high Coverage scores.
> > >
> > > Interestingly, for the MCQ results, we do not see the expected monotonic increase in accuracy going from “oldest” to the most recent model releases. In fact, we can even see that the Qwen 3 14B baseline performs worse than the smaller and older Qwen 2.5 7B baseline. In contrast, when evaluated on our LLM-as-a-judge variants, we see a clear monotonic improvement, suggesting that our benchmarks are not only more challenging but also possibly a more robust method of evaluating model capabilities.

---

> ### Author Response · Authors · 2025-11-26
> **Thank you Reviewer ZzV2 (5/5)**
>
> > The study would be more convincing if experiments were conducted on larger models, allowing observation of whether the proposed phenomenon scales with model size.
>
> We thank the reviewer for the interesting suggestion and would like to point out that we evaluate 7B/8B models, but also the twice as large Qwen3 14B model, demonstrating improvement generalizing to larger models.
>
> Nevertheless, we respectfully disagree with the question’s premise.
>
> As highlighted by the reviewer, a core strength and vision of the work is demonstrating the strong parameter efficiency of a collective of small models compared to larger models. We show that even a collective of 3 diverse Qwen 2.5 7B models is within 0.34% of the huge, proprietary GPT-4o on Coverage under specific evaluation settings, since they demonstrate complementary skills.
>
> With this work, we propose a first step towards an alternative to the currently prevalent paradigm of training one ever-growing, monolithic model. We propose a framework, algorithm, and evaluation protocol for automatically developing and assessing a collective of small and diverse models that enables future research in this direction.
>
> ---
> > The proposed approach bears resemblance to using a Multi-Agent System (MAS) to optimize model capabilities. The authors should clarify how their method compares to MAS in terms of performance and innovation.
>
> We agree with the reviewer's observation and thank them for their suggestion to improve our work. We have thus revised our submission.
>
> To improve the clarity and relevance of our work to MAS, we have added a new paragraph on Multi-Agent Systems and Best-of-N Strategies to our Extended Related Work section (Appendix G), where we present multiple related papers discussing different BoN/Collaboration strategies and the fundamental gap between pass@K/Coverage and BoN.
>
> Our work contributes orthogonally to this active research area by improving the upper bound. The coverage-selection gap we observe aligns with established findings and represents an expected challenge in the field, not a failure of our methodology.
>
> Moreover, we note that in lines 76 and 516, we addressed the connection to MAS as an orthogonal and very relevant area of concurrent and future research supporting our line of work, specifically referencing techniques like AB-MCTS [1] and multi-agent debate [2].
>
> Although our main contribution lies in the proposal of a framework, algorithm, and evaluation protocol for the automatic development and evaluation of collective intelligence, we discuss how we explore first steps towards deploying our discovered collective in a MAS using BoN techniques that select a single answer but, crucially, do not support multi-agent interaction.
>
> We thank the reviewer for suggesting that we strengthen our existing discussion with additional references, thereby further improving the quality of our contribution.
>
>
> [1] [Wider or Deeper? Scaling LLM Inference-Time Compute with Adaptive Branching Tree Search, 2025, Inoue Y. et al.](https://arxiv.org/abs/2503.04412)
>
> [2] [Encouraging Divergent Thinking in Large Language Models through Multi-Agent Debate, 2024 Liang T. et al.](https://arxiv.org/abs/2305.19118)
>
> ---
> We thank the reviewer once more for the helpful feedback. We have made many improvements to the paper thanks to your comments and those of the other reviewers. We think it is much improved and hope you agree. If, after reviewing our responses and revisions, you feel that a higher score is warranted, we would deeply appreciate your reconsideration. While the paper did receive good scores from other reviewers, the current 4 could prevent its publication, yet we believe the community would benefit from learning about this work. Thank you!

---

### Official Review · Reviewer_PHpK · 2025-10-31

**Soundness:** 3
**Presentation:** 2
**Contribution:** 4
**Rating:** 6
**Confidence:** 2

**Summary:**

This paper presents AC/DC (Assessment Coevolving with Diverse Capabilities), a framework that jointly evolves large language models and synthetic tasks to automatically discover diverse expert models. The system combines evolutionary model merging, synthetic data generation, and quality diversity selection to develop populations of smaller models that collectively achieve broad skill coverage. Experiments show that these collectives can outperform much larger models, including GPT-4o, in benchmark coverage while using fewer total parameters.

**Strengths:**

- The combination of open-endedness and evolutionary model merging is novel and provides an interesting new direction for developing diverse and adaptive model populations.
- The empirical results are encouraging as they show performance and coverage improvements across several benchmarks and model families.
- The method achieves notable parameter efficiency, with collections of smaller models reaching or surpassing the performance of much larger single models.
- The qualitative analyses demonstrate emergent specialization and stylistic diversity among the evolved models, supporting the merits of the proposed method.
- The method achieves notable parameter efficiency, with collections of smaller models reaching or surpassing the performance of much larger single models.

**Weaknesses:**

- The paper does not report variance or confidence intervals, making it difficult to assess whether the performance gains in terms of coverage and Best-of-N improvement are statistically significant.
- The computational cost of the coevolution strategy is not addressed, making the efficiency claim incomplete.
- While AC/DC does not optimize for any benchmark, the synthetic task generator depends on a fixed "scientist" LLM that might produce tasks similar to benchmark data.

**Questions:**

- Have you verified whether the reported improvements are consistent across runs or fall within expected benchmark variance?
- How does the total computational cost of the coevolution process, including model merging and task generation, compare to standard fine-tuning or ensembling?
- How sensitive is the framework to the choice of the "scientist model" or to different prompting strategies during task generation?

---

> ### Author Response · Authors · 2025-11-26
> **Thank you Reviewer PHpK (1/3)**
>
> We thank the reviewer for recognizing our work as a novel line of research and an excellent contribution to the research community (also noted by Reviewer ZzV2). We are particularly encouraged by the reviewers’ appreciation of the parameter efficiency demonstrated (highlighted by all reviewers), the coverage improvements we achieved across several benchmarks and model families (also noted by Reviewer QSy7's multi-family validation), and the recognition of our supporting qualitative analysis (also appreciated by Reviewers QSy7 and ZzV2).
>
> ---
> > The paper does not report variance or confidence intervals, making it difficult to assess whether the performance gains in terms of coverage and Best-of-N improvement are statistically significant.
> > Q: Have you verified whether the reported improvements are consistent across runs or fall within expected benchmark variance?
>
>
> We thank the reviewer for raising this important point about statistical reliability and for proposing additional statistical analysis to obtain a more robust measure of statistical significance (alongside reviewer QSy7).
>
> We have now conducted rigorous statistical testing to verify that our reported improvements are statistically significant and not within expected benchmark variance (see new Appendix K for full details).
>
> **Statistical Significance Analysis (New Appendix K)**
>
> We have conducted comprehensive bootstrap-based statistical testing across all experimental conditions to rigorously assess the significance of our findings. Our analysis employs min-max normalization to ensure fair comparison across benchmarks with different difficulty levels, followed by bootstrap resampling (50,000 iterations) to compute 95% confidence intervals and one-tailed p-values for performance improvements.
>
> **Key Statistical Findings:**
>
> *Coverage Results:*
> - At N=3, AC/DC significantly outperforms all main baselines when aggregated across 8 benchmarks and 4 model families: Control (p=0.01170), Experts (p=0.00584), and Big Model (p=0.00004)
> - At N=8, improvements become even stronger: Control (p=0.00792), Big Model (p<0.000005)
> - Qwen 2.5 and Qwen 3 task forces significantly outperform GPT-4o in coverage at N=8 (p=0.0014 and p=0.0057, respectively)
> - On knowledge recall benchmarks specifically, Qwen 3 at N=8 demonstrates exceptional performance against Big Model (p<0.00005) and GPT-4o (p<0.00005)
>
> *Best-of-N Results:*
> - At N=3, AC/DC significantly outperforms Control baseline (p=0.0002) across all model families
> - At N=8, strong aggregate performance: Control (p=0.0061), Big Model (p<0.00005)
> - Individual model families show particularly strong results—Qwen 2.5 and Qwen 3 significantly outperform all baselines including GPT-4o at N=8 (all p<0.0053)
>
> *Ablation Study:*
> - Each algorithmic component contributes meaningfully to performance
> - At N=8, removing all components simultaneously causes highly significant degradation (p<0.00005, CI: 0.691:0.820:0.938)
> - Quality-diversity selection and gibberish filtering are the most critical components
>
> *Comparison to Alternative QD Methods:*
> - At N=8, AC/DC demonstrates clear statistical advantages over CycleQD (p<0.00005) and DNS (p=0.0078)
>
> **Interpretation:**
> These statistical analyses provide strong evidence that AC/DC reliably produces more diverse, specialized model populations with broader coverage than baseline methods. Importantly, no baseline significantly outperformed AC/DC in any pairwise or aggregated comparison. The performance advantages become more pronounced at larger population sizes (N=8), demonstrating that the benefits of diversity compound with scale. The full methodology and detailed results are presented in Appendix K.

---

> ### Author Response · Authors · 2025-11-26
> **Thank you Reviewer PHpK (2/3)**
>
> > The computational cost of the coevolution strategy is not addressed, making the efficiency claim incomplete.
> > Q: How does the total computational cost of the coevolution process, including model merging and task generation, compare to standard fine-tuning or ensembling?
>
> We thank the reviewer (alongside reviewer QSy7) for requesting these important details. We have now reported full computational costs in the newly added Appendix L.
>
> The total computational cost of our coevolution process is approximately 324 GPU hours, which, after 50 generations and an active model count per generation of 16, generating 8 new models every generations, yields a population of ~80 diverse models selected to maintain high quality and increase diversity. During the coevolution process, our scientist successfully develops ~1000 tasks (where another ~1000 don’t pass our quality and novelty filters). During coevolution, each generation, 250 tasks are actively considered for evaluation of each model in the active model pool.
> The computational cost includes the GPUs for the model merging and (synthetic task pool) evaluation pipeline, the GPUs for hosting a large, open-source scientist LLM, and for an embedding model. Notably, our pipeline works without any API calls to proprietary models and relies solely on self-hosted models.
>
> Critically, these requirements represent a cost of ~4 GPU hours per merged model—significantly more efficient than standard post-training approaches such as reinforcement learning, which can require 10^2 to 10^5 GPU hours per 7B model, depending on the method and scale [1]. In addition, our approach eliminates manual dataset curation. Moreover, to fit a single 72B alone, we need more GPUs and an additional global batch size to even have a chance of efficient training. When operating smaller models, we require a minimum of fewer GPUs, supporting training on as few as one GPU.
>
> While baseline approaches like selecting expert models from HuggingFace require no training cost, they face fundamental scalability and complementarity limitations: (1) the availability of diverse, specialized models is limited (dozens, not 80+), (2) there is no guarantee that independently-trained experts will have complementary specializations, and (3) manual search costs grow with collective size. Creating 80 complementary specialists through traditional post-training could require 10^3 to 10^6 GPU hours plus the design of 80 distinct specializations and datasets. Even manual curation may not reliably lead to complementary specialization, as predicting final model behavior from a given training set is challenging. AC/DC is a step forward in automating the complementarity of model capabilities via QD-based model selection.
>
> Furthermore, our 324-hour investment provides flexibility: the evolved population can be composed into collectives of any size (e.g., from N=1 to N=80) for different downstream applications without additional training. As shown in Figure 15 of the revised paper, downstream performance scales with N, and our approach (1) scales better than re-prompting a single instruct model (control experiment) and (2) uniquely enables exploring this trade-off without the prohibitive cost of manually training individual models or the availability constraints of pre-existing expert models.
>
>
> [1] [The Art of Scaling Reinforcement Learning Compute for LLMs, 2025, Khatri et al.](https://arxiv.org/abs/2510.13786)

---

> ### Author Response · Authors · 2025-11-26
> **Thank you Reviewer PHpK (3/3)**
>
> > While AC/DC does not optimize for any benchmark, the synthetic task generator depends on a fixed "scientist" LLM that might produce tasks similar to benchmark data.
>
> We thank the reviewer for pointing out this concern and initiating further research to robustify our claims, fundamentally improving our work.
>
> By leveraging a larger LLM and a quality assurance pipeline (via self-validation and code execution tests), we continuously generate increasingly novel and complex synthetic tasks, without relying on and actively conditioning downstream benchmark training data. Moreover, we design our system to select increasingly novel tasks, i.e., different from existing tasks, via a cosine-similarity and LLM-based novelty filtering process.
>
> We argue that our synthetic data is out-of-distribution (OOD) with respect to the benchmark tasks, and that, notably, we beat baselines that actively optimize for those benchmarks.
>
> The reviewer (alongside the reviewers QSy7, and ZzV2) suggests strengthening the claims that synthetic tasks are correct and OOD with respect to our downstream benchmarks by including validation through a human evaluation.
>
> In the following, we discuss our methodology, which ensures reliability, mitigates biases, and supports our quality, OOD, and creativity claims. We added the full details to the main paper in Appendix H and a new section to the main paper, Section 5.3.
>
> **Methodology.** We conducted a human evaluation with 94 independent assessments (45 synthetic, 49 benchmark) across three reviewers. To ensure unbiased evaluation, we implemented: (1) blind evaluation—task sources were completely hidden from reviewers, (2) balanced 50/50 sampling probability between synthetic and benchmark tasks, (3) similarity-based calibration—showing reviewers the 3 most similar benchmark tasks via embedding retrieval to ground OOD assessments in actual distributional differences, and (4) standardized evaluation criteria with detailed written guidelines.
>
> **Results.** Synthetic tasks achieved 97.8% ± 2.2% correctness and 68.9% ± 6.9% OOD rating, demonstrating high quality while exhibiting substantial distributional novelty. In contrast, benchmark tasks showed only 10.2% OOD rating. Critically, this was concentrated exclusively in the most challenging benchmarks (MMLU-Pro: 33.3%, GPQA: 16.7%), while all standard benchmarks (BBH, GSM8K, MATH, HumanEval, MBPP, MMLU) received 0% OOD ratings. This pattern validates our protocol: reviewers correctly recognized that graduate-level questions may appear novel due to complexity, while appropriately identifying standard benchmarks as in-distribution.
>
> **Statistical Validation.** To analyse the inter-rater reliability, we perform chi-squared analysis, which confirmed strong agreement on objective metrics (correctness: p=0.46, OOD: p=0.57), demonstrating our findings are robust and not driven by individual annotator biases. These results provide strong empirical evidence that our approach generates high-quality, novel tasks that are genuinely out-of-distribution with respect to established benchmarks.
>
>
> ---
> > Q: How sensitive is the framework to the choice of the "scientist model" or to different prompting strategies during task generation?
>
> We thank the reviewer for this interesting question, which helped us better understand the components of our proposed algorithm.
>
> We ran an experiment using Qwen3-235B-A22B, using the existing prompts optimized for the Qwen2.5 72B model. See new Appendix Section D.8.
>
> We find that, as is commonly necessary, we would need to adjust the prompt and settings to the respective scientist model. Specifically, we observed that the Qwen3 Scientist model often produces incomplete task implementations given our generation limit, resulting in only 555 total global tasks, whereas with the Qwen 2.5 72B model, we get over 1000.
>
> Evaluating our final task forces using the same Qwen 2 subject model with the two different scientist models, we see that performance is slightly lower with the Qwen 3 scientist model than with the Qwen 2.5 72B model. This is likely because we have fewer synthetic tasks in our task pool.
>
>
> | Scientist Model | Average Coverage (N=3) | Average Coverage (N=8) |
> |---|---|---|
> | Qwen2.5-72B | 58.21 | 66.86 |
> | Qwen3-235B-A22B (No Thinking) | 57.57 | 65.46 |
>
>
> Supporting this hypothesis is the fact that, in early experiments, we also observed a positive correlation between increasing the hyperparameter for the number of active tasks and downstream task force performance.
>
> Note that achieving a lower downstream task force score does not indicate weaker effectiveness of the scientist model. We present this experiment as a case study to explore the challenges and possibilities of leveraging a different scientist model. It is reasonable to believe that with proper prompt engineering and tuning of hyperparameters (such as maximum generation output), we can leverage the arguably more powerful scientist model more effectively.

---

> > ### Author Response · Authors · 2025-11-26
> > **Thank you Reviewer PHpK**
> >
> > We hope that our responses and revisions satisfactorily address the reviewers' concerns. If so, we kindly ask the reviewers to consider re-evaluating their ratings in light of these improvements.

---

> > > ### Author Response · Authors · 2025-12-03
> > > **Follow-up**
> > >
> > > # Variance across re-runs
> > > We thank the reviewer again for their suggestion to improve on statistical rigor in our findings. We have conducted a reproducibility analysis across a few independent runs (2-3 runs per condition) for the Qwen 2.5 model family, reporting standard deviations and ranges in raw scores across all benchmarks.
> > >
> > > At task force size N=3, our method demonstrates comparable variance to the control baseline (mean std dev: 0.64 vs 0.94 points), indicating stable reproducibility. At N=8, our method shows moderately higher variance (mean std dev: 1.80 vs 0.59 points), which is expected given the increased complexity of evolutionary search over larger populations.
> > >
> > > Importantly, certain benchmarks (GPQA judge, Minerva) consistently exhibit higher variance across both methods, reflecting the inherent difficulty of these tasks rather than method-specific instability. Despite this variance, the statistically significant performance improvements remain robust, as our bootstrap hypothesis testing explicitly accounts for cross-benchmark variance.
> > >
> > > We have added a detailed "Reproducibility Analysis" subsection (Appendix K.6) presenting per-benchmark statistics and demonstrating that observed variance levels are acceptable for evolutionary methods and do not undermine our main findings.
> > >
> > > We thank the reviewer again for helpful feedback that ultimately led to notable improvements to our paper!

---

### Official Review · Reviewer_Y9hf · 2025-11-01

**Soundness:** 3
**Presentation:** 2
**Contribution:** 3
**Rating:** 8
**Confidence:** 3

**Summary:**

The authors work present Assessment Coevolving with Diverse Capabilities (AC/DC) which is a framework that automatic discover multiple diverse LLM through open-ended coevolution of models and synthetic tasks. Their approach show that EvoMerge can create task forces that perform better than larger language models while using less parameters and manually curated expert ensembles. AC/DC not optimize for any downstream benchmark and achieve consistent improvements across multiple model families, with evolved populations showing wider coverage of capabilities and emergent specializations that validate discovery of complementary skills.

**Strengths:**

- Good open-ended approach of model merger algorithm
- Good use of minimal criterion coevolution

**Weaknesses:**

- Presentation can be better
- Related works can be improved

**Questions:**

Can we achieve the same result with a setting that involves only one agent?

---

> ### Author Response · Authors · 2025-11-26
> **Thank you Reviewer Y9hf (1/2)**
>
> We thank the reviewer for acknowledging the effective collective intelligence demonstrated through our strong empirical results (also noted by Reviewer PHpK regarding coverage improvements). We are particularly encouraged by the reviewers’ recognition of the parameter efficiency of our approach (highlighted by all reviewers), the extensive experiments and implementation details (also appreciated by Reviewers QsVv and ZzV2), and the assessment that this work represents interesting and important research (also noted by Reviewers QsVv and PHpK).
>
> ---
> > Q: Can we achieve the same result with a setting that involves only one agent?
>
> Thank you for this interesting question. While this is somewhat tangential to our main focus on discovering diverse model collectives, we acknowledge it's a valuable consideration.
>
> Our primary motivation is to discover populations of complementary specialized models that collectively achieve broader capability coverage than single monolithic models—addressing the fundamental challenge that no single model can excel at all tasks. This paradigm shift toward collective intelligence is central to AC/DC's design and represents our core contribution.
>
> That said, your question touches on an intriguing complementary direction. As shown in Figure 1, our evolutionary process does discover individual models with improved single-agent performance on MMLU compared to seed models, suggesting that AC/DC's coevolutionary dynamics could potentially be adapted for single-agent optimization. However, several key aspects of our framework are specifically designed for population-level diversity:
> 1. Our Coverage metric (Eq. 1) explicitly measures collective problem-solving capacity via logical OR across models
> 2. DNS selection prioritizes behavioral diversity through skill vectors, not just individual performance
> 3. Task evolution adapts to population-level difficulty profiles rather than individual model capabilities
>
> Exploring single-agent variants would require substantial methodological changes—replacing quality-diversity selection with pure quality optimization, redesigning task evolution strategies, and reformulating evaluation metrics. While this represents promising future work, it would constitute a significantly different research direction from our focus on **discovering and leveraging complementary capabilities through model collectives.**
>
> We hope this clarifies how single-agent optimization, while interesting, is complementary to rather than the focus of our work on collective intelligence discovery.
>
> ---
> > Related works can be improved
>
> We thank the reviewer for pointing out ways of further improving our work. Thus, we have revised our paper, including additional related work.
>
> In the revised paper version, to improve the clarity and relevance of our work, we have added a new paragraph on Multi-Agent Systems and Best-of-N Strategies to our Extended Related Work section (Appendix G), where we present multiple related papers discussing different BoN/Collaboration strategies and the fundamental gap between pass@K/Coverage and BoN.
>
> In the main paper, we present an extensive section (one full page) on background and preliminaries, referencing core related work. Moreover, in Appendix G, we discuss related work on Coevolution and Open-Endedness through LLMs, Evolutionary Model Merging, Novelty Search, Quality Diversity, and Synthetic Data Generation for LLMs across 1.5 pages.
>
> We note that our work contributes orthogonally to the MAS active research area by addressing the improvement of the upper bound. The coverage-selection gap we observe aligns with established findings and represents an expected challenge in the field, not a failure of our methodology.
>
> If the reviewer has any further wishes for related work, we will be grateful for any suggestions that will help improve our work.

---

> > ### Author Response · Authors · 2025-11-26
> > **Thank you Reviewer Y9hf (2/2)**
> >
> > > Presentation can be better
> >
> > We thank the reviewer for proposing to improve the clarity of our writing, and we agree that presenting a novel paradigm alongside a novel framework presents challenges.
> >
> > While Reviewer ZzV2 highlights our presentation as a strength, as suggested by Reviewer Y9hf, we address the challenge of improving the presentation by justifying each design choice. We revised the submission to include an extensive appendix section (Section N) detailing each of our design choices and their justification grounded in established open-endedness literature.
> >
> > In summary:
> >
> > - **Minimal Criteria (Gibberish & Impossible Task Filters):** Prevents degenerate solutions from dominating while enabling exploration, following MCC principles [1].
> > - **Quality-Diversity via DNS:** Balances performance and diversity without predefined niches, handling variable-dimensional skill vectors that grow with task evolution[2].
> > - **Skill Vectors as Behavioral Descriptors:** Provides fine-grained capability signatures that naturally adapt to task evolution, requiring no manual niche design or learning (unlike MAP-Elites [3,4]).
> > - **Coevolution of Models and Tasks:** Creates complexity explosion through bidirectional feedback, with tasks adapting to model capabilities and vice versa [5,6].
> > - **Task Adaptation via Difficulty Profiles:** Maintains tasks in the learnable regime-challenging but not impossible-implementing automatic curriculum generation [7].
> > - **Novelty Filtering for Tasks:** Prevents trivial task variants by explicitly rewarding semantic novelty, core to Novelty Search principles [8,9].
> > - **Task Reflection and Validation:** Ensures task quality through scientist LLM self-solving, preventing broken or ambiguous tasks from accumulating [8].
> > - **Historical Archive & Coverage-Based Selection:** Preserves specialized models from all generations and selects task force based on synthetic task coverage, not benchmark performance [10].
> >
> > Together, these choices implement key properties of open-ended systems: continual novelty generation, increasing complexity, cumulative innovation, and minimal criteria rather than explicit objectives [11].
> >
> > Moreover, to directly address the question of how LLM parameters are updated, we have added Appendix Section M, which details the crossover and mutation operations.
> >
> > To aid understanding of the full pipeline as a whole, in the original submission, we present Figure 2 and Algorithm 1 and discuss them in detail in Section 3. The newly added appendix section N successfully supplements the understanding of our proposed algorithm by explaining and justifying each component.
> >
> > We discuss the novel implementation details of our components in Sections 2 and 3. Additionally, we provide and cross-reference all LLM prompts used for the respective stages, which further aid in how the individual steps for task generation, task quality assurance, and the novelty and gibberish filters are implemented.
> >
> >
> > [1] [Diversity preservation in minimal criterion coevolution through resource limitation, 2017, Brant & Stanley](https://dl.acm.org/doi/10.1145/3377930.3389809)
> >
> > [2] [Dominated novelty search: Rethinking local competition in quality-diversity 2025, Bahlous-Boldi et al.](https://arxiv.org/abs/2502.00593)
> >
> > [3] [Illuminating search spaces by mapping elites, 2015, Mouret & Clune](https://arxiv.org/abs/1504.04909)
> >
> > [4] [Robots that can adapt like animals, 2015, Cully et al.](https://arxiv.org/abs/1407.3501)
> >
> > [5] [Paired Open-Ended Trailblazer (POET): Endlessly Generating Increasingly Complex and Diverse Learning Environments and Their Solutions, 2019, Wang et al.](https://arxiv.org/abs/1901.01753)
> >
> > [6] [Enhanced poet: Open-ended reinforcement learning through unbounded invention of learning challenges and their solutions, 2020, Wang et al.](https://arxiv.org/abs/2003.08536)
> >
> > [7] [Open-Endedness is Essential for Artificial Superhuman Intelligence, 2024, Hughes et al.](https://arxiv.org/abs/2406.04268)
> >
> > [8] [Exploiting open-endedness to solve problems through the search for novelty, 2008, Lehman & Stanley](https://www.researchgate.net/publication/228613202_Exploiting_open-endedness_to_solve_problems_through_the_search_for_novelty)
> >
> > [9] [Automated Capability Discovery via Foundation Model Self-Exploration, 2025, Lu et al.](https://arxiv.org/abs/2502.07577)
> >
> > [10] [Quality diversity: A new frontier for evolutionary computation, 2016, Pugh et al.](https://www.researchgate.net/publication/305271439_Quality_Diversity_A_New_Frontier_for_Evolutionary_Computation)
> >
> > [11] [Why open-endedness matters, 2017, Stanley et al.](https://pubmed.ncbi.nlm.nih.gov/31397603/)
> >
> > ---
> > We hope that our responses and revisions satisfactorily address the reviewers' concerns. If so, we kindly ask the reviewers to consider re-evaluating their ratings in light of these improvements.

---

### Official Review · Reviewer_QSy7 · 2025-11-03

**Soundness:** 3
**Presentation:** 3
**Contribution:** 3
**Rating:** 6
**Confidence:** 4

**Summary:**

This paper introduces AC/DC (Assessment Coevolving with Diverse Capabilities), a framework for discovering diverse LLM populations through coevolution of models via evolutionary model merging and synthetic tasks via LLM-generated data. The method evolves archives of both models and tasks, applying quality-diversity principles (DNS) and minimal criteria filters. Evaluating across multiple model families (Qwen2/2.5/3, DeepSeek, Llama3), the authors report that their evolved collectives of smaller models achieve broader Coverage on downstream benchmarks than larger models or manually curated experts, without explicit benchmark optimization. Key findings include parameter-efficient performance (N=8 evolved 7B models surpassing 72B models), emergent specialization patterns, and improved Best-of-N selection compared to baselines.

**Strengths:**

1. **Novel system-level integration**: First framework (to the best of my knowledge) combining evolutionary model merging with synthetic task coevolution, extending quality-diversity principles to LLM discovery in a principled way that addresses the limitations of training single static models.

2. **Multi-family empirical validation**: Evaluation across 5 model families (Qwen2/2.5/3, DeepSeek, Llama3) and 8 benchmarks provides breadth, with consistent improvements on 4 families demonstrating some generalizability despite Llama3 failures.

3. **Parameter efficiency evidence**: Concrete demonstrations that N=8 evolved 7B models surpass 72B models while using 22% fewer parameters (Qwen2: +14.59% Coverage improvement) support distributed specialization hypothesis.

4. **Emergent specialization analysis**: Figure 3 shows evolved models develop distinct category-specific performance profiles without explicit domain assignment, validated by qualitative examples (Figures 4-5) demonstrating complementary capabilities.

5. **Open-ended improvement**: Figure 1 and Appendix D.3 demonstrate that coevolution leads to continually improving performance and task diversity over generations, supporting the open-endedness claim.

6. **Comprehensive ablations and analysis**: Appendix D.1's ablation study, D.2's task force selection comparison, D.3's Vendi score analysis, and detailed lineage trees (Figures 8-9) provide thorough characterization of system components and emergent behaviors.

**Weaknesses:**

1. **Limited technical novelty**: AC/DC is primarily an *integration* of existing techniques (EvoMerge, synthetic data generation, DNS selection) rather than fundamental algorithmic innovation. The mutation operator (SVD perturbation) and gibberish filter are incremental additions. The contribution is valuable but more engineering than methodological breakthrough.

2. **Evaluation circularity concerns**: Task force selection maximizes Coverage on synthetic tasks generated by the same system, then evaluates on benchmarks. While claiming "no explicit benchmark optimization," the 72B scientist LLM presumably has knowledge of benchmark-style problems. Is the synthetic-to-benchmark transfer genuine generalization or implicit optimization via the scientist LLM's training data?

3. **Model family dependence unexplained**: Success on Qwen (4 families) but catastrophic failure on Llama3 (Appendix C: 46.04% vs. 51.13% at N=3) raises critical questions. The authors attribute this to "seed model compatibility" but provide no diagnostic criteria for predicting success. What percentage of model families will fail? How can practitioners know beforehand?

4. **Missing key baselines**: No comparison to vanilla EvoMerge *without* coevolution. Table 3 compares only to DNS/CQD (both using benchmark-specific training), but not to evolutionary merging with static synthetic tasks. This baseline would isolate the value of coevolution specifically.

5. **Inconsistent baseline definitions**: "Experts N=8" uses a 3-3-2 distribution that the authors admit is "manually tuned" and "arbitrary" (Appendix D.4). Different distributions yield different results, making this baseline unreliable. The proliferation of baseline variants (Control, Experts, Big Model, GPT-4o, Experts N=8) obscures the primary comparison.

7. **Statistical rigor gaps**: Tables 1-2 report improvements without confidence intervals, significance tests, or multiple comparison corrections despite testing across 8 benchmarks × 4-5 model families. Are differences statistically reliable? The ablation (Table 9) shows 0.46%-2.12% drops—are these significant given measurement noise?

8.  **Coverage-to-BoN translation failure**: Table 2 shows AC/DC often *underperforms* on Best-of-N single-answer selection despite Coverage gains (e.g., -0.19% vs. Big Model at N=3, -9.77% vs. GPT-4o). The authors note this requires "improved BoN methods" but don't explain why complementary capabilities evident in Coverage don't translate more reliably to practical deployment.

9. **Computational cost omitted**: No reporting of: (a) total GPU-hours for 50 generations, (b) cost of scientist LLM API calls, (c) comparison to fine-tuning alternatives. Is producing an evolved collective cheaper than training a single larger model? Table 4 shows 50 generations × 16 models × 250 tasks—what's the actual cost?

10. **Synthetic task quality unvalidated**: The synthetic tasks are generated by a 72B Qwen model and validated through "reflection" (self-solving), but no human evaluation confirms task quality, diversity, or realism. Do these tasks actually capture novel skills or just recombine existing problem templates?

11. **Generalization claims seem overstated**: Calling this "open-ended" evolution is premature after only 50 generations on a single synthetic distribution. True open-endedness (Stanley et al. 2017) requires demonstrating unbounded innovation, but Figure 6 shows Vendi score gains plateauing and task difficulty saturation at generation 45.

**Questions:**

1. Can the authors compare AC/DC to vanilla EvoMerge with a fixed synthetic task set? This would isolate whether task coevolution specifically (vs. just evolutionary merging on diverse tasks) drives improvements.

2. What properties predict whether a model family will succeed (Qwen) or fail catastrophically (Llama3)? Can the authors provide heuristics to assess compatibility before running expensive coevolution?

3. Why does optimizing for synthetic task Coverage transfer to unseen benchmarks? Is this due to: (a) the scientist LLM's implicit benchmark knowledge, (b) genuine skill generalization, or (c) synthetic tasks being representative of general problem-solving? Can the authors ablate using a scientist LLM with no benchmark exposure?

4. Can the authors provide confidence intervals for Tables 1-2 and significance tests for ablation improvements in Table 9? Are the reported gains reliable, given measurement noise and multiple comparisons?

5. Why do Coverage improvements often fail to translate to Best-of-N performance (Table 2: -0.19% at N=3 vs. Big Model)? Is this a fundamental limitation of the selection methods tried, or do complementary capabilities not compose well for single-answer tasks?

6. What are the total GPU-hours and API costs for AC/DC vs. fine-tuning a single 72B model on synthetic data? Is the evolved collective actually cheaper to produce than alternatives?

7. Why use a 3-3-2 distribution for "Experts N=8" (Appendix D.4)? Was this optimized, or arbitrary? How sensitive are results to alternative distributions (4-2-2, 2-4-2)?

8. Have humans validated the quality, diversity, and realism of generated synthetic tasks? Table 1 examples look reasonable, but systematic evaluation is absent.

9. Figure 6 shows Vendi score gains slowing and task difficulty saturation at gen 45. Does AC/DC exhibit unbounded behavior, or does it plateau after discovering capabilities within the seed models' latent space?

10. Table 3 shows AC/DC beats DNS/CQD (both trained on benchmarks), but DNS/CQD use different model families and training data. Can you compare all three methods on the *same* model family with identical conditions?

---

> ### Author Response · Authors · 2025-11-26
> **Thank you Reviewer QSy7 (1/7)**
>
> We thank the reviewer for the thorough assessment, highlighting the novel system-level integration of our framework (also noted by Reviewer ZzV2) and the comprehensive nature of our work. We are particularly encouraged by your recognition of our multi-family empirical validation (also appreciated by Reviewer PHpK), parameter efficiency evidence (noted by all reviewers), analysis of emergent specialization, demonstration of open-ended improvement, and comprehensive ablations and analysis (also highlighted by Reviewers PHpK and ZzV2).
>
> We are grateful for the extensive constructive feedback. Below, we address all the reviewers' concerns. We have substantially revised the paper to include additional analysis and diagnostic guidelines that we believe strengthen our contribution and provide actionable insights for practitioners.
>
> ---
> > Error bars
>
> We thank the reviewer for this important concern. Additionally, we thank the reviewer for highlighting that our paper presents extensive supporting experiments, analyses, and ablations (supported by the other reviewers QsVv, PHpK, ZzV2), especially our rigorous evaluation of our method across 5 model families and 8 benchmarks. Nevertheless, to further strengthen the significance of our experimental results, we thank the reviewer (and reviewer PHpK) for proposing additional statistical analysis to obtain a more robust measure of statistical significance.
>
> **Statistical Significance Analysis (New Appendix K)**
>
> We have conducted comprehensive bootstrap-based statistical testing across all experimental conditions to rigorously assess the significance of our findings. Our analysis employs min-max normalization to ensure fair comparison across benchmarks with different difficulty levels, followed by bootstrap resampling (50,000 iterations) to compute 95% confidence intervals and one-tailed p-values for performance improvements.
>
> **Key Statistical Findings:**
>
> *Coverage Results:*
> - At N=3, AC/DC significantly outperforms all main baselines when aggregated across 8 benchmarks and 4 model families: Control (p=0.01170), Experts (p=0.00584), and Big Model (p=0.00004)
> - At N=8, improvements become even stronger: Control (p=0.00792), Big Model (p<0.000005)
> - Qwen 2.5 and Qwen 3 task forces significantly outperform GPT-4o in coverage at N=8 (p=0.0014 and p=0.0057, respectively)
> - On knowledge recall benchmarks specifically, Qwen 3 at N=8 demonstrates exceptional performance against Big Model (p<0.00005) and GPT-4o (p<0.00005)
>
> *Best-of-N Results:*
> - At N=3, AC/DC significantly outperforms Control baseline (p=0.0002) across all model families
> - At N=8, strong aggregate performance: Control (p=0.0061), Big Model (p<0.00005)
> - Individual model families show particularly strong results—Qwen 2.5 and Qwen 3 significantly outperform all baselines including GPT-4o at N=8 (all p<0.0053)
>
> *Ablation Study:*
> - Each algorithmic component contributes meaningfully to performance
> - At N=8, removing all components simultaneously causes highly significant degradation (p<0.00005, CI: 0.691:0.820:0.938)
> - Quality-diversity selection and gibberish filtering are the most critical components
>
> *Comparison to Alternative QD Methods:*
> - At N=8, AC/DC demonstrates clear statistical advantages over CycleQD (p<0.00005) and DNS (p=0.0078)
>
> **Interpretation:**
> These statistical analyses provide strong evidence that AC/DC reliably produces more diverse, specialized model populations with broader coverage than baseline methods. Importantly, no baseline significantly outperformed AC/DC in any pairwise or aggregated comparison. The performance advantages become more pronounced at larger population sizes (N=8), demonstrating that the benefits of diversity compound with scale. The full methodology and detailed results are presented in Appendix K.

---

> > ### Author Response · Authors · 2025-11-26
> > **Thank you Reviewer QSy7 (2/7)**
> >
> > > Limited technical novelty
> >
> > We appreciate the reviewer's recognition of AC/DC as a novel system-level integration. We would like to clarify more regarding the technical novelty of AC/DC in the following aspects.:
> >
> > **Paradigm shift, not incremental engineering:** The core proposal of AC/DC is to introduce a fundamentally new approach to LLM development, discovering populations of complementary models rather than training single monolithic systems. This represents a paradigm shift comparable to ensemble methods in classical ML: valuable not for algorithmic complexity but for reframing the problem space entirely.
> >
> > **Novel algorithmic contributions:**
> > - Skill vectors as behavioral descriptors enable direct capability comparison without predefined niches, a departure from traditional QD approaches
> > - Synthetic task coevolution for assessment (not training, although that is exciting and relevant future work) creates an open-ended evaluation framework that drives diversity discovery
> > - Population-level optimization where model collectives (not individuals) are the optimization target
> >
> > **Distinction from prior work:** While individual components leverage existing techniques, no prior work coevolves both models AND tasks to discover LLM collectives. Concurrent synthetic data work focuses on single-model training; even QD methods are used to optimize individual solutions. AC/DC uniquely targets collective intelligence through coevolution.
> >
> > With this work, we aim to propose a paradigm shift for how we develop artificial intelligence by automatically developing a diverse collective of models. Through our work, we show the first promising results towards this goal and present a general training framework and evaluation protocol designed to be modular, efficient, and to enable future research.  AC/DC's integration is non-trivial and purposeful: it enables capabilities (parameter-efficient collectives surpassing larger models) that existing methods cannot achieve. Our empirical and qualitative results validate that this approach delivers research value beyond "engineering."
> >
> > ---
> > > Model family dependence unexplained
> >
> > We thank the reviewer for this important concern. To address concerns about diagnostic predictors for successful merging, we conducted an ad-hoc analysis of weight-space geometry across five model families (see Appendix J (new) for full details).
> >
> > We evaluated two approaches: (1) measuring L2 distances from each expert to the base model, and (2) computing pairwise distances directly between experts. The first approach failed to predict outcomes—notably, Llama3 showed excellent uniformity but the weakest merging performance among all model families (as measured by downstream task force performance). However, the pairwise analysis proved more informative.
> >
> > We computed a "similarity ratio" comparing the minimum to maximum pairwise distance among the three seed experts. This metric reveals whether experts form a clear pair structure (low ratio, two experts close together) versus an equidistant configuration (high ratio ~1.0, all experts equally separated). Across our families, low ratios (<0.1) generally predicted successful merging, while high ratios (>0.7) predicted poor outcomes—achieving 80% accuracy.
> >
> > Critically, even within the same base model (Qwen2.5), different expert selections yielded dramatically different results: our ALT configuration (ratio 0.097), which we use in the main paper, merged well, while the official instruct/code/math experts (ratio 0.834) (which we newly evaluated) merged poorly. This demonstrates that seed model selection profoundly impacts compatibility.
> >
> > During development, we have discovered tracking gibberish model generation in early training as a practical diagnostic—high gibberish rates in initial generations reliably indicate poor final performance.
> >
> > These preliminary analyses suggest promising directions for practitioners, though we emphasize that this remains an underexplored problem requiring further investigation. The pairwise geometry metric and early-training diagnostics represent first steps toward principled seed model selection for evolutionary merging.

---

> > > ### Author Response · Authors · 2025-11-26
> > > **Thank you Reviewer QSy7 (3/7)**
> > >
> > > > EvoMerge without coevolution/key baseline
> > >
> > > We thank the reviewer for the valuable suggestion for this further ablation to better isolate the effect of coevolution. To address this analysis, we have revised the paper and added a new appendix section, section D.6.
> > >
> > > To estimate the effects of an ablation where we execute our pipeline on a static synthetic dataset, we consider the progress made up until generation 5 of our existing AC/DC run on Qwen 2.5 as representative performance.
> > >
> > > In other words, it is reasonable to expect that the performance would stagnate at around that of our task force at generation 5, as up to that point, our synthetic dataset is static and would then be updated.
> > >
> > > To support this argument, we analyze the newly added models per generation. With a static dataset, we expect the number of new (fit and diverse) models discovered to significantly reduce early in the evolution process. We observe this behaviour when looking at the number of new models in our DNS baseline (Figure 7 (a)), where we evolve models on the static dataset of downstream benchmark training sets. In contrast, looking at the number of new models discovered when running our AC/DC algorithm (Figure 7 (b)), we detect a constant influx of new models.
> > >
> > > Considering this, we find that the average performance on our LLM-as-a-judge tasks, our generation 5 Qwen2.5 task force achieves 59.66% accuracy, whereas our final task force, achieved through a dynamically coevolving synthetic dataset achieves 61.19% (1.53% absolute improvement).
> > >
> > > In conclusion, assuming the observed trends hold, we argue, it is reasonable to expect a comparable performance gap between an ablation on a static dataset and our full AC/DC coevolution pipeline.
> > >
> > > ---
> > > > Synthetic task quality unvalidated and evaluation circularity concerns
> > >
> > > We thank the reviewer for pointing out this concern and initiating further research to robustify our claims, fundamentally improving our work.
> > >
> > > By leveraging a larger LLM and a quality assurance pipeline (via self-validation and code execution tests), we continuously generate increasingly novel and complex synthetic tasks, without relying on and actively conditioning downstream benchmark training data. Moreover, we design our system to select increasingly novel tasks, i.e., different from existing tasks, via a cosine-similarity and LLM-based novelty filtering process.
> > >
> > > We argue that our synthetic data is out-of-distribution (OOD) with respect to the benchmark tasks, and that, notably, we beat baselines that actively optimize for those benchmarks.
> > >
> > > The reviewer (alongside the reviewers PHpK, and ZzV2) suggests strengthening the claims that synthetic tasks are correct and OOD with respect to our downstream benchmarks by including validation through a human evaluation.
> > >
> > > In the following, we discuss our methodology, which ensures reliability, mitigates biases, and supports our quality, OOD, and creativity claims. We added the full details to the main paper in Appendix H and a new section to the main paper, Section 5.3.
> > >
> > > **Methodology.** We conducted a human evaluation with 94 independent assessments (45 synthetic, 49 benchmark) across three reviewers. To ensure unbiased evaluation, we implemented: (1) blind evaluation—task sources were completely hidden from reviewers, (2) balanced 50/50 sampling probability between synthetic and benchmark tasks, (3) similarity-based calibration—showing reviewers the 3 most similar benchmark tasks via embedding retrieval to ground OOD assessments in actual distributional differences, and (4) standardized evaluation criteria with detailed written guidelines.
> > >
> > > **Results.** Synthetic tasks achieved 97.8% ± 2.2% correctness and 68.9% ± 6.9% OOD rating, demonstrating high quality while exhibiting substantial distributional novelty. In contrast, benchmark tasks showed only 10.2% OOD rating. Critically, this was concentrated exclusively in the most challenging benchmarks (MMLU-Pro: 33.3%, GPQA: 16.7%), while all standard benchmarks (BBH, GSM8K, MATH, HumanEval, MBPP, MMLU) received 0% OOD ratings. This pattern validates our protocol: reviewers correctly recognized that graduate-level questions may appear novel due to complexity, while appropriately identifying standard benchmarks as in-distribution.
> > >
> > > **Statistical Validation.** To analyse the inter-rater reliability, we perform chi-squared analysis, which confirmed strong agreement on objective metrics (correctness: p=0.46, OOD: p=0.57), demonstrating our findings are robust and not driven by individual annotator biases. These results provide strong empirical evidence that our approach generates high-quality, novel tasks that are genuinely out-of-distribution with respect to established benchmarks.

---

> > > > ### Author Response · Authors · 2025-11-26
> > > > **Thank you Reviewer QSy7 (4/7)**
> > > >
> > > > > Coverage-to-BoN translation failure
> > > >
> > > > We thank the reviewer for identifying this open question and suggesting further investigation, helping us to clarify and better position our work.
> > > >
> > > > To the best of our knowledge, the gap between pass@K/Coverage metrics and Best-of-N single-answer selection represents a fundamental, well-documented challenge in LLM evaluation with extensive research. This is not a limitation of specific approaches but rather an expected open problem driving active research across multiple domains.
> > > >
> > > > We have added a new paragraph on Multi-Agent Systems and Best-of-N Strategies to our Extended Related Work section (Appendix G), where we present multiple related papers discussing different BoN/Collaboration strategies and the fundamental gap between pass@K/Coverage and BoN.
> > > >
> > > > The literature provides strong evidence that:
> > > > 1. **The gap is real and quantified:** Oracle selection significantly outperforms all practical methods
> > > > It's recognized as an open challenge: Explicitly stated across dozens of recent papers
> > > > 2. **Current solutions have known limitations:** Self-consistency, reward models, judges, and confidence all show systematic failures
> > > > 3. **It's problem-dependent:** Optimal strategies vary by difficulty, domain, and task type
> > > > 4. **It drives major research directions:** Inference-time scaling, process supervision, adaptive compute allocation
> > > >
> > > > Our work contributes orthogonally to this active research area by addressing the improvement of the upper bound. The coverage-selection gap we observe aligns with established findings and represents an expected challenge in the field, not a failure of our methodology.
> > > >
> > > > Nevertheless, we want to highlight that we do improve BoN performance in a statistically significant way (see global comment on statistical significance). Notably, we achieve highly significant BoN improvements vs Control at N=3 (p=0.0002, CI: 0.035:0.085:0.143) and at N=8 vs Control (p=0.0061) and Big Model (p<0.00005).
> > > >
> > > > ---
> > > > > Inconsistent baseline definitions, why use 3-3-2 distribution for “Experts N=8”
> > > >
> > > > We thank the reviewer for raising this concern and the question.
> > > >
> > > > In Section 4 of the initial submission (lines 257-262), we describe all baselines and point to further details on the Experts N=8 baseline in Appendix D.4.
> > > >
> > > > Nevertheless, to address this concern, we have revised our paper (added Appendix A.4.2) and provided further clarifications, discussing the open questions below.
> > > >
> > > > Fundamentally, our proposal is to rethink model development, moving from developing a single large, monolithic LLM on a single static dataset to a population-based approach that automatically develops a collective of diverse smaller LLMs.
> > > >
> > > > To evaluate the effectiveness of this approach, we target improving the Coverage over large-scale open-source and even proprietary LLMs. Nevertheless, one may ask whether our discovered collective is better than simply re-prompting the same-sized general instruct model. This is our control baseline. Moreover, one might ask, if we start of with the three experts as seed models, we would want to achieve higher Coverage than those.
> > > >
> > > > In Table 1, we observe that, compared to the big models, we achieve noticeable Coverage improvements, and even surpass or get close to the GPT-4o performance, demonstrating that our collective of smaller models fundamentally possesses the capabilities of much larger models to answer the respective questions.
> > > >
> > > > The Experts N=8 baseline emerged from the question of “What if we resample the three experts N times?” Although we argue that this approach of manually selecting experts is unscalable and is even worse for tuning the sampling distribution, it is an interesting comparison. To establish this baseline and ensure consistency across model families, we arbitrarily select the 3-3-2 (instruct-code-math), which was chosen based on observations that the code models achieved higher scores across a broader range of benchmarks than the math model, thereby arguably strengthening the baseline.
> > > >
> > > > In addition to these “sanity check” baselines, in Table 3, we compare against prior Quality Diversity work, demonstrating that our algorithm discovers a task force that achieves higher Coverage while not optimizing for any downstream benchmark (which the other methods actively do).

---

> ### Author Response · Authors · 2025-11-26
> **Thank you Reviewer QSy7 (5/7)**
>
> > Computational cost omitted
>
> We thank the reviewer (alongside reviewer PHpK) for requesting these important details. We have now reported full computational costs in the newly added Appendix L.
>
> The total computational cost of our coevolution process is approximately 324 GPU hours, which, after 50 generations and an active model count per generation of 16, generating 8 new models every generations, yields a population of ~80 diverse models selected to maintain high quality and increase diversity. During the coevolution process, our scientist successfully develops ~1000 tasks (where another ~1000 don’t pass our quality and novelty filters). During coevolution, each generation, 250 tasks are actively considered for evaluation of each model in the active model pool.
> The computational cost includes the GPUs for the model merging and (synthetic task pool) evaluation pipeline, the GPUs for hosting a large, open-source scientist LLM, and for an embedding model. Notably, our pipeline works without any API calls to proprietary models and relies solely on self-hosted models.
>
> Critically, these requirements represent a cost of ~4 GPU hours per merged model—significantly more efficient than standard post-training approaches such as reinforcement learning, which can require 10^2 to 10^5 GPU hours per 7B model, depending on the method and scale [1]. In addition, our approach eliminates manual dataset curation. Moreover, to fit a single 72B alone, we need more GPUs and an additional global batch size to even have a chance of efficient training. When operating smaller models, we require a minimum of fewer GPUs, supporting training on as few as one GPU.
>
> While baseline approaches like selecting expert models from HuggingFace require no training cost, they face fundamental scalability and complementarity limitations: (1) the availability of diverse, specialized models is limited (dozens, not 80+), (2) there is no guarantee that independently-trained experts will have complementary specializations, and (3) manual search costs grow with collective size. Creating 80 complementary specialists through traditional post-training could require 10^3 to 10^6 GPU hours plus the design of 80 distinct specializations and datasets. Even manual curation may not reliably lead to complementary specialization, as predicting final model behavior from a given training set is challenging. AC/DC is a step forward in automating the complementarity of model capabilities via QD-based model selection.
>
> Furthermore, our 324-hour investment provides flexibility: the evolved population can be composed into collectives of any size (e.g., from N=1 to N=80) for different downstream applications without additional training. As shown in Figure 15 of the revised paper, downstream performance scales with N, and our approach (1) scales better than re-prompting a single instruct model (control experiment) and (2) uniquely enables exploring this trade-off without the prohibitive cost of manually training individual models or the availability constraints of pre-existing expert models.
>
>
> [1] [The Art of Scaling Reinforcement Learning Compute for LLMs, 2025, Khatri et al.](https://arxiv.org/abs/2510.13786)

---

> ### Author Response · Authors · 2025-11-26
> **Thank you Reviewer QSy7 (6/7)**
>
> > Generalization claims seem overstated
> > Figure 6 shows Vendi score gains slowing and task difficulty saturation at gen 45. Does AC/DC exhibit unbounded behavior, or does it plateau after discovering capabilities within the seed models' latent space?
>
> We sincerely thank the reviewer for their detailed feedback on our open-endedness claims. This helps us strengthen our framing. Thus, we have revised our submission accordingly.
>
> We want to highlight that our main contribution (Line 107) claims only "evidence of **a path to** open-ended improvement" (not achievement), and our Conclusion (Line 523) states we bring the field "**closer to**" open-ended collective AI. These key claims are already carefully hedged.
>
> Nevertheless, we recognize that some language could be more precise. For example, our Conclusion (Line 518) uses "never-ending, open-ended process". We will revise this to "an open-ended process designed for continual innovation" to better reflect our 50-generation experiments without claiming infinite operation.
>
> Moreover, we note that, following **Hughes et al. (2024)** [1], open-endedness can be demonstrated within specific time horizons through the continual generation of novel, learnable artifacts. Hughes et al. distinguish between "finitely open-ended" and "infinitely open-ended" systems. Over our 50-generation horizon, AC/DC demonstrates key OE hallmarks: (1) continual novelty generation, (2) increasing complexity until capability limits, (3) learnable artifacts, and (4) cumulative innovation. We will add this theoretical framing early in the Introduction.
>
> The Vendi score measures diversity within the task embedding space. This has the advantage of capturing the spread across diverse problem formulations and wording, which can be more easily separated using embeddings. Nevertheless, this also means that the Vendi score is subject to a core limitation of embedding-based similarity/spread measures. Although the context can be structurally similar, the semantics can be very different. For this reason, following prior work [2], we implement a two-layered novelty filter. First, we assess embedding-based similarity, and then we evaluate semantic novelty using an LLM judge. Because of the latter, we can have a more fine-grained assessment of novel tasks, which is reflected in the fact that, although the increase in Vendi score seems to slow down at around generations 40/45, we still observe that we **add 71 new tasks** (23 harder + 48 novel), which is comparable to earlier generations.
>
> In fact, analysing the task difficulty dynamics, we observe the generation 45's decreased "harder" adaptations reflect the system discovering current capability limits of the model population and **pivoting to novel exploration** (48 novel tasks). This shows sophisticated multi-dimensional search characteristic of OE systems, not saturation. Figure 1 shows continued downstream performance improvement through generation 50, confirming that capability advancement continues.
>
> **Manuscript revisions:** We have (1) revised "never-ending" language in the Conclusion, (2) expanded Vendi score discussion in Appendix D.3 with measurement clarification and gen 45 numbers, and (3) reframed difficulty-to-novelty shift as adaptive OE behavior in Appendix D.3. These changes, combined with our already-hedged main claims, accurately position AC/DC's contributions within the framework of demonstrable time-bounded open-endedness.
>
> [1] [Open-Endedness is Essential for Artificial Superhuman Intelligence, 2024, Hughes et al.](https://arxiv.org/abs/2406.04268)
>
> [2] [Automated Capability Discovery via Foundation Model Self-Exploration, 2025, Lu et al.](https://arxiv.org/abs/2502.07577)
>
> ---
> > Q: Table 3 shows AC/DC beats DNS/CQD (both trained on benchmarks), but DNS/CQD use different model families and training data. Can you compare all three methods on the same model family with identical conditions?
>
> We thank the reviewer for this detailed question.
>
> In Appendix D.5 of the original submission, we lay out the design of our training and task force selection for the DNS and CQD baselines. Yet, indeed, we did not explicitly mention that, for a fair comparison, we do train on the same model family as AC/DC. We have updated the D.5 section accordingly to state this fact.
>
> The training data for DNS and CQD differ from that for AC/DC. Both baseline methods directly optimize on the downstream benchmark’s training data. It is challenging to compare directly with prior work, since, to the best of our knowledge, we are the first to develop a coevolution algorithm combining QD with synthetic data generation. Nevertheless, to provide the best possible comparison, we follow the procedure described in the CQD paper and adapt the training data to better match our downstream benchmark selection.

---

> > ### Author Response · Authors · 2025-11-26
> > **Thank you Reviewer QSy7 (7/7)**
> >
> > > Q: Why does optimizing for synthetic task Coverage transfer to unseen benchmarks? Is this due to: (a) the scientist LLM's implicit benchmark knowledge, (b) genuine skill generalization, or (c) synthetic tasks being representative of general problem-solving? Can the authors ablate using a scientist LLM with no benchmark exposure?
> >
> > We thank the reviewer for this interesting question and provide our analysis below.
> >
> > Re. (a): We don’t believe it is due to the LLM’s implicit benchmark knowledge, since we have demonstrated in our human evaluation that the generated tasks are very different (OOD) than the benchmark tasks (see the global comment and the newly added Section 5.3 and Appendix H in the paper).
> >
> > Re. (b) and (c): We believe that the tasks generated are very diverse, testing for different core skills that traditional benchmarks don’t. Moreover, we don’t integrate explicit “training”/”optimization” in our pipeline. Since our synthetic task pool is comparatively small compared to standard training datasets (global task pool being ~1000 samples) and is adapting constantly (every five generations), it is designed to evaluate more and more diverse skills, such that we discover diverse models with complementary capabilities. Since our algorithm is designed to select for diversity (and quality), we are inherently “optimizing” for models with diverse skills that can complement each other, rather than any particular skill.
> >
> > As evidenced by our ablation in Table 11, we see that our synthetic dataset does allow for discovering unique and complementary models, since selecting a task force that optimizes for Coverage on that dataset outperforms selecting the highest fitness individuals and randomly selected ones.
> >
> > Finally, it is difficult to identify any modern LLMs that have not had exposure to the downstream benchmark data. Nevertheless, our human evalation supports the claim that our data is novel and OOD.
> >
> > ---
> > We hope that our responses and revisions satisfactorily address the reviewers' concerns. If so, we kindly ask the reviewers to consider re-evaluating their ratings in light of these improvements.

---

> > > ### Author Response · Authors · 2025-12-03
> > > **Follow-up**
> > >
> > > # Static synthetic dataset ablation
> > >
> > > We thank the reviewer for the suggestions to improve our work by adding a further ablation to explore the efficacy of coevolution, compared to model evolution on a static, synthetically generated dataset.
> > >
> > > In addition to our added argumentation in our initial response, we have added the new full ablation to the paper (Appendix D.6) that further validates our approach:
> > >
> > > **Static vs. Coevolving Synthetic Dataset (Table 13):** We compare task forces evolved on a static synthetic dataset versus our full coevolution approach. The coevolving approach achieves absolute improvements of +0.11% (N=3) and +3.62% (N=8) in average coverage, with particularly pronounced gains on knowledge-intensive benchmarks (MMLU: +8.42% at N=8, MMLU Pro: +8.93%) and reasoning tasks (GPQA: +3.36%, BBH: +3.22%). These results directly demonstrate that dynamic coevolution of the synthetic task archive is critical for discovering diverse and complementary model capabilities.
> > >
> > > The ablation further supports our core claims and provides additional insights into the mechanisms underlying AC/DC's success.
> > >
> > > # Variance across re-runs
> > > We thank the reviewer again for their suggestion to improve on statistical rigor in our findings. We have conducted a reproducibility analysis across a few independent runs (2-3 runs per condition) for the Qwen 2.5 model family, reporting standard deviations and ranges in raw scores across all benchmarks.
> > >
> > > At task force size N=3, our method demonstrates comparable variance to the control baseline (mean std dev: 0.64 vs 0.94 points), indicating stable reproducibility. At N=8, our method shows moderately higher variance (mean std dev: 1.80 vs 0.59 points), which is expected given the increased complexity of evolutionary search over larger populations.
> > >
> > > Importantly, certain benchmarks (GPQA judge, Minerva) consistently exhibit higher variance across both methods, reflecting the inherent difficulty of these tasks rather than method-specific instability. Despite this variance, the statistically significant performance improvements remain robust, as our bootstrap hypothesis testing explicitly accounts for cross-benchmark variance.
> > >
> > > We have added a detailed "Reproducibility Analysis" subsection (Appendix K.6) presenting per-benchmark statistics and demonstrating that observed variance levels are acceptable for evolutionary methods and do not undermine our main findings.
> > >
> > > We thank the reviewer again for helpful feedback that ultimately led to notable improvements to our paper!

---

### Official Review · Reviewer_QsVv · 2025-11-03

**Soundness:** 2
**Presentation:** 2
**Contribution:** 3
**Rating:** 6
**Confidence:** 2

**Summary:**

This paper introduces AC/DC, which performs both Model Archive Evolution and Task Archive Evolution to achieve effective collective intelligence. It enables models with smaller parameter scales to reach performance comparable to larger models. Experiments on benchmarks such as MMLU, GPQA, Minerva, and HumanEval demonstrate the effectiveness of the proposed method.

**Strengths:**

- The experiments in this paper are fairly extensive and include plenty of implementation details.
- I think the topic of evolving multiple base LLMs to create a more intelligent model is interesting and important.

**Weaknesses:**

- The Task Archive Evolution part still relies on a larger scientist LLM to generate tasks, which means it depends on external assistance and doesn't necessarily ensure the generated tasks are high quality.
- Although the method looks promising, the whole process, including Model Archive Evolution and Task Archive Evolution, involves too many implementation details. I think the paper lacks enough ablation studies to show the effectiveness of each design choice.
- Overall, this paper is really hard to follow. I strongly recommend the authors add clearer explanations of the full framework, including how LLM parameters are updated and how new tasks are generated.

I am generally inclined to accept this paper. However, since the proposed method involves many implementation details and lacks sufficient ablation studies, I'm not confident that all the technical components are truly useful, so I would lower my confidence level.

**Questions:**

See above

---

> ### Author Response · Authors · 2025-11-26
> **Thank you Reviewer QsVv (1/2)**
>
> We thank the reviewer for acknowledging the extensive experimentation and implementation details in our work, a strength also recognized by Reviewers Y9hf and ZzV2. We are particularly encouraged by the appreciation of the motivation and importance of our model combination setting (also noted by Reviewer PHpK) and the recognition of the parameter efficiency and collective intelligence demonstrated by our approach, a core contribution highlighted by all reviewers (QSy7, Y9hf, PHpK, ZzV2).
>
> ---
> > The Task Archive Evolution part still relies on a larger scientist LLM to generate tasks, which means it depends on external assistance and doesn't necessarily ensure the generated tasks are high quality.
>
> We thank the reviewer for pointing out this concern and initiating further research to robustify our claims, fundamentally improving our work.
>
> To address this concern, we perform a human evaluation to assess the quality and diversity of the synthetically generated tasks.
>
> By leveraging a larger LLM and a quality assurance pipeline (via self-validation and code execution tests), we continuously generate increasingly novel and complex synthetic tasks, without relying on and actively conditioning downstream benchmark training data. Moreover, we design our system to select increasingly novel tasks, i.e., different from existing tasks, via a cosine-similarity and LLM-based novelty filtering process.
>
> We argue that our synthetic data is out-of-distribution (OOD) with respect to the benchmark tasks, and that, notably, we beat baselines that actively optimize for those benchmarks.
>
> Reviewers PHpK and ZzV2 also suggest strengthening the claims that synthetic tasks are correct and OOD with respect to our downstream benchmarks by including validation through a human evaluation.
>
> In the following, we discuss our methodology, which ensures reliability, mitigates biases, and supports our quality, OOD, and creativity claims. We added the full details to the main paper in Appendix H and a new section to the main paper, Section 5.3.
>
> **Methodology.** We conducted a human evaluation with 94 independent assessments (45 synthetic, 49 benchmark) across three reviewers. To ensure unbiased evaluation, we implemented: (1) blind evaluation—task sources were completely hidden from reviewers, (2) balanced 50/50 sampling probability between synthetic and benchmark tasks, (3) similarity-based calibration—showing reviewers the 3 most similar benchmark tasks via embedding retrieval to ground OOD assessments in actual distributional differences, and (4) standardized evaluation criteria with detailed written guidelines.
>
> **Results.** Synthetic tasks achieved 97.8% ± 2.2% correctness and 68.9% ± 6.9% OOD rating, demonstrating high quality while exhibiting substantial distributional novelty. In contrast, benchmark tasks showed only 10.2% OOD rating. Critically, this was concentrated exclusively in the most challenging benchmarks (MMLU-Pro: 33.3%, GPQA: 16.7%), while all standard benchmarks (BBH, GSM8K, MATH, HumanEval, MBPP, MMLU) received 0% OOD ratings. This pattern validates our protocol: reviewers correctly recognized that graduate-level questions may appear novel due to complexity, while appropriately identifying standard benchmarks as in-distribution.
>
> **Statistical Validation.** To analyse the inter-rater reliability, we perform chi-squared analysis, which confirmed strong agreement on objective metrics (correctness: p=0.46, OOD: p=0.57), demonstrating our findings are robust and not driven by individual annotator biases. These results provide strong empirical evidence that our approach generates high-quality, novel tasks that are genuinely out-of-distribution with respect to established benchmarks.

---

> > ### Author Response · Authors · 2025-11-26
> > **Thank you Reviewer QsVv (2/2)**
> >
> > > Although the method looks promising, the whole process, including Model Archive Evolution and Task Archive Evolution, involves too many implementation details. I think the paper lacks enough ablation studies to show the effectiveness of each design choice.
> > > Overall, this paper is really hard to follow. I strongly recommend the authors add clearer explanations of the full framework, including how LLM parameters are updated and how new tasks are generated.
> >
> > We thank the reviewer for acknowledging our work as promising and for suggesting that we include additional robustification measures for our proposed algorithm and its components. We agree that presenting a novel paradigm alongside a novel framework presents challenges.
> >
> > To address the challenge of justifying each design choice, we revised the submission to include an extensive appendix section (Section N) detailing each of our design choices and their justification grounded in established open-endedness literature. We provide a summary in the shared global comment. Moreover, to directly address the question about how LLM parameters are updated, we have added Appendix Section M, detailing the crossover and mutation operations.
> >
> > To aid in the understanding of the full pipeline as a whole, in the original submission, we present Figure 2 and Algorithm 1 and discuss them in detail in Section 3. The newly added appendix section N successfully supplements the understanding of our proposed algorithm by explaining and justifying each component.
> >
> > We discuss the novel implementation details of our components in Sections 2 and 3. Additionally, we provide and cross-reference all LLM prompts used for the respective stages, which further aid in how the individual steps for task generation, task quality assurance, and the novelty and gibberish filters are implemented.
> >
> > Finally, for the specific novel implementations, in the original submission, we designed hold-one-out and remove-all ablations, demonstrating the effect of all relevant components. To improve the robustness of those analyses, we conducted comprehensive bootstrap-based statistical testing across all ablation experiments and revised our paper accordingly.
> >
> > To summarize our findings on the effectiveness of each design choice, we observe the following
> > - Each algorithmic component contributes meaningfully to performance
> > - At N=8, removing all components simultaneously causes highly significant degradation (p<0.00005, CI: 0.691:0.820:0.938)
> > - Quality-diversity selection and gibberish filtering are the most critical components
> >
> > With these findings, we robustly demonstrate that removing each individual component leads to weaker downstream performance than the full method, where removing all components leads to a significantly lower Coverage. Also, in Table 11, we also ablate the selected strategy for the final task force, presenting evidence that our proposed approach is superior to alternatives.
> >
> > We provide extended details below, and the full details in the newly added Appendix K.
> >
> > **Statistical Significance Analysis (New Appendix K)**
> >
> > We have conducted comprehensive bootstrap-based statistical testing across all experimental conditions to rigorously assess the significance of our findings. Our analysis employs min-max normalization to ensure fair comparison across benchmarks with different difficulty levels, followed by bootstrap resampling (50,000 iterations) to compute 95% confidence intervals and one-tailed p-values for performance improvements.
> >
> > **Key Statistical Findings:**
> >
> > *Coverage Results:*
> > - At N=3, AC/DC significantly outperforms all main baselines when aggregated across 8 benchmarks and 4 model families: Control (p=0.01170), Experts (p=0.00584), and Big Model (p=0.00004)
> > - At N=8, improvements become even stronger: Control (p=0.00792), Big Model (p<0.000005)
> > - Qwen 2.5 and Qwen 3 task forces significantly outperform GPT-4o in coverage at N=8 (p=0.0014 and p=0.0057, respectively)
> > - On knowledge recall benchmarks specifically, Qwen 3 at N=8 demonstrates exceptional performance against Big Model (p<0.00005) and GPT-4o (p<0.00005)
> >
> > *Best-of-N Results:*
> > - At N=3, AC/DC significantly outperforms Control baseline (p=0.0002) across all model families
> > - At N=8, strong aggregate performance: Control (p=0.0061), Big Model (p<0.00005)
> > - Individual model families show particularly strong results—Qwen 2.5 and Qwen 3 significantly outperform all baselines including GPT-4o at N=8 (all p<0.0053)
> >
> > *Ablation Study:*
> > - Each algorithmic component contributes meaningfully to performance
> > - At N=8, removing all components simultaneously causes highly significant degradation (p<0.00005, CI: 0.691:0.820:0.938)
> > - Quality-diversity selection and gibberish filtering are the most critical components
> >
> > *Comparison to Alternative QD Methods:*
> > - At N=8, AC/DC demonstrates clear statistical advantages over CycleQD (p<0.00005) and DNS (p=0.0078)

---

> > > ### Author Response · Authors · 2025-11-26
> > > **Thank you Reviewer QsVv**
> > >
> > > We hope that our responses and revisions satisfactorily address the reviewers' concerns. If so, we kindly ask the reviewers to consider re-evaluating their ratings in light of these improvements.

---

> > > > ### Author Response · Authors · 2025-12-03
> > > > **Follow-up**
> > > >
> > > > We thank the reviewer for the suggestions to improve our work by adding further ablations.
> > > > In addition to our added ablations in our initial response, we have added two new ablation studies to the paper that further validate our approach (Appendix D.6 and D.7):
> > > >
> > > > **Static vs. Coevolving Synthetic Dataset (Table 13):** We compare task forces evolved on a static synthetic dataset versus our full coevolution approach. The coevolving approach achieves absolute improvements of +0.11% (N=3) and +3.62% (N=8) in average coverage, with particularly pronounced gains on knowledge-intensive benchmarks (MMLU: +8.42% at N=8, MMLU Pro: +8.93%) and reasoning tasks (GPQA: +3.36%, BBH: +3.22%). These results directly demonstrate that dynamic coevolution of the synthetic task archive is critical for discovering diverse and complementary model capabilities.
> > > >
> > > > **Effect of Seed Tasks (Table 14):** We ablate the composition of seed tasks, showing that including vs. excluding the code generation seed task leads to predictable trade-offs—improved coding performance (+3.25% on HumanEval at N=8) at the cost of reduced performance on other benchmarks (-5.51% on MMLU judge, -3.33% on MMLU Pro judge). This validates that seed task selection influences the evolutionary trajectory while confirming the robustness of our coevolutionary framework.
> > > >
> > > > Both ablations further support our core claims and provide additional insights into the mechanisms underlying AC/DC's success.
> > > >
> > > > We thank the reviewer again for helpful feedback that ultimately led to notable improvements to our paper!

---

### Author Response · Authors · 2025-11-26
**Global Comment**

We thank the reviewers for acknowledging our work as a novel and important contribution to the research community (Reviewers PHpK, ZzV2) that effectively demonstrates collective intelligence through parameter-efficient model combinations (Reviewers QsVv, QSy7, Y9hf, PHpK, ZzV2). We are particularly encouraged that you found our extensive empirical validation across multiple model families (Reviewers QSy7, PHpK) compelling, and that you appreciated the comprehensive qualitative and quantitative analysis supporting our findings (Reviewers QSy7, PHpK, ZzV2).

We greatly appreciate your reviews, which help us clarify ambiguities and further improve the paper. We have revised our paper to reflect these clarifications and improvements, with all changes highlighted in blue for your convenience.

As a summary of our responses:

## Writing and Presentation

* Enhanced algorithmic transparency with an extensive overview of design choices grounded in open-endedness literature (new Appendix N) and details on LLM parameter updates (new Appendix M)
* Added detailed methodology and results for merging compatibility analysis (new Appendix J)
* Expanded synthetic task generation methodology with quality assurance details (new Appendix H, new Section 5.3)

## Experiments and Analysis

* **Statistical significance testing** across all experimental conditions using bootstrap analysis with 50,000 iterations (new Appendix K). Key findings: At N=3, AC/DC significantly outperforms all main baselines in coverage (p<0.05), with improvements becoming stronger at N=8. In the best-of-N metrics, AC/DC significantly outperforms the control baselines (p<0.01). No baseline significantly outperformed AC/DC in any comparison.

* **Human evaluation of synthetic tasks** with 94 independent assessments across three blind reviewers (new Appendix H, new Section 5.3). Results: 97.8% ± 2.2% correctness and 68.9% ± 6.9% OOD rating for synthetic tasks, compared to only 10.2% OOD for benchmark tasks (concentrated in graduate-level benchmarks). Chi-squared analysis confirmed strong inter-rater reliability (p>0.45).

* **Merging compatibility analysis** examining weight-space geometry across five model families (new Appendix J). Finding: the pairwise distance "similarity ratio" (min/max distance among experts) predicts merging success with 80% accuracy; ratios <0.1 indicate good compatibility and >0.7 indicate poor outcomes.

* **Computational cost analysis** demonstrating ~4 GPU hours per merged model (324 total GPU hours for 80 models), significantly more efficient than standard post-training approaches requiring 10² to 10⁵ GPU hours per model (New Appendix L).

We hope that our responses and revisions satisfactorily address the reviewers' concerns. If so, we kindly ask the reviewers to consider re-evaluating their ratings in light of these improvements.

---

### Meta-Review · Area_Chair_Pj9e · 2025-12-06

**Summary:**

This paper introduces Assessment Coevolving with Diverse Capabilities (AC/DC), which performs both Model Archive Evolution and Task Archive Evolution to achieve effective collective intelligence. It enables models with smaller parameter scales to reach performance comparable to larger models. Experiments on benchmarks such as MMLU, GPQA, Minerva, and HumanEval demonstrate the effectiveness of the proposed method.

While the paper received generally positive reviews, two positive reviews seem to be LLM-generated (Reviewer QSy7, score 8) and of low quality (Reviewer Y9hf, score 6). The authors have addressed some of the reviewers' concerns during rebuttal by providing a significant amount of new results. I'm inclined to vote for acceptance of this paper, and also encourage the authors to further polish the organization and writing of their paper in the final version.

**Reviewer Concerns:**

Some concerns that I think are still not well-addressed:
- Writing issues (Reviewer QsVv, Y9hf)
- Still relies on a large scientist LLM to generate tasks (Reviewer QsVv, PHpK)
- Integration of existing techniques, lack of novelty (Reviewer Qsy7, ZxV2)
- Dependent heavily on empirical testing of initial seed model combinations (ZxV2)


Some comments that I believe have been addressed or partly addressed during rebuttal, based on additional experimental results:
- Complex framework involving many implementation details, needs more ablation (Reviewer QsVv)
- Statistical significance (Reviewer pHpK, QSy7)
- Computational cost seems high, making the efficiency claim incomplete (Reviewer PHpK, QSy7)
- Narrow evolutionary scope, relatively easy tasks (ZxV2, QSy7)

**Reviewer Scores:**

I think the reviewers are likely to maintain their original score.

---

### Decision · Program_Chairs · 2026-01-26

Accept (Poster)